



# Future Retreat of Great Aletsch Glacier and Hintereisferner – application of a full-Stokes model to two valley glaciers in the European Alps

Martin Rückamp[1], Gong Cheng[2], Karlheinz Gutjahr[3], Marco Möller[1], Petri K. E. Pellikka[4,5], and Christoph Mayer[1]

[1]Bavarian Academy of Sciences and Humanities, Munich, Germany
[2]Department of Earth Sciences, Dartmouth College, Hanover, NH, USA
[3]Joanneum Research Forschungsgesellschaft mbH, Graz, Austria
[4]Earth Change Observation Laboratory, Department of Geosciences and Geography, University of Helsinki, Finland
[5]State Key Laboratory for Information Engineering in Surveying, Mapping and Remote Sensing, Wuhan University, Wuhan 430079, China

**Correspondence:** Martin Rückamp (martin.rueckamp@badw.de)

**Abstract.** We simulate the future evolution of two valley glaciers in the European Alps over the course of the 21st century. The model setup combines a numerical realization of full-Stokes ice dynamics and a surface mass balance model forced with the sustained (inline with the Paris Agreement) and highest climate emission scenarios based on CMIP5 and CMIP6 data pools. The initialization of the three-dimensional glacier flow model is based on data assimilation, where a detailed observed ice surface velocity map serves as reference for constraining unknown parameters by means of inversion techniques. This setup is applied to Great Aletsch Glacier (GAG) and Hintereisferner (HEF) to assess their individual responses to climate change in the western and eastern European Alps, respectively. The model results of both glaciers are calibrated with comprehensive glaciological observations over several years to ensure a realistic glacier response in the observation period. The end-of-the-century projections reveal a substantial volume loss of both glaciers: HEF is projected to vanish in the middle of the 21st century regardless of the climate emission scenario. GAG is likely to disappear at the end of the 21st century under high-emission scenarios RCP 8.5 and SSP5-8.5, whereas low-emission scenarios RCP 2.6 and SSP1-2.6 predict a median ice volume reduction of 67.7% [62.2 to 77.6%] and 86.4% [76.2 to 89.4%], respectively (values in brackets correspond to the 17th to 83rd percentile range). Our individual and detailed results of glacier evolution provide well-constrained estimates to complement large-scale modelling efforts. In general, our findings of substantial volume loss at the end of the 21st century align with large-scale modelling outcomes; however, a rough model-intercomparison study reveals a large spread of volume projections with the different glacier models.

## 1 Introduction

The recent retreat of glaciers is unprecedented at least during the last thousand years (IPCC, 2023), and the glacier mass loss is expected to continue in the 21st century (e.g., Marzeion et al., 2020), maintaining their role as a major contributor



to recent sea level rise. Beyond their global impact, glaciers are important for regional water storage and supply, high-alpine ecosystems, hydroelectric power production, tourism and natural hazard regulation in mountain valley regions. Predicting the future development of alpine glaciers is therefore important for various aspects and of high interest for a wide group of stakeholders. In addition to their social and economic importance, glaciers are also one of the most obvious indicators of changes in the climate system. Their geometry responds to variations in atmospheric and climatic conditions through the

interaction between surface mass balance and ice dynamics, results in glacier changes related to typical periods of climatic fluctuations. Therefore, the observation and modelling of glacier changes can be used to better understand past, present and future climatic conditions (Pellikka and Rees, 2009).

In view of their enormous importance, there is great interest in accurately simulating glacier evolution over time. Several models for simulating glacier evolution have been developed, ranging from individual glacier applications (Jouvet et al.,

2009, 2011; Jouvet and Huss, 2019; Zekollari et al., 2014; Réveillet et al., 2015; Peyaud et al., 2020; Gilbert et al., 2020) to regional-scale/global-scale models (Zekollari et al., 2019, 2024; Hanzer et al., 2018; Maussion et al., 2019; Marzeion et al., 2020; Rounce et al., 2020, 2023; Schuster et al., 2023; Hartl et al., 2025). However, due to computational constraints, projecting the temporal evolution of glaciers on large spatial scales (global and regional) requires models based on various simplifications, especially with regards to ice dynamics (GloGEM: Huss and Hock (2015), GloGEMflow: Zekollari et al. (2019), PyGEM:

Rounce et al. (2020), OGGM: Maussion et al. (2019)). In most of these models, ice flow is treated with the Shallow Ice Approximation (SIA), which is computationally efficient but only valid for extensive continuous ice masses (e.g. the interior of ice sheets) with a small aspect ratio (thickness / length). Despite potential shortcomings due to a reduced representation of ice dynamics, large-scale studies of mountain glaciers with such a simplified approach (Zekollari et al. (2019, 2024)) suggest a glacier volume loss of 94-99% in Central Europe by the end of the century for the high-emission scenarios RCP 8.5 (Representa-

tive Concentration Pathway, Moss et al., 2010) and SSP5-8.5 (Shared Socioeconomic Pathways Meinshausen et al., 2020). For the low-emission scenarios RCP 2.6 and SSP1-2.6, which are basically in line with the political global warming target of 1.5°C negotiated in the Paris Agreement (UNFCCC, 2015), an abating volume loss of 63-86% at the end of the century is estimated.

In turn, detailed ice dynamic studies of individual glaciers often focus on specific applications (e.g., to forecast possible

hazards or process understanding), but are also required to assess the impact of simplifications made by large-scale models. Depending on the model complexity, the better representation of physical processes comes with large (or even huge) additional computational costs, which directly transfers into a cumbersome/expensive model tuning and parameter sensitivity evaluation, and even numerical issues (e.g., achieving convergence of a full-Stokes model is challenging). However, with recent advances of performance in computational resources, solving a full-Stokes ice flow model for a complex three-dimensional glacier

geometry has become much more affordable. In the hierarchy of ice flow models, the most comprehensive description of ice flow is given by the full-Stokes equations (Hindmarsh, 2004) and is most accurate for mountain glaciers with steep slopes and a high aspect ratio (Le Meur et al., 2004). However, glacier evolution studies under future climate warming scenarios until the end of this century and utilizing a full-Stokes model, or an appropriate higher-order model, are very scarce. A prominent example is Jouvet and Huss (2019), where they estimated a volume loss of the Great Aletsch Glacier (Switzerland) at the end





of the 21st century ranging from 60% (median of full climate forcings) for RCP 2.6 to an almost complete deglaciation for the RCP 8.5 scenario based on a full-Stokes model.

Large uncertainties remain in projections of glacier volume loss on global and regional scales (Marzeion et al., 2020). These uncertainties are not only due to the glacier model used. The choice of the utilized surface mass balance schemes (e.g., temperature index models versus surface energy balance model (e.g., Gabbi et al., 2014), validation of the model response

to the type of mass balance observation (glacier-specific vs. regional, Zekollari et al., 2024) or climate data input options (daily vs. monthly resolution, downscaling methods and bias corrections of climate data)) have a substantial influence on the simulated glacier development. In addition, a common issue of glacier evolution calculations is related to model initialization. This is a well-known problem in glaciology, as illustrated by recent intercomparison experiments focused specifically on the initialization of ice sheet models (Goelzer et al., 2018; Seroussi et al., 2019). While such an intercomparison has not

yet been conducted for mountain glaciers, the challenge of initialization remains comparable, as highlighted by Zekollari et al. (2022, Sect. 6 therein). We do review the different initialization methods, but the three methods discussed in Zekollari et al. (2022) have their own advantages and shortcomings and need to be carefully selected in terms of the research question, model capabilities and performance. In case of sufficiently available data, the most straightforward method consists of starting simulations from an observed state where unknown parameters are constrained by observations. But the initial glacier evolution

is often subject to an artificial dynamical shock as the model tends to adjust to the numerical environment and not solely respond, e.g. to the imposed climatic boundary conditions. Usually, an arbitrary relaxation is performed to level out the dynamic shock. Initialization approaches avoiding an artificial dynamical shock come with long spin-up times which are unfeasible for a full-Stokes model and often lack a good agreement of the initialized glacial state with observations (both, ice geometry and ice velocity).

In this paper, our aim is to complement the current research of large-scale glacier volume projections of Central Europe with an individual glacier evolution model based on a higher complexity to investigate the potential variability of glacier responses in the frame of physical process representation. Ice dynamics is solved with a full-Stokes model, whereas the coupled surface energy balance model computes the SMB on a daily basis. The initialization of the glacier model is based on data assimilation, constraining unknown parameters by an inversion method with respect to observed ice geometry and a large coverage of ice

velocities across the glacier. The latter is a common initialization approach for large-scale ice sheet modelling (e.g., Goelzer et al., 2020), but to our best knowledge this is the first time applied for mountain glaciers. In order to capture and analyze regional differences of ice volume loss, the model is applied to two valley glaciers in the Alps: Hintereisferner (Austria) is located in the eastern Alps, while the Great Aletsch Glacier (Switzerland) is located in the western Alps. After initialization, we carry out comprehensive model validation to ensure a realistic response of the model in the observation period. Subsequently,

ensemble projections are performed for SSP5-8.5, SSP1-2.6, RCP 8.5 and RCP 2.6 to simulate the glacier evolution under different climate scenarios. Finally, these projections are compared to existing large scale experiments.





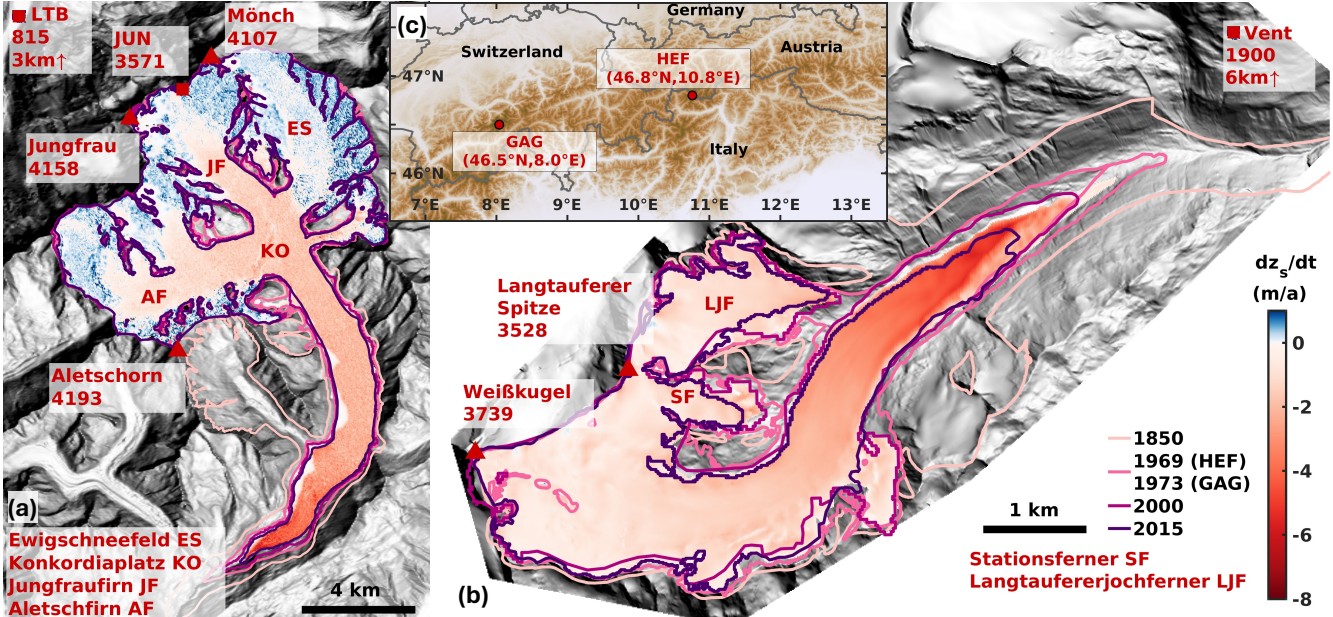

**Figure 1.** Map of Great Aletsch Glacier (GAG, a) and Hintereisferner (HEF, b) and their location in the European Alps (c). For GAG, mean annual elevation changes, $\partial z_s/\partial t$, are calculated from 2011 and 2019 digital elevation models (DEM, Leinss and Bernhard, 2021). For HEF, mean annual elevation changes are calculated from 2001 and 2013 DEMs (Sailer et al., 2017; Strasser et al., 2018). Prominent mountain summits are displayed with triangles and meteorological stations with squares. Note that the stations Lauterbrunnen (LTB) and Vent are outside the figure limits. Glacier outlines are taken from the Randolph Glacier Inventory (RGI v7.0) (RGI Consortium, 2023). The hillshade backgrounds are 2011 and 2001 DEMs for GAG and HEF, respectively.

## 2 Study sites

This study focusses on two valley glaciers in the European Alps. Great Aletsch Glacier (GAG, Switzerland, 46.5°N, 8.0°E) is in the western part, while Hintereisferner (HEF, Austria, 46.8°N, 10.8°E) is in the eastern part. An overview of mean annual

elevation changes in recent years, glacier outlines from the last decades, and the geographical setting is presented in Fig. 1.

GAG is the largest ice mass in the Alps and originates from the northern main ridge of the Alps. Three main tributaries – Aletschfirn, Jungfraufirn and Ewigschneefeld – join at Konkordiaplatz and form a 15 km long curved tongue that extends to the southwest. The glacier covers an area of 82 km² in 1999 (Farinotti et al., 2009), while the elevation ranges from 1600 to 4100 m. The volume of ice amounts to 15 km³ (Farinotti et al., 2009) with ice thicknesses > 900 m at Konkordiaplatz. The

glacier has been retreating since the Little Ice Age and its volume loss has been estimated to be 4.8 km³ in the period 1880 to 1999, while most of this volume loss has occurred since 1980 (Bauder et al., 2007). The climate at the glacier tongue is relatively dry, but very large amounts of precipitation are measured in the glacier accumulation area, as this area is influenced by precipitation events from the north (Schwarb et al., 2001; MeteoSwiss, 2025).




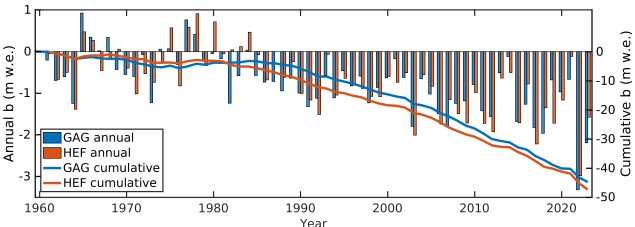

**Figure 2.** Annual (bars) and cumulative (lines) surface mass balance of GAG (blue) and HEF (red) since 1961. Mass balance data are based on GLAMOS (2023) and WGMS (2024) for GAG and HEF, respectively.

HEF is a typical valley glacier near the main ridge of the Eastern Alps in Austria. The glacier originally consisted of three main tributary basins. Langtaufererjochferner and Stationsferner were assumed to be disconnected from the main HEF tongue in 1969 and 2000, respectively. However, they are still treated as part of the glacier to maintain consistency in mass balance assessments. The glaciers show one of the longest time series of observations in the Alps (Klug et al., 2018). The accumulation area extends from the Northeast to the Southeast, whereas the long and narrow curved tongue stretches to the Northeast. Weißkugel (3739 m) is the highest point of the HEF, while the terminus of its tongue lies at an elevation of around 2400 m. Ground Penetrating Radar (GPR) surveys in 1997 and 2002 revealed an ice thickness of the tongue exceeding 200 m (Span et al., 2005; Fischer and Kuhn, 2013), while the mean ice thickness in the upper parts of the glacier is less than 100 m. In 2001, the glacier covered an area of 8.25 km$^2$ with an ice volume of 0.57 km$^3$ (Fischer, 2010; Fischer and Kuhn, 2013). The HEF is located in the inner dry Alpine zone (Frei and Schär, 1998), which is characterized by very low precipitation compared to the rest of the Alps. The average annual precipitation sum in Vent, about 12 km downstream from HEF at 1900 m a.sl., reaches less than 700 mm.

For both glaciers, an exceptional long time series of surface mass balance measurements at stakes over several decades is available (GLAMOS, 2023; WGMS, 2024). Area integrated mass balance terms and spatial distributions are calculated from these measurements with additional data such as snow pit measurements (Huss et al., 2008; Fischer, 2010). Figure 2 shows the annual and cumulative mass balance values for both glaciers. The cumulative mass balance curves reveal a similar behaviour for GAG and HEF. Occasionally, positive years of mass balance occur before 1985. Since then, both glaciers have experienced exclusively negative mass balances, which last until today.

## 3 Climate data

For the period 1961–2023 we used climate variables from the ERA5 reanalysis dataset on a 0.25° spatial and daily temporal resolution (Hersbach et al., 2020). The dataset provides the input variables required for the incoming short- and long-wave radiation, the near-surface air temperature, the surface wind speed, the near-surface specific humidity and the precipitation that are needed to drive the energy balance model (EBM, Sect. 4.3). Note that ERA5 does not provide near-surface specific humidity, but this variable can be computed from dew point temperature and surface pressure (see Appendix A1).





We expect biases of the ERA5 dataset compared to observations at meteorological stations in the vicinity of the individual glacier. Therefore, we applied a simple correction for the near-surface air temperature and precipitation to adjust the ERA5

time series. We use direct measurements for near-surface air temperature and precipitation at a nearby meteorological stations. For GAG, observed precipitation time series are taken from Lauterbrunnen (815 m a.s.l.) and air temperature from Jungfraujoch (3571 m a.s.l.). For HEF, we used the respective time series of temperature and precipitation recorded at the Vent meteorological station (1900 m a.s.l.). The mean annual values of the closest grid cell of the ERA5 dataset is compared with the meteorological station over the period 1961–1990. The period from 1961–1990 is considered as neutral climate period with minor temperature

changes (see ERA5 in Fig. 3a) and almost minor mass balance changes (see Fig. 2). To account for the topographic difference between both data points, we use a temperature lapse rate and a precipitation gradient (see below). The inferred bias is used to shift and scale the global circulation model/regionla climate model (GCM/RCM) time series of near-surface air temperature and precipitation, respectively. Although we used a corrected ERA5 time series, we further term the data ERA5.

Future projections (2023–2100) are based on two different ensemble datasets: (1) Similar to Zekollari et al. (2019), climate

change projections are taken from the EURO-CORDEX ensemble (Jacob et al., 2014; Kotlarski et al., 2014) based on phase 5 of the Coupled Model Intercomparison Project (CMIP5, Taylor et al., 2012). From the entire ensemble, we rely on simulations showing the highest resolution of $0.11°$ (approx. 12 km horizontal resolution) for the emission scenarios RCP 8.5 and RCP 2.6 and simulations which provide the EBM-relevant climate variables on a daily resolution. This selection corresponds to a total of 65 and 22 simulations for RCP 8.5 and RCP 2.6, respectively, consisting of different combinations of thirteen RCM's, six

GCM's and various realizations. (2) As a second data set, we use GCM simulations on a $0.5°$ (approx. 60 km horizontal resolution) regular grid from phase 3b of the Inter-Sectoral Impact Model Intercomparison Project (ISIMIP3b, Lange, 2019). The chosen ISIMIP3b simulations are based on CMIP6 global climate model simulations (Eyring et al., 2016). We run our ice dynamic simulations for two SSPs: the low-emission scenario SSP1-2.6 and the very high-emission scenario SSP5-8.5. Again, we select simulations from the entire ensemble that provide the required climate variables on a daily resolution. For

both scenarios, SSP5-8.5 and SSP1-2.6, ten GCMs are used. A similar dataset has also been employed by Schuster et al. (2023, they use 5 GCMs) using the Open Global Glacier Model (OGGM, Maussion et al., 2019) as a glacier projection tool.

Usually, GCMs and RCMs have systematic biases in their output caused by various factors (e.g. Wood et al., 2004; Maraun, 2016). Therefore, the outputs of global climate models cannot be used directly at local scales to assess the impacts of climate change. Errors or biases are due to limited spatial resolution (large grid sizes), simplified processes and physics, or incomplete

understanding of the global climate system. To overcome these biases, downscaling is essential to adjust the statistical properties of the raw GCM/RCM outputs to be consistent with local climate conditions. The time series of all individual GCM/RCM outputs have been downscaled to the ERA5 grid cell closest to the center point of the GAG and HEF using detrended quantile mapping techniques (DQM) (e.g. Chadwick et al., 2023); note that the ERA5 temperature and precipitation time series have already received a bias adjustment (see above). This procedure is similar to Jouvet and Huss (2019), but we use the ERA5 dataset

as an observational basis instead of measurements at meteorological stations, since downscaling can be performed for the six required variables (which are not or only partially available at the respective metrological stations). Note that ISIMIP3b GCMs





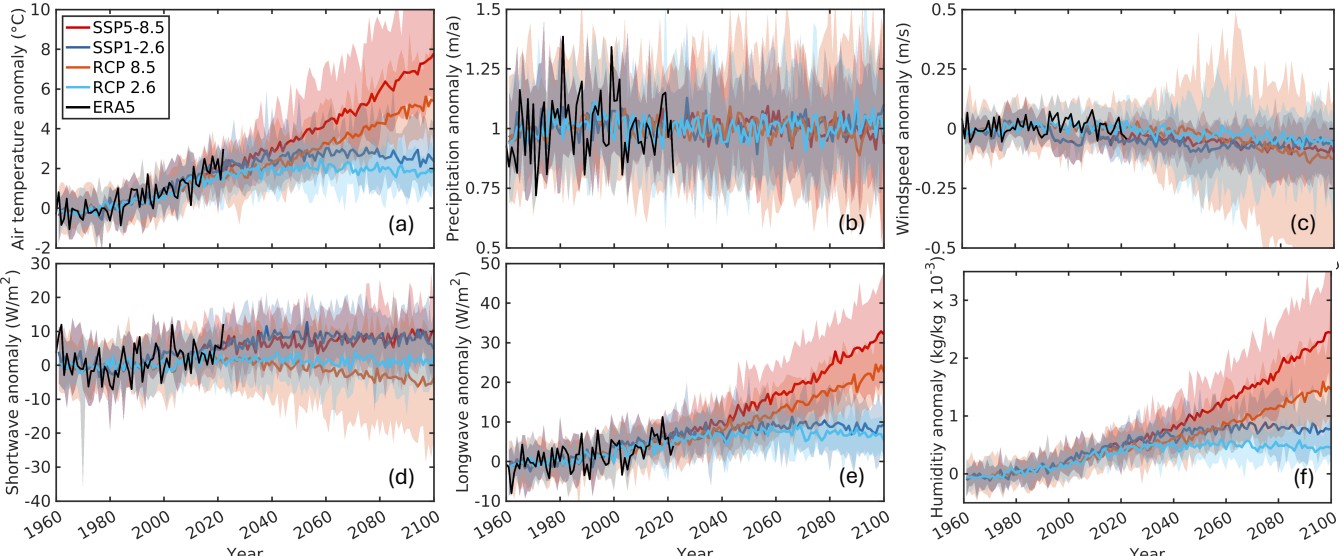

**Figure 3.** Air temperature (a), precipitation (b), windspeed (c), shortwave radiation (d), longwave radiation (e) and humidity (f) anomalies between 1961 and 2100 relative to 1961–1990 of the downscaled GCM/RCM time series of the EURO-CORDEX (RCP scenarios) and ISIMIP3b ensembles (SSP scenarios). The straight lines show the ensemble mean, the lighter background shading covers the area between the ensemble minimum and maximum of each scenario. All variables are additive relative to the reference period, except for precipitation, which is multiplicative.

are already internally bias-adjusted to W5E5 over the period 1979–2014 (Lange, 2019). However, an additional adjusting of the ISIMIP3b ensemble ensures consistency with the EURO-CORDEX ensemble.

Figure 3 displays the temporal evolution of annual incoming short- and longwave radiation, near-surface air temperature,
surface wind speed, near-surface specific humidity, and precipitation change averaged over the Alps for SSP5-8.5, SSP1-2.6, RCP 8.5 and RCP 2.6. The high emission scenarios SSP5-8.5 and RCP 8.5 reveal a mean increase in temperature of approximately $7.8 \pm 1.7$ and $5.4 \pm 1.1°$C by 2100 compared to pre-industrial levels, respectively. The low emission scenarios SSP1-2.6 and RCP 2.6 show a warming of $1.9 \pm 0.4$ and $2.4 \pm 1.1°$C by 2100, which is above the political global warming target of $1.5°$C negotiated in the Paris Agreement (UNFCCC, 2015). As expectable, changes in longwave radiation and humidity
reveal a similar pattern: The largest increase happens in SSP5-8.5 followed by RCP 8.5, SSP1-2.6 and RCP 2.6. Changes in precipitation, windspeed, and shortwave radiation are low and not very pronounced.

Since the adjusted GCM/RCM time series are representative for the elevation at the selected ERA5 grid cell (2246 m a.s.l. and 2175 m a.s.l. for GAG and HEF, respectively), we apply elevation gradients for mapping the data to the glacier area. For the near-surface air temperature, a gradient of $-6.5°$C km$^{-1}$ is used (Strasser et al., 2018). Downward long- and short-
wave radiation are also assumed to decrease with elevation, showing gradients of $-29$ W m$^{-2}$ km$^{-1}$ and $-13$ W m$^{-2}$ km$^{-1}$, respectively (Marty et al., 2002). Precipitation is assumed to increase with elevation, following a gradient of $0.35$ m a$^{-1}$ km$^{-1}$, which has been used for modelling of the GAG before (Jouvet et al., 2011; Jouvet and Huss, 2019). For windspeed no gradient





is applied due to lack of empirical data. We acknowledge that the applied elevation gradients for the different climate variables
might not be fully representative for both glaciers, GAG and HEF, but the general lack of related observational data justify this

approach. We also intend to use the same gradients to facilitate the comparison of the modelling results.

## 4  Model description

We employ the Ice-sheet and Sea-level System Model (ISSM; Larour et al., 2012). The applied setup consists of an ice flow
component (Sect. 4.1), a glacier evolution component (Sect. 4.2), and a surface mass balance component (Sect. 4.3). All
components are combined to simulate the temporal evolution of the glacier.

### 4.1  Ice dynamics

In the case of glaciers, the Reynolds number is very low (e.g., Fowler and Larson, 1978) and therefore the inertial term in the
Navier–Stokes equations can be neglected. This approximation is often referred to as Stokes flow, and the dynamics of the ice
flow component is solved with the full-Stokes (FS) equation. The core of the FS equation system builds on the mass balance
and momentum equation for incompressible ice and is written as

$$\operatorname{div} \boldsymbol{v} = 0, \tag{1}$$

$$\operatorname{div} \mathbf{t} = \varrho_i \boldsymbol{g}, \tag{2}$$

with the density of ice $\varrho_i = 917 \, \mathrm{kg \, m^{-3}}$, the three-dimensional velocity field $\boldsymbol{v} = (v_x, v_y, v_z)$ in Cartesian coordinates, the
gravitational acceleration vector pointing downward $\boldsymbol{g} = -g\boldsymbol{e}_z$ ($g = 9.81 \, \mathrm{m \, s^{-2}}$), and the Cauchy stress tensor $\boldsymbol{t}$. We split the
Cauchy stress into a deviatoric part $\mathbf{t}^{\mathrm{D}}$ and an isotropic pressure $p$

$$\mathbf{t} = \mathbf{t}^{\mathrm{D}} + p\mathbf{I}, \tag{3}$$

with $p = -\frac{1}{3}\operatorname{tr}(\mathbf{t})$ and $\mathbf{I}$ the identity tensor. The constitutive equation for the non-Newtonian fluid links the stress tensor to
strain rates (i.e., velocity gradients).

$$\mathbf{t}^{\mathrm{D}} = 2\eta \mathbf{D}, \tag{4}$$

where $\mathbf{D}$ is the strain rate tensor ($\mathbf{D} = \frac{1}{2}\left(\operatorname{grad} \boldsymbol{v} + (\operatorname{grad} \boldsymbol{v})^T\right)$). The viscosity is given by the Glen-Steinemanns flow law (Glen,
1955; Steinemann, 1954)

$$\eta = \frac{1}{2}B\dot{\varepsilon}_e^{(1-n)/n}, \tag{5}$$

with the flow law exponent $n = 3$, the ice hardness $B$, and the effective strain rate $\dot{\varepsilon}_e$ being the second invariant of the strain-
rate tensor. The ice hardness parameter usually covers the temperature dependence of ice deformation and occasional effects
like softening due to crevasses. Here, $B$ is inferred by an inversion approach (see Sect. 5.1).





The glacier base is subject to basal sliding according to a friction law for the basal shear stress $\boldsymbol{\tau}_\mathrm{b}$ in the tangential plane

$$\boldsymbol{\tau}_\mathrm{b} = -\beta^2 \boldsymbol{v}_\mathrm{b}, \tag{6}$$

$$\boldsymbol{v} \cdot \boldsymbol{n} = 0, \tag{7}$$

with the unit normal vector $\boldsymbol{n}$ pointing out of the ice, the basal drag parameter $\beta^2$, and $\boldsymbol{v}_\mathrm{b}$ is the velocity in the tangential plane at the base. Basal refreezing or melting is neglected. The basal drag parameter for both glaciers is inferred by an inversion

approach (see Sect. 5.1). The boundary conditions at the ice base are enforced by Nitsche's method (weak implementation, which reveals better convergence and smoother results than the strong implementation for both glaciers, Cheng et al., 2020).

### 4.2    Glacier evolution

The glacier surface, $z_s(x, y, t)$, is updated at every time step through the free surface equation (Greve and Blatter, 2009):

$$\frac{\partial z_s}{\partial t} = \boldsymbol{v}_\mathrm{s} \cdot \boldsymbol{n} + \mathrm{a}_s, \tag{8}$$

where $\mathrm{a}_s = a_s(x, y, t)$ is the surface mass balance. Assuming an impenetrable glacier bed, $z_b$, the surface $z_s$ must meet the condition: $z_s - H_\mathrm{min} > z_b$. If $z_s$ falls below $z_b + H_\mathrm{min}$ the surface height is constrained to $z_b + H_\mathrm{min}$. The minimum ice thickness, $H_\mathrm{min}$, is set to $5\,\mathrm{m}$ and ensures numerical stability. We assume that the ice base, $z_b$, is stationary.

The horizontal extension of the glacier is calculated with a level-set method (LSM, Bondzio et al., 2016), i.e. the glacier front advances horizontally with the ice velocity (no frontal/ice-cliff melt is considered). However, a thickness constraint of

$H_\mathrm{min} = 5\,\mathrm{m}$ deactivates/activates elements that fall below/exceed this threshold.

### 4.3    Surface mass balance

Equation 8 needs a reliable SMB at the glacier surface. Here, we rely on an energy balance model that computes the glacier melt. Snow accumulation, $P_s$, is calculated from total precipitation, $P$. The total precipitation is separated into liquid precipitation (rainfall) and solid precipitation (snowfall): Above a temperature threshold of $2°C$ precipitation is assumed to be liquid, below

$0°C$ precipitation is completely solid. We used a smooth cosine interpolation to retrieve the precipitation fraction from solid to liquid between the temperature thresholds of 0 and $2°C$.

For ice melt, we employ the surface energy balance model (EBM) by Evatt et al. (2015). The EBM was originally designed for debris-covered glaciers as it accounts for energy fluxes in a dry porous debris layer on the glacier surface. Both glaciers considered here are mostly debris-free, and therefore the EBM reduces to a clean-ice scheme, where a modulation of the surface

heat flux due to a porous supraglacial surface layer is absent. For an exhaustive description, the reader is referred to Evatt et al. (2015). The EBM is forced with the daily near-surface temperature, short- and longwave downward radiation, near-surface windspeed and near-surface specific humidity time series taken from the selected EURO-CORDEX RCMs and the ISIMIP3b GCMs (see Sect. 3). The SMB scheme is run on a daily timestep, but the ice flow model (IFM) is forced with the yearly SMB, as we are interested in the long-term response and not the seasonal variations of the glaciers.



A crucial ingredient for calculating the surface mass balance is the surface albedo $\alpha$. Here, we use a simple parametrization to calculate the effective albedo which follows the idea of Oerlemans (1992) to make the albedo dependent on the equilibrium line altitude (ELA).

$$\alpha = \alpha_{\text{ice}} + (\alpha_{\text{snow}} - \alpha_{\text{ice}})(1 + \tanh(\pi(z_s - \text{ELA} + \Delta)/\delta))/2, \tag{9}$$

where $\alpha_{\text{ice}}$ is the albedo of bare ice, $\alpha_{\text{snow}}$ is the albedo of snow, $\Delta$ a tuneable value, and ELA is the spatially mean ELA of the glacier. The parameter $\delta$ ensures a smooth transition from snow to bare ice albedo. Note that the unknown albedo parameters will be tuned to resemble a realistic surface mass balance (SMB) profile over HEF and GAG (see Sect. 5.2). We also compared our albedo scheme with a more sophisticated albedo parameterization that includes a snow thickness (Oerlemans and Knap, 1998), a temperature-dependent (Slater et al., 1998) or age-dependent snow albedo (Oerlemans and Knap, 1998), but found that the added value is too small given the increase in model complexity and computational cost.

We remain with the same model parameters as in Evatt et al. (2015, Tab. 1), but climate variables are replaced with the daily time series of the GCM/RCM inputs, and we apply the albedo parametrization (Eq. 9). Note that the thickness of the debris is zero.

### 4.4 Numerical Model

Numerical solutions of the described ice flow model and related modules are obtained using ISSM. The governing equations are unstable if discretized using the Galerkin finite element method. For the FS equations (Eq. 2), we use condensed MINI elements (Gresho and Sani, 2000) to fulfil the compatibility Ladyzhenskaya-Babuška-Brezzi (LBB) condition (Larour et al., 2012). The computations are run in parallel using the iterative GMRES solver preconditioned using the Additive Schwartz Method (Widlund and Toselli, 2005) with an overlap of 1.

The free surface equation (Eq. 8) is hyperbolic and stabilization is achieved by streamline upwind Petrov–Galerkin diffusion (SUPG, Brooks and Hughes, 1982). We use triangular Lagrange P1 elements (piecewise linear). Similarly, the LSM method is stabilized with SUPG and P1 elements are employed. The reinitialization frequency is set to the ISSM default value of 5. Both equations are solved with GMRES preconditioned with Block Jacobi. However, the numerical handling of the LSM method is studied in detail by Cheng et al. (2024).

The mesh is generated from a regular 2D triangular grid with an edge length of 25 m, and extruded vertically into five equally spaced layers. We ensure that each time step complies with the Courant-Friedrichs-Levy condition (Courant et al., 1928) for numerical stability. Having expected maximum ice velocities of about $200 \, \text{m} \, \text{a}^{-1}$ at GAG and $40 \, \text{m} \, \text{a}^{-1}$ at HEF we chose a time step of 0.05 and 0.25 a, respectively, for the transient simulations.

### 5 Initialization

When simulating glacier temporal evolution, initializing a glacier model is a major challenge that can have a large impact on future projections (e.g., Goelzer et al., 2018; Zekollari et al., 2022). This is a well-known problem and several methods




**Table 1.** Overview of the data used in the initialization. Step refers to the step used in the initialization procedure: DA – data assimilation (Sect. 5.1), SC – smb gradient calibration (Sect. 5.2), VA – validation (Sect. 5.3). Time range provides the year(s) used in the respective step.

| Data product | step | GAG time range | HEF time range |
|---|---|---|---|
| Bed topography | DA | Grab et al. (2021) fixed | Lambrecht and Kuhn (2007) fixed |
| Surface topography | DA | Leinss and Bernhard (2021) 2011 | Lambrecht and Kuhn (2007) 1998 |
| Surface velocity | DA | Leinss and Bernhard (2021) mean of 2011–2019 | this study (see Appendix B1) mean of 1995–1996 |
| SMB gradient | SC | GLAMOS (2023) 2011–2022 | Fischer (2010), Fischer et al. (2013) 1998–2022 |
| Elevation change | VA | Leinss and Bernhard (2021) 2011, 2019 | Sailer et al. (2017) 2001, 2013 |
| Mass balance | VA | GLAMOS (2023) 2011–2022 | WGMS (2024) 1998–2022 |

exist which address this problem. All of the proposed methods have their own advantages and drawbacks in order to have an initialized model that resembles both the present-day state and changes of glacier variables (e.g. surface elevation). The choice of the used method is often dependent on the research question, the capability of the model, the availability of observations, or computational resources.

Here, we aim to follow a data assimilation approach: in a first step (DA), the modelled ice surface velocities are inferred to match the observations by means of an inversion technique (Sect. 5.1). In a next step (SC), we calibrate the gradient of the surface mass balance by tuning unknown parameters of the EBM to reproduce observations (Sect. 5.2). In a final step (VA), we run a transient simulation to validate that the transient response matches observations of elevation change and mass balance (Sect. 5.3). Table 1 gives an overview of the data used in the steps described below. Note that the remotely sensed ice surface

velocity map of HEF is derived within this work (Appendix B1).

## 5.1   Data assimilation

The simulations presented make use of observations for initialization to resemble a certain known state of the glacier. This approach requires the contemporaneity of the products to maintain that, e.g. the ice surface velocity is consistent with the ice geometry. For both glaciers, the required datasets are available (see Tab. 1) and are bi-linear interpolated on the 25 m triangular

mesh. The initialization year for GAG is 2011; for HEF it is 1998.





After initializing the geometry with observed data, an inversion approach is used to infer unknown parameters to resemble the ice dynamics. In particular, the basal drag coefficient $\beta$ in Eq. 6, which cannot be measured directly, is inferred using an inversion method (Morlighem et al., 2010). In addition, the ice hardness factor $B$ (Eq. 5) is unknown. Usually, it is recovered by a thermochemical coupled model, since $B$ depends on the temperature (Greve and Blatter, 2009). For reasons of ease of

calculation, we do not use such a model here and assume a constant rheology during the friction inversion. We initialize $B$ by computing a vertical temperature profile based on the solution provided by Robin (1955). The computed temperature profile is transferred to $B$ using the relation given by Cuffey and Paterson (2010, p. 75). However, we update $B$ by a subsequent inversion of ice hardness (Borstad et al., 2013) that uses the inferred friction coefficient. The inferred rheology remains unchanged in the projections.

Both inversions minimize a cost function that measures the misfit between observed horizontal velocities and modelled horizontal velocities $\left( v = \sqrt{v_{\mathrm{x}}^2 + v_{\mathrm{y}}^2} \right)$ (Morlighem et al., 2010, 2013). Observed horizontal surface velocities are available for both glaciers with large coverage (Tab. 1). The cost function is defined as follows:

$$J(\mathbf{v}, \beta) = \gamma \frac{1}{2} \int_{\Gamma_s} \log \left( \frac{\sqrt{v_{\mathrm{x}}^2 + v_{\mathrm{y}}^2} + \varepsilon_v}{\sqrt{v_{\mathrm{x,obs}}^2 + v_{\mathrm{y,obs}}^2} + \varepsilon_v} \right)^2 d\Gamma_s + \gamma_t \frac{1}{2} \int_{\Gamma_b} \nabla(\beta) \cdot \nabla(\beta) \, d\Gamma_b, \tag{10}$$

where $\Gamma_s$ and $\Gamma_b$ are the ice surface and ice base, respectively. The cost function consists of one term that fits the velocities ($J_0$).

The second term ($J_{\mathrm{reg}}$) is a Tikhonov regularization to avoid oscillations. The parameters $\gamma$ and $\gamma_t$ weight the contributions to the cost function. Following Wolovick et al. (2023), $\gamma$ and $\gamma_t$ are defined as follows:

$$\gamma = \mathrm{var}(\log(|v_{\mathrm{obs}}|)) A, \tag{11}$$

$$\gamma_t = \lambda \frac{1}{2} \left( \frac{\pi \sigma_{(k \text{ or } B)}}{\bar{H}} \right) A, \tag{12}$$

where $A$ is the area of the ice domain, $\bar{H}$ is the mean ice thickness, and $\sigma_{(k \text{ or } B)}$ is the standard deviation of the initial guess of

$k$ or $B$, and $\lambda$ is a dimensionless Tikhonov regularization parameter.

We determine the optimal regularization parameter value $\lambda$ using an L-curve analysis for both inversions. The L-curve is a log-log plot of the smoothness of the optimized variable, indicated by the term $J_{\mathrm{reg}}$, and the mismatch between the model and the observations represented by the term $J_0$.

For L-curve analysis of the friction inversion, we sample the range $10^{-2} \leq \lambda \leq 10^3$ with 21 logarithmically spaced samples.

For the analysis of the L-curve inversion of ice hardness, we sample the range $10^{-3} \leq \lambda \leq 10^3$ with 16 logarithmically spaced samples. For each sample, we run the inverse model to convergence and record each contribution to $J$. The results of the analysis of the L-curve of each glacier are shown in Fig. 4. All L-curves provide a suitable monotony for easily picking an optimised trade-off value of $\lambda$. For GAG, the selected lambda value of the L-curve analysis of the basal friction is $\lambda = 3.16$ and $\lambda = 0.25$ for the inversion of $B$; for HEF it is $\lambda = 0.18$ and $\lambda = 1.58$, respectively. The final results of the inferred ice flow

are presented in Fig. 5. For both glaciers, the observed surface velocity is reproduced with great fidelity (Fig. 5c, d); we obtain a root mean square (RMS) error of $2.66 \, \mathrm{m \, a^{-1}}$ for GAG and an RMS error of $0.36 \, \mathrm{m \, a^{-1}}$ for HEF. The maximum ice surface





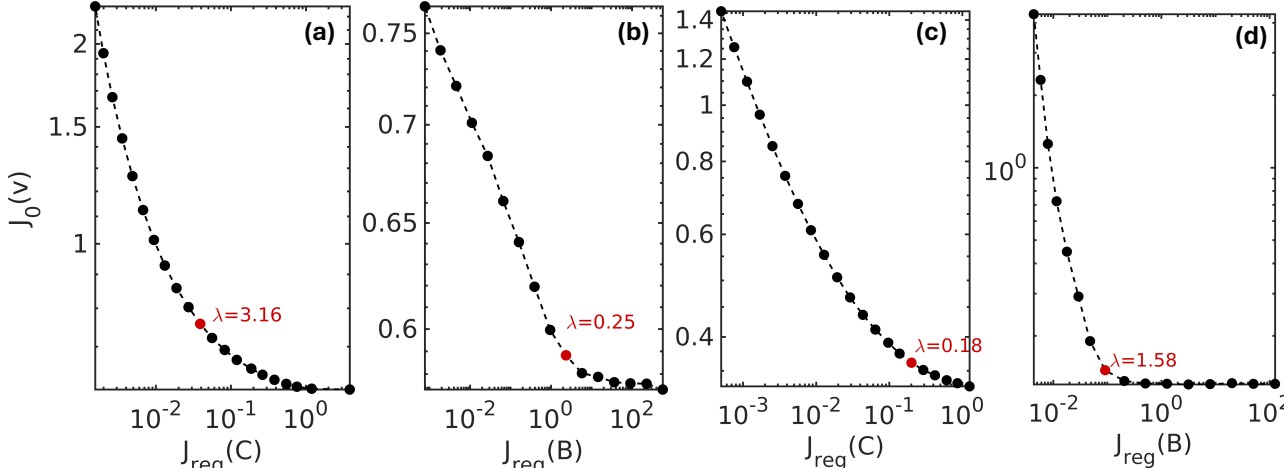

**Figure 4.** L-curves of the inversion for the basal friction coefficient (a, c) and the rheology $B$ parameter (b, d) for HEF (a, c) and GAG (b, d). The selected regularization parameter $\lambda$ of each inversion is highlighted in red.

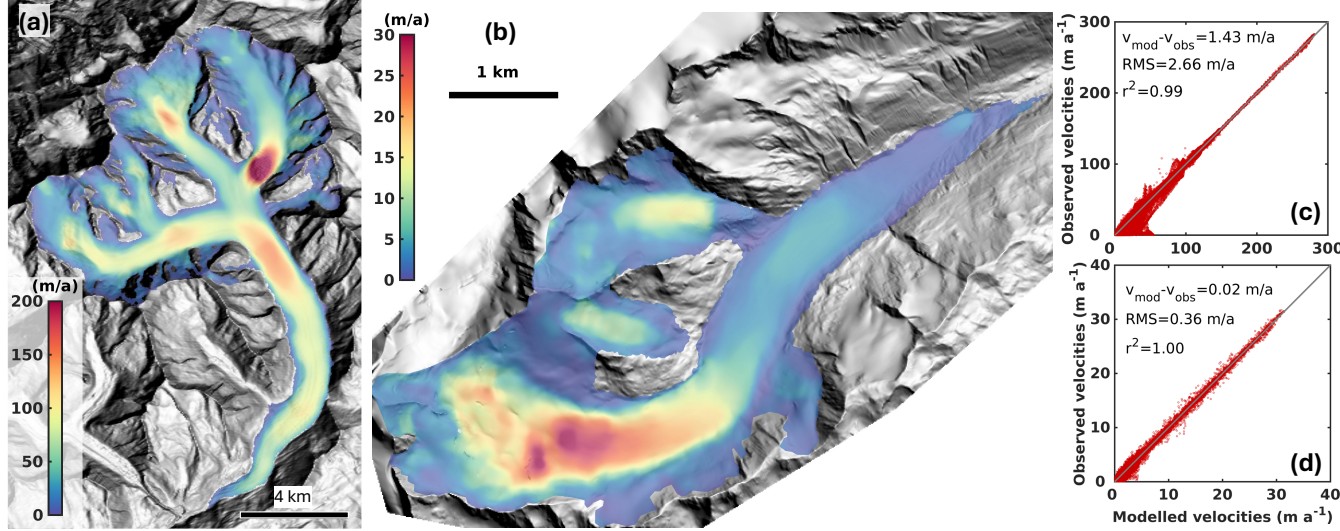

**Figure 5.** Inferred surface velocity field of GAG (a) and HEF (b) with the corresponding scatter plots in comparison with the observations (c, d) demonstrating the quality of the inversion, respectively.

velocity from this analysis at GAG is around $200\,\mathrm{m\,a^{-1}}$ at a steep ice fall of ES. HEF reaches maximum ice velocities of about $30\,\mathrm{m\,a^{-1}}$ in the accumulation area, close to the equilibrium line.

After inversion simulations, an artificial relaxation run is performed to avoid spurious noise and to allow the model to adjust
to its boundary conditions. The primary aim of the relaxation run is to level out the so called dynamical shock but we also intend to preserve the glacial state (surface elevation and ice velocities) of the simulation start date. We experimented with the





length of the relaxation period and with the SMB forcing. We found that a relaxation time of ten years and an average SMB forcing from the time period 1952 to 2000 showed that both glacier's ice volume do not deviate much from the literature values at this time. After relaxation, the ice volume of GAG and HEF is $12.7\,\mathrm{km}^3$ and $0.57\,\mathrm{km}^3$, respectively (corresponding literature

(extrapolated) values are approx $14\,\mathrm{km}^3$ and $0.57\,\mathrm{km}^3$, respectively.

## 5.2    Calibration of the surface mass balance gradient

The surface mass balance gradient is well known from observations over the last decades for both glaciers. Therefore, we tune unconstrained parameters in the albedo scheme to resemble a valid surface mass balance gradient. We sample ranges of the parameters in the albedo equation (Eq. 9) $\alpha_{\mathrm{snow}} = \{0.7, 0.75, 0.8, 0.85, 0.9\}$, $\alpha_{\mathrm{ice}} = \{0.1, 0.2, 0.3, 0.4, 0.5\}$, $\Delta = \{300, 400, 500, 600, 700\}$ m,

and $\delta = \{1500, 1750, 2000, 2250, 2500\}$ m and compare the mean modelled and observed SMB gradients over the periods 2001–2013 for HEF and 2011–2019 for GAG. These periods correspond to the times of available elevation changes (Tab. 1). The parameters finally selected to result in a satisfying root mean square error (RMS) and mean signed difference (MSD) for HEF are $\alpha_{\mathrm{snow}} = 0.8$, $\alpha_{\mathrm{ice}} = 0.1$, $\Delta = 700\,\mathrm{m}$, and $\delta = 1500\,\mathrm{m}$; for GAG $\alpha_{\mathrm{snow}} = 0.9$, $\alpha_{\mathrm{ice}} = 0.2$, $\Delta = 600\,\mathrm{m}$, and $\delta = 2000\,\mathrm{m}$. The results of the optimized elevation-dependent SMB for both glaciers are shown in Fig. 6b and d and a map view in Fig. 7.

In order to demonstrate that the tuned values are valid for another climatic period, we re-run the EBM with a mean climate for the neutral climate period 1961–1990. For both glaciers, the SMB profile shows a similar good agreement to that period (Fig. 6a and c). Note that the 1961–1990 SMB calculations make use of the 2011 and 1998 geometry for GAG and HEF, respectively. However, a general feature observed is that the modelled SMB does not represent the SMB inversion in the upper (accumulation) part of the glacier, which is likely a result of snow redistribution due to wind and avalanches; such processes

are not included in our EBM (see Discussion 7.1).

## 5.3    Validation of glacier evolution until 2023

Based on model initialization and SMB calibration, we run the model forward in time during the period where observations of elevation changes and glacier mass balance of both glaciers exist. Climate data is taken from ERA5, and the corresponding simulation will be termed the ERA5 reference simulation. For GAG, simulation starts at the initial time of 2011 and lasts until

2023; for HEF the start time is 1998 and extends until 2023. Although we put much effort in matching available observations by the initialization and calibration approach, a non-physical dynamical shock at the beginning of the simulation can still occur due to a possible imbalance of the ice mass flux and the SMB. This is often a numerical issue due to, e.g., numerical discretization and diffusion or unresolved processes in the IFM. In order to justify that the model is reasonably initialized for the projections, the modelled elevation changes and cumulative SMB in the observation period are compared to the respective

observations. For GAG, the comparison period extends from 2011 to 2019; for HEF from 2001 to 2013 (Tab. 1). This approach also ensures that independent model fields are used for model tuning (SMB gradients, ice velocity) and validation (long-term cumulative SMB, elevation change).

Modelled elevation changes are shown in Fig. 8 and the respective cumulative mass balance over the simulation periods in Fig. 7c. The modelled cumulative mass balance reveals a reasonably good agreement with the observations. Despite the




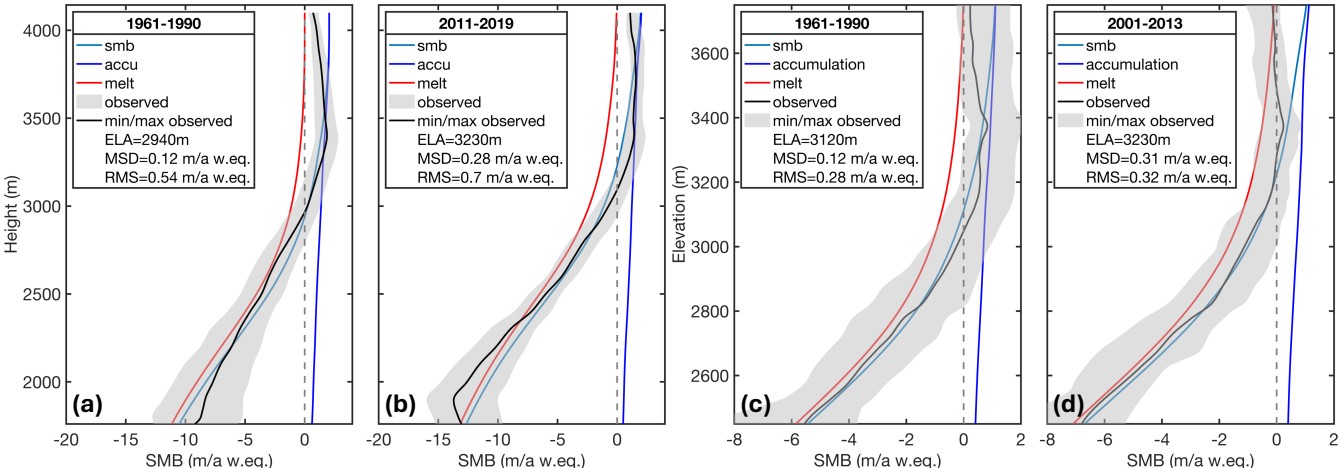

**Figure 6.** Yearly averaged SMB gradients as computed by the EBM compared to observed SMB gradients. (a, b) Computed SMB gradients for GAG for the period 1961–1990 (a) and 2011–2019 (b). (c, d) Computed SMB gradients for HEF for the period 1961–1990 (c) and 2001–2013 (d).

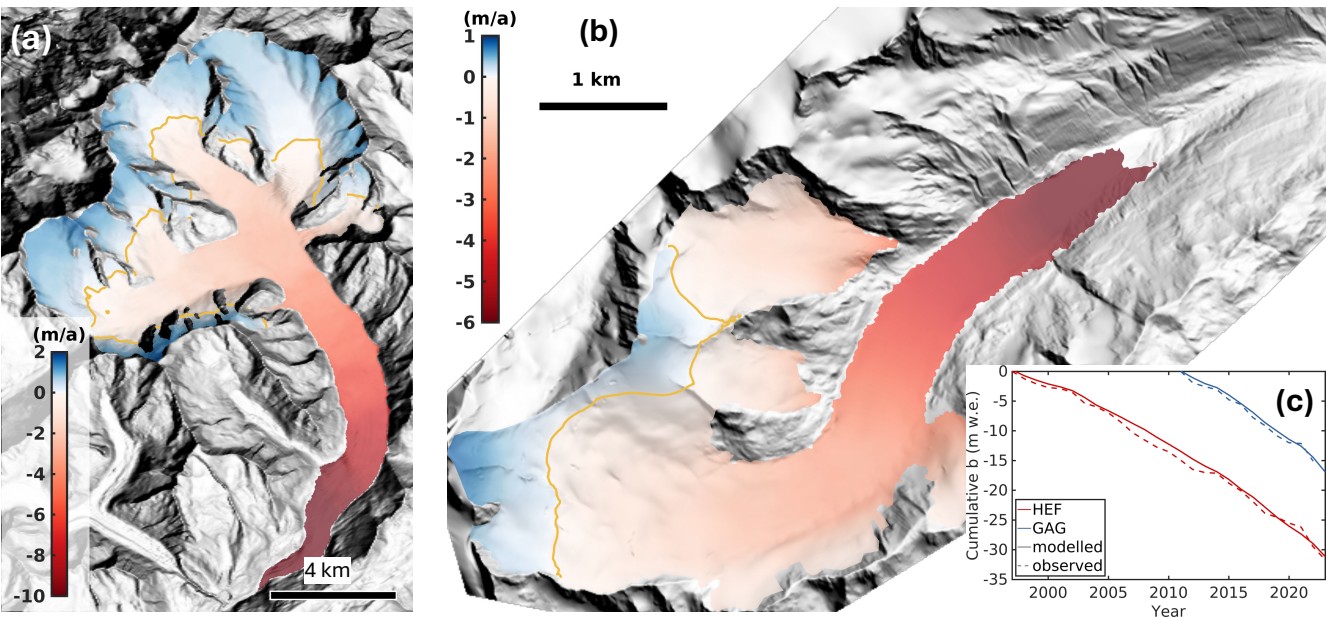

**Figure 7.** Map view of the mean SMB at GAG for the period 2011–2019 (a) and the mean SMB at HEF for the period 2001–2013 (b). The yellow line in (a) and (b) depicts the location of the modelled ELA (see Fig. 6). The inset (c) shows the temporal evolution of the cumulative SMB compared to the observations with respect to the start time of the simulations.





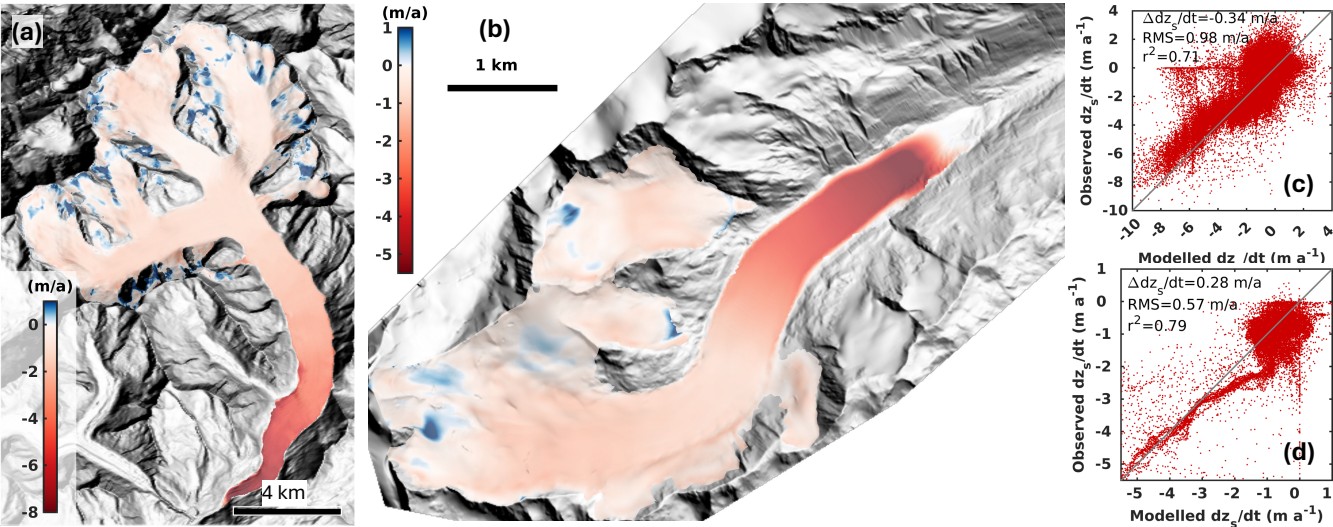

**Figure 8.** Modelled annual elevation change at GAG for the period 2011–2019 (a) and at HEF for the period 2001–2013 (b). The scatter plots show observed vs. modelled elevation change for GAG (c) and HEF (d).

minor variability of the observed cumulative mass balance, the overall decreasing trend is well reproduced by the model of both glaciers. However, the modelled spatial elevation change (Fig. 8) reveals patterns that do not match perfectly with the observed maps (Fig. 1a). At GAG, glacier thinning seems to extend too far into the upper parts (particularly at ES). At HEF, the modelled elevation change shows localized spots with glacier thickening in the upper part that are not observed (Fig. 1b). Overall, the agreement is rather satisfying, represented by reasonable RMS values (RMS=0.98 and 0.57 m a$^{-1}$ for GAG and HEF, respectively)

# 6 Glacier evolution in the 21st century

Ensembles have been simulated for GAG and HEF for the RCM/GCM models described in Sect. 3. For ensemble statistics we report a multi-model median followed by a likely range defined as the 17th to 83rd percentile range to quantify the uncertainty, unless stated differently in the text.

## 6.1 Hintereisferner

Figure 9a and b displays the simulation results for HEF's ice volume in the 21st century, including the median, the 17th-83rd percentile range and the total range for the SSP and RCP scenarios and the ERA5 reference simulation (Sect. 5.3). All climate scenarios show continued ice loss with a substantial loss of ice volume at the end of the century. The median reduction in ice volume is 99.7% with a likely range of 97 to 100% at the end of the century among all scenarios.





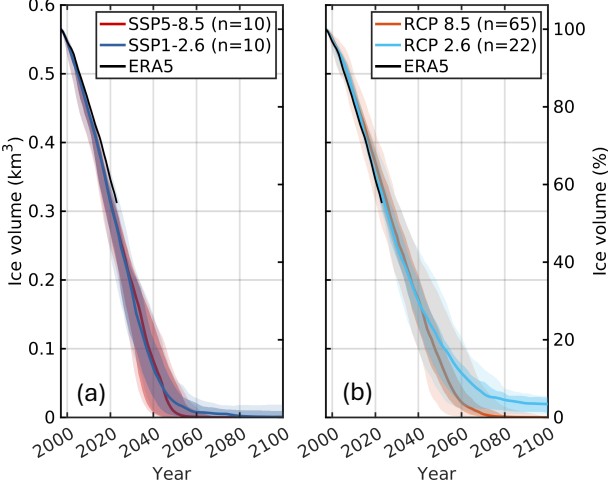

**Figure 9.** Glacier volume projections of HEF for CMIP6 scenarios SSP5-8.5 and SSP1-2.6 (a) and the CMIP5 scenarios RCP 8.5 and RCP 2.6 (b). In addition an ice volume projection based on the ERA5 reanalysis is shown to demonstrate the scenarios agreement over the past period (see Sect. 5.3). The ensemble median (thick lines), the 17th-83rd percentiles (dark shaded areas) and total range (light shaded areas) is shown.

According to the ensemble median, HEF disappears completely in the RCP 8.5 and both SSP scenarios by the end of the century; for the RCP 2.6 an ice volume of $0.019\,\mathrm{km}^2$ with a likely range of $0.028\,\mathrm{km}^2$ and $0.007\,\mathrm{km}^2$ remains. In general, the evolution of glacier volume is very similar between each scenario. At the beginning of the century, the SSP's show a somewhat stronger decline in glacier volume than in the RCP scenarios. RCP 2.6 diverge from the other scenarios around 2045 with a somewhat lower volume decrease and a tendency to stabilize around 2080.

We define a completely disappeared glacier ('gone') as the year in which less than 1% of the initial glacier volume is left (see Tab. 2); a 'mostly gone' glacier is defined as the year in which less than 10% of the initial glacier volume is left. For SSP5-8.5, HEF is gone in 2051 with a likely range of 2041 to 2064. Under the SSP1-2.6 pathway, HEF disappeared in 2068 [2047 and beyond 2100]. The RCP 8.5 project a complete disappearance in 2070 [2063 to 2078]. Although the RCP 2.6 scenarios do not project a complete disappearance, the HEF can be considered to be mostly gone, since only 3.4% [1 to 5%] of the 1997 ice volume remains.

Figure 10a and b displays snapshots of ice area and ice volume per elevation band of single models that are closest to the ensemble median for the SSP5-8.5 and RCP 2.6 scenarios, respectively. The figures do not show the full range of possible results of the entire ensemble for each scenario and are intended as a general overview. In both scenarios, a complete disintegration of the glacier tongue is expected to occur. Under RCP 2.6 scenarios, ice patches located beneath Weißkugel and Langtaufererjochferner at elevations above approximately 3200 m a.s.l. are projected to persist until the year 2100, although they could potentially disappear after this period. In the mid-century, Langtaufererjochferner and Stationsferner disconnected from the accumulation part of the glacier and become independent ice patches.




**Table 2.** Projected year for each climate scenario when GAG or HEF are gone (i.e. volume drops below 1% of the initial volume) or mostly gone (i.e. volume drops below 10% of the initial volume). Values in the brackets show the likely ranges defined as the 17th to 83rd percentiles. GAG's initial volume in 2011 is 12.7 km³; HEF's initial volume in 1997 is 0.57 km³.

| scenario | GAG gone | GAG mostly gone | HEF gone | HEF mostly gone |
|---|---|---|---|---|
| SSP5-8.5 | 2095 (2085->2100) | 2076 (2069-2089) | 2051 (2041-2064) | 2044 (2034-2050) |
| SSP1-2.6 | - | - | 2068 (2047->2100) | 2043 (2036-2048) |
| RCP 8.5 | - | 2095 (2086->2100) | 2070 (2063-2078) | 2054 (2049-2059) |
| RCP 2.6 | - | - | - | 2063 (2051-2067) |

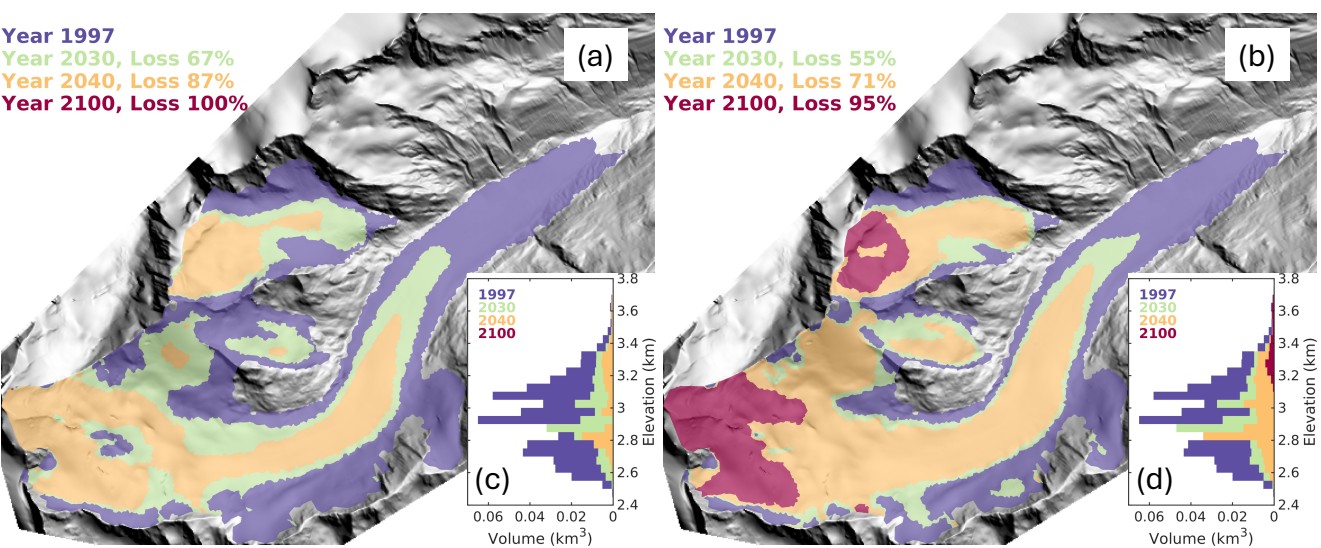

**Figure 10.** Ice area of HEF in different years for SSP5-8.5 (a) and RCP 2.6 (b) of a single model which is closest to the ensemble median volume. The percentage values presents the ice volume loss of the individual years with respect to 1997. Insets (c) and (d) show the distribution of ice volume per 50 m elevation bands in different years.

## 6.2 Great Aletsch Glacier

Figure 11 displays the simulation results for GAG's ice volume in the 21st century, including the median, the 17th-83rd percentile range and the total range for the SSP and RCP scenarios and the ERA5 reference simulation. Similarly as HEF's ice volume projection, GAG shows continued ice loss with a substantial loss of the ice volume at the end of the century.

Until ≈2040, the reduction in ice volume is almost similar for individual climate scenarios, but diverges afterward. In 2040, the median reduction in ice volume is approximately 31.4% [26.8 to 35.7%] among all scenarios. The spread of individual climate scenarios in terms of ice volume is considerable at the end of the century. In 2100, the median reduction in ice volume



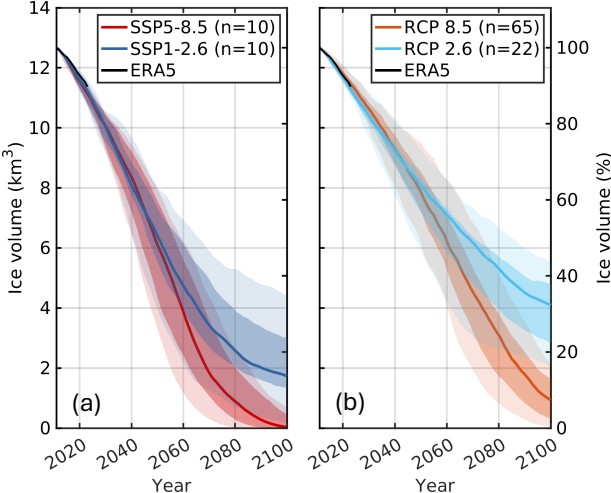

**Figure 11.** Glacier volume projections of GAG for CMIP6 scenarios SSP5-8.5 and SSP1-2.6 (a) and the CMIP5 scenarios RCP 8.5 and RCP 2.6 (b). In addition an ice volume projection based on the ERA5 reanalysis is shown to demonstrate the scenarios agreement over the past period (see Sect. 5.3). The ensemble median (thick lines), the 17th-83rd percentiles (dark shaded areas) and total range (light shaded areas) is shown.

is 88.5% [71.8 to 97.7%] among all scenarios. For RCP 2.6 and SSP1-2.6 an ice volume reduction of 67.7% [62.2 to 77.6%] and 86.4% [76.2 to 89.4%] is expected, respectively. Both low emission scenarios tend to stabilize towards the end of the century above 10% of the initial ice volume, but the decline in ice volume seems to continue after 2100. The high emission scenarios RCP 8.5 and SSP5-8.5 leading to an ice volume reduction of 92.6% [87.2 to 97.7%] and 99.7% [96.4 to 100%], respectively. The SSP5-8.5 project a complete deglaciation in 2095 [2085 and beyond 2100] (Tab. 2). Ice volumes drop below
10% in the RCP 8.5 in 2095, which is about 19 years later than in SSP5-8.5 (Tab. 2).

Figure 12a and b displays snapshots of ice area and ice volume per elevation band of single models that are closest to the ensemble median for the SSP5-8.5 and RCP 2.6 scenarios, respectively. These examples, representing the upper and lower extremes, do not encompass the full range of possible results, but are intended to provide an insight of the glacier retreat. In both scenarios, a complete disintegration of the 14 km long glacier tongue is expected to occur. In SSP5-8.5 an almost
complete GAG disappearance by 2100 is expected with only tiny ice patches in regions above ≈3400 m a.s.l. persisting until the year 2100 (Fig. 12c). For RCP 2.6 a major retreat of GAG by the end of the century is expected, but Konkordiaplatz is still connected to the three main tributaries Aletschfirn, Jungfraufirn and Ewigschneefeld. Most of the ice remains in the higher parts (Fig. 12d), however some ice is present in the lower-lying regions. The prominent feature is the peak of ice volume around the 2400 m elevation band in the year 2100. This peak is associated with the Konkordiaplatz site, which initially has a surface
elevation of ≈2800 m a.s.l. and an ice thickness exceeding 900 m. As the ice at this location persists but the surface elevation decreases, the initial peak at around 2800 m, a.s.l. decreases and shifts to a lower elevation. Although RCP 2.6 is expected





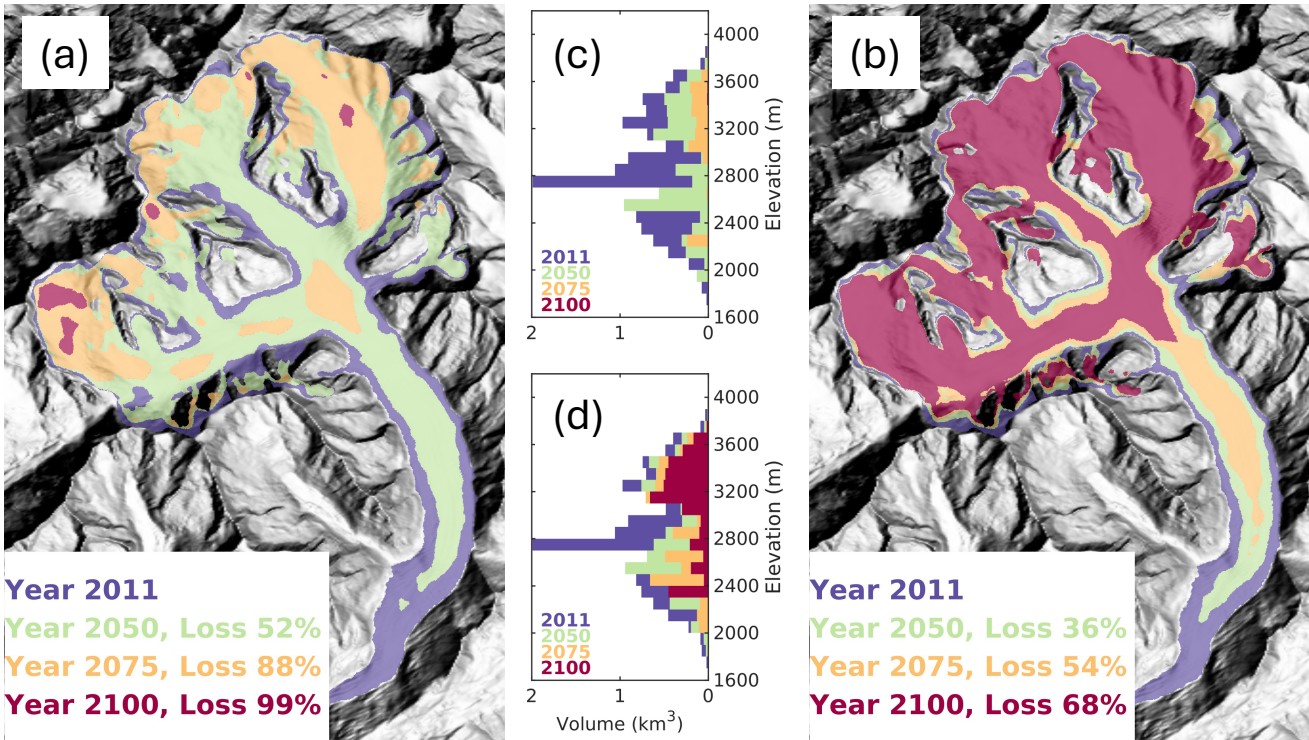

**Figure 12.** Ice area of GAG in different years for SSP5-8.5 (a) and RCP 2.6 (b) of a single model which is closest to the ensemble median volume. The percentage values presents the ice volume loss of the individual years with respect to 2011. Insets (c) and (d) show the distribution of ice volume per 100 m elevation bands in different years.

to preserve more ice at the end of the simulation period than SSP5-8.5, for example, the landscape will change significantly compared to today.

## 7 Discussion

### 7.1 Uncertainties of the model setup

Despite the fact that we use a rather complex model in terms of FS ice dynamics compared to large-scale glacier models relying mainly on SIA (e.g., Zekollari et al., 2019; Schuster et al., 2023) and a more sophisticated SMB model employing a surface energy balance model compared to simpler models (e.g. temperature index models, Schuster et al., 2023), our results are subject to several uncertainties, and sime of them we will discuss below.

Our inversion approach makes use of a remotely sensed surface velocity product as a target to constrain unknown parameters. However, the target velocity field itself is affected by uncertainties that are transferred to the IFM. On the one hand, retrieving the velocity of the ice surface of slow flowing glaciers from satellite sensors is challenging and subject to large errors. Particu-



larly in slow flowing parts (i.e. accumulation zones), the retrieved observed ice flow velocity is subject to metholodigcal errors which leads to a uncertain observed ice velocity map. The transport of ice, particularly in the accumulation area, downstream of

the glacier could be subject to this uncertainty. On the other hand, the inferred basal state resulting from the inversion approach remained unchanged during the projection period. The basal friction parameter and thus the associated with basal slipperiness could change during the projection period due to basal erosion, increased availability of meltwater at the base, and decreased overburden ice pressure. These effects are not captured by our model (basal drag is only dependent on velocity (Eq. 6)), and the basal state reflects the initial state. Furthermore, thermo-mechanics are not taken into account, which might have an impact

on the overall glacier evolution (Yan et al., 2023).

In addition to the uncertainties related to the IFM, the external forcing data are subject to several uncertainties. We are relying on future climate data from CMIP5 and CMIP6, where each GCM/RCM has its own uncertainty that produces a large spread in glacier projections. For example, in some individual GCM/RCM projections of the GAG, low-emission projections of the ice volume at 2100 overlap with simulations of high-emission scenarios (compare the full spread of SSP1-2.6 with the

full spread of SSP5-8.5 and RCP 8.5 in Fig. 12). The comparison of the overlapping projections is certainly hampered, as they stem from different GCM/RCMs but demonstrate the spread of the projections.

The study by Matiu et al. (2024) reveals the challenges encountered when using climate model data in mountainous environments. The GCM/RCM data used have a rather coarse resolution (between 12 (EURO-CORDEX) and 60 km (ISIMIP3b)) in terms of the length scale of mountain glaciers in the European Alps. Mountainous regions are characterized by complex to-

pography and the local climate is driven mainly by the interaction between large-scale atmospheric flows and local topography. Although the employed downscaling and bias correction ensure a reasonable level of the climate variables, the overall climate output has systematic biases. Small scale variability of atmospheric dynamics, originating from complex air flow modifications by mountainous terrain is not resolved in the rather large-scale resolution GCM data. This might introduce unquantifiable inaccuracies and biases in the climate data and has to be borne in mind, when interpreting the results. A distinctly higher spatial

resolution of the climate data would be necessary to adequately represent the relevant physical processes. For instance, for better resolving the turbulent fluxes (latent and sensible), requires a more sophisticated windspeed model accounting for small scale variability. However, this is beyond the scope of this study and computationally way too expensive to be accounted for in the scope of this study given the 107 different climate projections used in the modelling process.

Uncertainties in the calculated SMB are not solely due to climate data; they also arise from potential overestimations in

snow accumulation at higher elevations, as we lack a model for snow redistribution by wind transport and avalanches. We experimentally applied a snow redistribution model based on surface slope and curvature following Huss et al. (2008), but it resulted in only minor improvements relative to the validated SMB gradients without incorporating a wind/avalanche redistribution model (see Fig. 6). Moreover, when incorporating snow redistribution based on surface slope and curvature, our model experiences instabilities in the evolution of the ice surface (e.g., holes in the surface). Although the parametrizations of snow

redistributions appear promising, further refinement for application in an IFM is required.





## 7.2 Comparison with previous results

The future evolution of glaciers in the European Alps has been projected with models of various complexity and based on diverse climate projections. Our results of a substantial volume loss until the end of the 21st century under CMIP5 and CMIP6 future climate scenarios are in line with previous estimates based on large-scale (Zekollari et al., 2019; Schuster et al., 2023)
and detailed glacier modelling studies (Jouvet and Huss, 2019) (Fig. 13). However, the timing of deglaciation and the remaining ice volume at the end of the 21st century vary considerably.

   The study by Jouvet and Huss (JH19, 2019) is comparable to our research with respect to the model configuration. They utilize an individual full-Stokes model setup for GAG, with model tuning and validation based on observations (ice volume and length of glacier tongue, in-situ point measurements of ice surface velocities, and surface mass balance). However, JH19
predicts a larger volume loss by the end of the 21st century relative to our findings. For RCP 8.5, they predict a 98.5% reduction in ice volume, while our model predicts a reduction of 92.6%. Specifically, under RCP 2.6 they predict a 59.5% reduction in ice volume, contrasting our model prediction of 67.7%. The discrepancies may originate from the different models employed for calculating surface glacier melt. Our study utilizes a physically-based energy balance model, while JH19 employs a temperature index model (Hock, 1999; Huss et al., 2008). Although both models tend to overestimate the mass budget and have a limited
ability to reproduce ice volume on decadal time scales, the mass budget computed by an energy balance model, despite its physical character, is probably too high (compare HTI and EB schemes in Gabbi et al., 2014). This behaviour might explain the lower loss of ice volume compared to JH19 for RCP 8.5; and for RCP 2.6 until ≈2075.

   The volume projections of the large-scale models are generally situated above (GloGEMflow) or, close (OGGM) to our projections. Under RCP 2.6, GloGEMflow predicts a volume reduction of 57.6% for GAG, aligning somewhat more closely
with our forecast than the JH19 study; likewise, for RCP 8.5, GloGEMflow projects a 89.4% volume decline, which is closer to our estimate than the JH19 study. OGGM displays an irregular pattern when compared to our projections. Under the SSP5-8.5 scenario, discrepancies are pronounced in the mid-century period but diminish significantly after 2075. Conversely, for the SSP1-2.6 scenario, the differences remain minor before 2075, although OGGM projects glacier growth in the subsequent years. In the case of HEF, the OGGM model aligns well with our projections over the entire projection period for the SSP5-8.5 and
SSP1-2.6 scenarios. In contrast, GloGEMflow exhibits significant discrepancies. Although GloGEMflow predicts an almost complete disappearance of GAG by the end of the century, the projected volume loss is significantly delayed (≈50 years) compared to OGGM and our study. Under the RCP 2.6 scenario, a substantial proportion of ice remains (32.5%) compared to OGGM and our study.

   Disentangling the reasons for the differences is challenging, particularly when comparing model projection outcomes based
on large-scale approaches to individual glacier applications, since model techniques are designed in a different manner. In addition to inherent differences arising from model concepts and design, other contributing factors include the climate forcing datasets used, the methodologies of model calibration and initialization, as well as the choice of the mass balance schemes. Please note that the comparison of the selected studies aims to demonstrate the possible spread of the model outcomes, but is not fully exhaustive, as we did not include additional studies that present projections of GAG and/or HEF (e.g., Hanzer et al.,




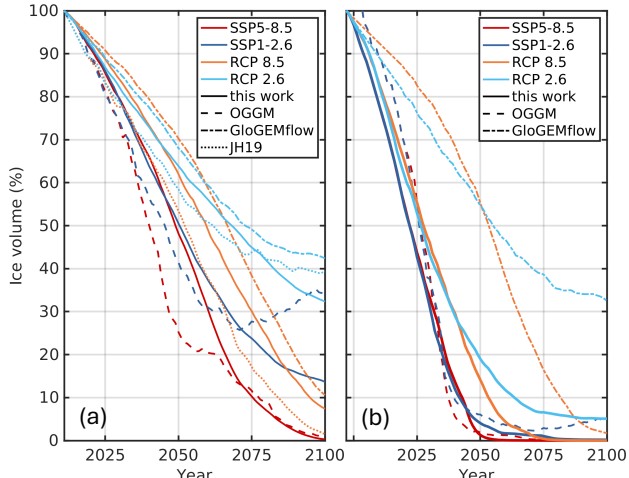

**Figure 13.** Comparison of ice volume evolutions from previous studies for GAG (a) and HEF (b). Results of the large-scale models OGGM and GloGEMflow are from Schuster et al. (2023) and Zekollari et al. (2019), respectively. JH19 refers to the detailed full-Stokes study of GAG by Jouvet and Huss (2019). For each study, the ensemble medians are shown with the exception of GloGEMflow, for which the ensemble mean is presented. Note, that the projection start date of GloGEMflow is 2017 and we use a linear extrapolation to 1997.

2018; Zekollari et al., 2024; Hartl et al., 2025). However, our selective comparison demonstrates the potential spread of glacier model projections with regard to model sophistication, initialisation and forcing.

### 7.3   East-west comparison

The most striking feature of the volume projections is that even in low emission scenarios, the HEF (almost) disappeared. In addition, the low- and high-emission scenarios at HEF are rather identical in terms of ice volume loss. The losses agree pretty

well with the expectation of (almost) full deglaciation of HEF in the middle of the 21st century. This is an intriguing result, since limiting global warming to 1.5 to 2°C above pre-industrial levels negotiated within the Paris Agreement would cause a (almost) complete disappearance of HEF, regardless whether the sustainable or the highest emission pathway is considered.

GAG has a longer lifetime than HEF in the different future climate scenarios. This is related to its larger ice thicknesses that persist even at low altitude (Fig. 12c, d), and its larger elevation range. While GAG is projected to disappear under high-

emission scenarios, low-emission scenarios suggest the need for stringent mitigation measures to prevent further reduction of GAG's glacier volume.

Figure 14 shows the distribution of ice area per elevation bands of GAG and HEF and the evolution of the ensemble-median ELA of each scenario. Notable is the increase of the ELA at HEF for each scenario reaching, or even exceeding, the ice coverage fraction at the highest elevation; merely the ELA under RCP 2.6 forcing reaches a height that is below a certain

fraction of the present-day ice coverage. At GAG, the increase of the ELA for the low-emission scenarios is muted compared to the high-emission scenarios. In particular, a large fraction of ice area above >3500 m a.s.l. would remain.



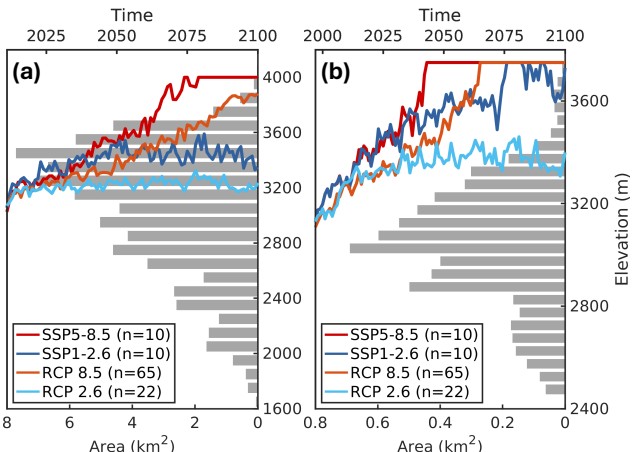

**Figure 14.** Distribution of ice area per 100 m and 50 m elevation bands for and GAG (a) and HEF (b), respectively, for the year 2023 ( with respect to bottom x-axis). Coloured lines show the evolution of the ELA of the four scenarios SSP5-8.5, SSP1-2.6, RCP 8.5 and RCP 2.6. (with respect to top x-axis).

The examined sample is certainly not representative for all glaciers in the eastern alps, but simply upscaling our projection results of HEF and GAG to all glaciers in the European Alps within an elevation range of ≈2000 to 3500 m a.s.l. would cause an almost complete wastage of these glaciers in the middle of the 21st century. This estimate is supported by Hartl et al. (2025),
where all warming-level scenarios lead to a nearly complete deglaciation in the Ötztal and Stubai Alps (western Austria) before 2100. In the western alps, glaciers covering an elevation range above ≈ 3500 m a.s.l. have a longer lifetime or even the potential to survive until 2100. However, in the worst scenarios, most glaciers in the western alps also disappear at the end of the 21st century.

The simple extrapolation of two exemplary glaciers to all glaciers in the European Alps is certainly hampered by several
factors. The regional setting of each glacier probably influences the glaciers' response to increasing temperatures. For example, GAG receives very high amounts of precipitation in the accumulation area due to moist air masses from the north. However, these air masses are blocked by the main alpine crest and create very dry conditions in the inner-Alpine valleys (e.g., Rhone valley), which leads to considerably smaller amounts of precipitation at GAG's glacier tongue.

HEF, as well as other low-lying glaciers found in the eastern European Alps, are particularly vulnerable to increasing tem-
peratures because they are situated at lower elevations than many of their western counterparts, which reside at higher altitudes. In addition, HEF experiences a somewhat more intense warming than GAG. For RCP 8.5 a stronger warming of 0.7±0.9°C is expected at HEF compared to GAG; for RCP 2.6, SSP5-8.5 and SSP1-2.6 0.3±0.6, 0.2±0.2, and 0.3±0.2°C, respectively. Note that the estimated values correspond to an elevation of ≈2200 m a.s.l. beneath GAG and HEF, but temperature changes are subject to local conditions and elevation (Matiu et al., 2024). In our analysis and model configuration, HEF is suffering
from a more pronounced warming than GAG.





## 8 Conclusions

We applied a three-dimensional full-Stokes ice-flow model to project the evolution of the Great Aletsch Glacier and Hintere-isferner, considered as representative valley glaciers for the European Alps. Our modelling framework diverges significantly from large-scale modelling efforts, yet remains consistent with their overall findings. Under all climate scenarios, both glaciers
are projected to lose a substantial volume of ice or complete deglaciation until the end of this century. Particularly, the retreat of Hintereisferner appears unavoidable even under sustained climate scenarios (inline with the political target negotiated in the Paris Agreement) compared to high-emission scenarios. The predicted 21st century retreat of the Great Aletsch Glacier is dramatic, although under sustained climate scenarios a portion of the Great Aletsch Glacier is likely to persist by 2100, but the decline in ice volume seems to continue after the projection end date of 2100. Our findings suggest that glaciers in the
eastern European Alps are likely to diminish by the mid-21st century, and only larger glaciers with higher elevation ranges in the western European Alps will remain until the end of the century. The near-total disappearance of glaciers in the European Alps is expected to affect water availability, pose hazards in a deglaciating environment, and impact tourism and the economy.

A comparison of our ice volume projections with previous studies reveals a large spread. The sources of projection differences are difficult to disentangle as there are several factors influencing the projections. It remains difficult to determine
whether a more physically-based methodology (e.g., full-Stokes versus simpler models, surface energy balance models versus temperature index models) is essential for narrowing uncertainties in projections, as factors such as the initialization approach, the calibration strategy of uncertain parameters, and the handling of climate data might exert a comparable or even larger influence on the projections. Consequently, the development of standardized tests for model intercomparison between individual glacier modelling to large-scale modelling would be of considerable value to foster model improvements. Such a comparison
is beyond the primary focus of GlacierMIP (2025), a framework for a coordinated intercomparison of global-scale glacier models, but would even complement and enhance this effort.

*Code and data availability.* Simulation results on the native grids and scalar values described in this paper will be made publicly available with a digital object identifier https://doi.org/10.5281/zenodo.15234334 (Rückamp et al., 2025, dataset not yet publicly available). The remotely-sensed surface velocity field of Hintereisferner is available through https://doi.org/10.5281/zenodo.15269581 (Gutjahr and Rück-
amp, 2025, dataset not yet publicly available). The EURO-CORDEX and ISIMIP3b climate data can be acquired from the respective data servers, for example https://esgf-data.dkrz.de/projects/esgf-dkrz/ (last access: July 1, 2025) and https://data.isimip.org/search/tree/ISIMIP3b/SecondaryInputData/climate/atmosphere/mri-esm2-0/ (last access: July 1, 2025), respectively. The code for the bias adjustment (Chadwick et al., 2023) of the climate data is open source and available at https://github.com/saedoquililongo/climQMBC/tree/main (last access: July 1, 2025). The ISSM ice flow model is open source and is freely available at https://issm.jpl.nasa.gov/ (last access: July 1, 2025), (Larour et al.,
2012). Here ISSM version 4.23 is used. The surface energy balance model (Evatt et al., 2015; Mayer and Licciulli, 2021) is available through https://github.com/carlolic/DebrisExp. (last access: July 1, 2025)



## Appendix A

### A1    Calculation of specific humidity

For running the EBM the variable near-surface specific humidity needs to be known. However, this variable is not provided by
the ERA5 climate dataset that is used as forcing, and needs to be calculated instead. On the basis of dew point temperature ($T_d$,
in $°C$) and surface pressure ($p$, in mb) specific humidity ($q$, in kg/kg) can be calculated as:

$$q = 0.622 \frac{P}{p - 0.378P}. \tag{A1}$$

Here, the vapour pressure ($P$ in mb) can be calculated following the Magnus equation (Alduchov and Eskridge, 1996):

$$P = 6.112 \exp\left(\frac{17.62 T_d}{T_d + 243.12}\right). \tag{A2}$$

## Appendix B

### B1    DInSAR surface velocity of Hintereisferner

Differential Synthetic Aperture Radar Interferometry (DInSAR) is a highly effective technique for measuring glacier move-
ments by analyzing phase differences between SAR images acquired at different times. Using a known digital elevation model
(DEM) to account for topographic effects, DInSAR can isolate motion-related phase changes, allowing accurate measurement
of glacier surface velocity and flow patterns. This approach has been widely applied in glaciology, for instance, to monitor
glacier dynamics in Greenland and Antarctica, offering insight into spatial and temporal variations in flow velocity (Rignot
et al., 2011). The use of a DEM simplifies the processing chain and improves the reliability of deformation measurements
(Nela et al., 2019), making DInSAR a valuable tool for studying glacier responses to climate change (Joughin et al., 2010).

The potential of this technique to map the motion field of small alpine ice and rock glaciers was initially explored two
decades ago (Nagler et al., 2002; Gutjahr et al., 2004), utilizing ERS-1/2 data and focussing on the Hintereisferner test site,
as also addressed in the current study. Hence, it is evident that both the data and the general workflow were established
during that period. However, significant advancements have occurred over the past twenty years in both understanding of data
and the methodologies employed for data processing. Regarding data understanding, notable progress includes the consistent
reprocessing of ERS-1/2 orbits (Otten and Visser, 2019) and upgrading the DEM of the SRTM-C band DEM to version 4.1
(Jarvis et al., 2008). In terms of processing methodologies, advances such as the development of phase filters (Li et al., 2008)
and phase unwrapping techniques, exemplified by the latest release of the Statistical-cost, Network-flow Algorithm for Phase
Unwrapping (SNAPHU) by Chen and Zebker (2002) in February 2024, have significantly enhanced capabilities.

Table B1 summarizes the ERS-1/2 interferograms utilized in this study to derive the motion field of the Hintereisferner test
site using the two-pass differential SAR interferometry approach (Kenyi and Kaufmann, 2003). To enhance coregistration and
optimize the imaging geometry, all other available ERS-1/2 SLCs from the year 1996 were incorporated into the preparatory
steps.



**Table B1.** ERS-1/2 Interferograms used in this study

| Id | Dates | B perp (m) |
|----|-------|-----------|
| \multicolumn Track 444, Frame 927, Ascending orbit, Acquisition time ~21:30 UTC: | | |
| 1 | 6/7 December 1995 | 207 |
| 2 | 10/11 January 1996 | 162 |
| 3 | 14/15 February 1996 | 138 |
| 4 | 20/21 March 1996 | 297 |
| Track 437, Frame 2655, Descending orbit, Acquisition time ~10:07 UTC: | | |
| 5 | 6/7 December 1995 | -97 |
| 6 | 10/11 January 1996 | -77 |
| 7 | 14/15 February 1996 | 51 |
| 8 | 20/21 March 1996 | 22 |

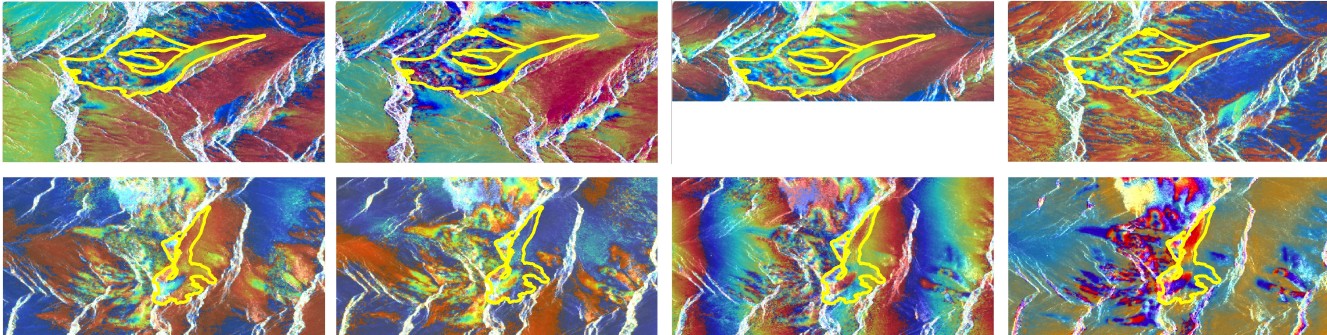

**Figure B1.** Colorized differential interferograms. Yellow line shows the 1998 glacier outline. Top row: Ascending orbit. Bottom row: Descending orbit. From left to right: 6/7 December 1995, 10/11 January 1996, 14/15 February 1996 and 20/21 March 1996.

The ascending and descending stacks were processed independently up to the conversion of differential interferograms into line-of-sight (LOS) displacements. Finally, the equations described by Yin and Busch (2018) were applied to transform the orbit-dependent LOS displacements into the east-west and up-down components of the underlying three-dimensional deforma-
tion pattern. All differential interferograms calculated are shown in Fig. B1.

*Author contributions.*   MR and CM designed the experiments. MR implemented the EBM (Sect. 4.3) and the streamline upwind Petrov–Galerkin (SUPG) stabilization for the free surface equation (Eq. 8). MR run the experiments with subsequent postprocessing. MM contributed in sur-



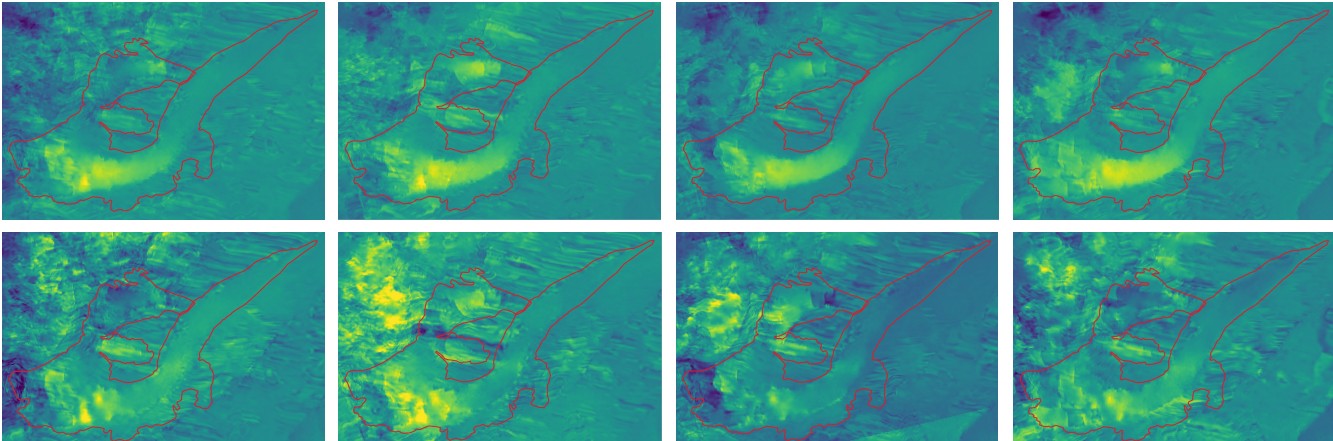

**Figure B2.** Interferometric glacier motion. Red line shows the 1998 glacier outline. Top row: East-west component [stretch -10 cm d$^{-1}$ to 10 cm d$^{-1}$]. Bottom row: Up-down component [stretch -3 cm d$^{-1}$ to 3 cm d$^{-1}$]. From left to right: 6/7 December 1995, 10/11 January 1996, 14/15 February 1996 and 20/21 March 1996

face mass balance modelling. GC implemented Nitsche's method into ISSM. KG and PP contributed with the HEF observed surface velocity map. All authors contributed to discuss model results and in writing the manuscript.

*Competing interests.* GC is a member of the editorial board of The Cryosphere. All other authors declare that they have no conflict of interest.



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
