# Peer review of "Future Retreat of Great Aletsch Glacier and Hintereisferner – application of a full-Stokes model to two valley glaciers in the European Alps"

_EGUsphere, 2025_

## Referee Comment (RC1)

**Review of Rückamp et al. (2025) 'Future Retreat of Great Aletsch Glacier and Hintereisferner – application of a full-Stokes model to two valley glaciers in the European Alps'**

**Summary**
This paper presents a full-Stokes modelling study of Hintereisferner and Great Aletsch Glacier over the course of the 21$^{st}$ century. Hintereisferner is initialised in 1997, Aletsch in 2011. The authors calibrate their model against observations up to the point these are available and additionally perform inversions for unknown parameters (basal sliding coefficient, ice hardness). The paper finds that Hintereisferner, even under a low-warming scenario compatible with the Paris Agreement, will likely disappear by mid-century, while Aletsch will likely survive in a dramatically reduced state. Under a high warming scenario, however, both glaciers will be (almost) entirely gone by 2100.

I think this paper is well-executed: the method is solid and the results convincing and thoroughly discussed. I have a few minor comments detailed below, the most substantial of which bears on re-writing some of the discussion to make it more impactful, but this is purely a question of emphasis and presentation rather than any sort of fundamental flaw in the paper. Therefore, I recommend that the paper be returned to the authors for minor revisions.

Page and line numbers refer to those in the clean version of the revised manuscript.

**Major Comments**
- Discussion: see the detail in the individual minor comments below, but I think the discussion could be reworked a bit to make it snappier and so that it focuses on the more interesting points the authors raise. At the moment, a lot of it feels either very obvious or doesn't really lead anywhere.

**Minor Comments**
- p.2, l. 28-43: To bring this completely up to date, might it be worth including IGM in the list here? It's not been applied globally yet, but there are some regional studies and it seems to be aiming for the same sort of applications as this paragraph is concerned with
- p.3, l.81: It may well be correct that this is the first time the specific form of data assimilation used here has been applied to mountain glaciers, but it's certainly not true in a more general sense, which is how this sentence reads. I might suggest toning this down slightly.
- p.6, l.125: Delete 'a'
- p.6, l.132: 'regional'
- p.6, l.139: The plural of 'RCM' is 'RCMs'. Same with 'GCMs' on the next line. No need to put an apostrophe in. Make sure there aren't other instances – I won't bother flagging them all up.
- p.7, l.164: 'As expected'
- Section 5.1: I think there are two points here. First, what happens if the inversions are done the other way round, i.e. an initial guess for beta is provided to do an inversion for B and then beta is inverted based on that B profile? Ideally, the model would end up in the same place, but it might not, and I think it would be worth checking this. Otherwise, it all feels a bit circular and arbitrary. Second, I'm not sure I agree about the L-curves. The one in panel d does have a clear corner and therefore optimal value, but the other three, especially a and c, really don't. Now, I don't think it would make much difference whether the next value up or down were picked on any of the curves, but I might walk back the statement about lambda being easily pickable at l.302-3.
- p.12, l.279: Is 'thermochemical' a typo for 'thermomechanical'?
- p.14, l.315: There's a missing closing bracket for the one opened before 'corresponding' on the previous line
- Figure 9 caption: Here, an apostrophe is needed for 'scenarios''
- p.17, l.362: …and here one isn't – it's just 'SSPs' for the plural
- p.17, l.363: 'diverges'
- p.17, l.368: 'projects'
- Figure 10 caption: 'present'
- Figure 11 caption: same as for Figure 9

- p.19, l.388: 'lead'
- p.19, l.389: 'projects'
- Figure 12 caption: same as for Figure 10
- p.20, l.409: 'some'
- p.21, l.416: There's a word missing – 'The basal friction parameter and thus the...what...associated…'
- p.21, l.417: OK, yes, fair enough, but are basal erosion rates really that high that one would expect them to lead to any sort of noticeable impact on the glacier's behaviour in the next 80 years? This feels a bit of a reach rather than a worthwhile point to include in the discussion. I'd replace it with changes in the glacier's stress regime, which would alter the distribution of sticky and slippery spots on the bed (and therefore beta) in a noticeable manner on the timescales relevant to this study.
- Section 7.2: I get that the authors are not attempting an exhaustive comparison with all previous studies, but this entire section pretty much just says 'different models with different set-ups give different results and we're not going to dig any further', which is maybe not the best use of a page-and-a-half of the discussion. That could be asserted in one line and no one would bat an eyelid. I feel it would be more valuable if the authors focused on the comparison with the Jouvet and Huss (2019) study, which should be most comparable, and tease out why they think the results are different, which then tells us something useful about the impact of different parts of the model setup. I would probably then just reduce the comparison with OGGM and GloGEM to a couple of lines, noting the difference, because there I agree with the authors that the models are so fundamentally different that it's very difficult to establish exactly where the differences in the results are coming from.
- Section 7.3: Similarly, this feels like quite a lot of effort to go to to point out that Hintereisferner is going to do worse than Aletsch because it's smaller, at a lower elevation, and in a drier part of the Alps. It's again something that would be an uncontroversial one-line assertion, particularly if the authors reference it to Figure 14, which summarises all the relevant information. The other material in this section about what this tells us about Alpine glaciers more generally is, I think, the interesting bit, so I'd suggest cutting down the first few paragraphs at the bottom of p. 23 and the last paragraph on p. 24 and condensing it into one paragraph that points out the obvious reasons for the different behaviour of the two glaciers.

---

## Referee Comment (RC2)

**Summary**

This manuscript of Rückamp et al. (2025) presents projections of the future evolution of two Alpine glaciers (Great Aletsch Glacier and Hintereisferner) using a full-Stokes ice flow model coupled with a surface mass balance (SMB) scheme driven by the surface energy balance method. The model is initialized using observed velocity fields, and future projections are forced with bias-corrected climate data under different scenario's. Their projections suggest that Hintereisferner will vanish mid-century even under the low-emission scenario, while Aletsch will significantly shrink, possibly with a near-complete disappearance under high-emission scenarios by 2100 AD. The paper addresses an important scientific and societally relevant topic, as glacier-scale projections using full-Stokes models remain relatively rare, and the work provides valuable insight into the dynamic response of two iconic glaciers. The manuscript is generally well written (also some typo's remain and some things can be formulated better) and the results are of interest to both glaciological and the broader climate-impact communities.

The manuscript is scientifically valuable and should ultimately be suitable for publication in The Cryosphere. However, several scientific and some small linguistic issues here and there need to be addressed before publication. I recommend major revisions to (i) strengthen the treatment of the model forcing and bias-corrected climate data, and (ii) elaborate a bit more on model uncertainty related to the future projections. I suggest publication once the comments below have been addressed.

**Major comments**

1) Model forcing and bias-corrected climate data: The model forcing is described in Section 3, but I would like to have a more detailed explanation of some things that have not become very clear to me from the text.

   - Applied bias correction: In L124, you mention a "simple correction" has been applied to ERA5-Land data using observational temperature and precipitation data, and in L132 you say that you "shifted and scaled" the temperature and precipitation for the 1961-2023 period. Can you be more specific about what you did? Did you just scale the data so the mean of the overlapping period matches? Did you also adjust the variability (standard deviation)? Did you scale temperature and precipitation differently (i.e. additive for temperature, multiplicative for precipitation)? Did you apply the bias correction with reference data using a daily temporal resolution? Moreover, you mention other data as well for the SEB model (wind speed, radiation,

humidity) but some explanation for a bias correction procedure for those are not mentioned for the "observational" (1961-2023) period. Did you scale these too for 1961-2023? I think the whole procedure can be further elaborated step by step into more detail to make it clearer for the reader. Moreover, applying a bias correction to the future climate data is a necessary step but it may result also in a change in the long-term trend of the data contained in the original output. Have you checked the preservation of the trends after bias correction?

- Compatibility of ERA5-Land for glacier-specific meteorology: You explained well that you use ERA5-Land for the bias correction of the meteorological forcing. Have you checked how well ERA5-Land and your observational data (from the meteo stations) agree during the overlapping periods, using some statistics? It is, for example, well-known that reanalysis data like ERA5-Land may not fully represent small-scale processes like glacier winds (which should be however the dominant wind regime over the glacier), but rather synoptic-scale wind patterns. Is this also applicable to your data? You briefly mention this in the text (in Section 7.1) in a qualitative way but I think that this warrants some further investigation or at least a statistical quantification of the level of agreement during the overlapping period.

- Model selection for future projections: It would be beneficial to have a list of used GCM/RCMs (for example as an Appendix) that are used for the future projections. I say this because a subset of the CMIP models are found to be 'too hot' and may lead to an overestimation of glacier mass loss. Hausfather et al. (2022) has suggested that models with a TCR that lies outside the 'likely' range of 1.4-2.2 °C should be left out to avoid overestimation. You may look into this or at least mention it in the text.

2) Uncertainty of future model output: Some more incorporation and/or discussion of model output uncertainty by the variation of some key variables (for example related to the SEB/SMB) is warranted in my opinion. Also, the evolution of supraglacial debris is not included in the model, which I can agree on given the current minor debris extent on both glaciers, but its potential future effects should be more clearly acknowledged in the text.

- Model output uncertainty: In my opinion the discussion of the model uncertainty can be improved. I understand that most quantified model uncertainty in the Figs. 9 and 11 and in the uncertainty intervals comes from the different GCMs/RCMs and future climate scenario's, but what about the uncertainty of the SMB profile that results from these climate data? This is only very briefly mentioned in section 7.1 but still important because the future evolution of the glaciers is in the end SMB-driven. For example, from my understanding, the gradients used in this study (L169-172) are taken over from other studies and not tested for validity for this study. How do they

affect the SMB profile? In other words, how robust is the SMB model to internal parameter choices? This can be briefly discussed. I understand that it is computationally expensive to also include the sensitivity of various parameters to the SMB model in all calculations, but I think the manuscript would benefit from at least a sensitivity analysis of the SMB model (for example, with a Monte Carlo approach or a figure or table summarizing the sensitivity to major parameter uncertainties).

- Mentioning of supraglacial debris cover effects: Currently, debris cover effects are minor but certainly already present. A clear inversion of the SMB gradient in the lower parts of Aletsch is seen in your figure 6b, and a clear dampening of the surface lowering on the southeastern part of the Hintereisferner is apparent from your figure 1b. This coincides with an area of debris cover on the snout, which is clearly seen on satellite imagery. The effects of debris are indeed highly glacier-specific (depending on debris thickness/area, debris properties, and climatic conditions) and future trends are difficult to establish. However, given that the effects of debris are already present to some degree and generally expected to increase in the future (e.g. due to enhanced melt-out, increased bedrock exposure and slope instability, decreased flow velocities/debris discharge off-glacier) and given that it can potentially have an impact in the future (which was already modelled on the specific Aletsch glacier by Jouvet et al. (2011)), I think it warrants at least some further explanation of why you didn't include it and/or discussion on its potential effects in the paper.

**Minor comments:**

L20: The new papers of Dussaillant et al. (2025) and GlaMBIE (2025) may be a good reference here.

L56: you can elaborate maybe a bit more on the specific advantages of using full-Stokes over the SIA or higher-order approximation (e.g. what type of stresses are included). Does it really make that much of a difference and if yes, in which areas do you expect the most significant improvements? From my experience the SIA and HO models usually work really well for glaciers, so what are the main advantages of using the full Stokes when compared to other approximations? As you mention in the conclusion, a detailed quantitative comparison would be out of scope but I do think it can be briefly discussed.

L61: remove second )

L71: you can maybe mention here the dynamic calibration procedure that is usually performed (e.g. artificially adjusting the historic mass balance after a steady-state spin-up so that the observed and modelled lengths over the historical period agree until presentday). Was this dynamical calibration procedure not feasible for your model? Were you able to reconstruct and compare historic front variations?

L82: I think you can already briefly mention here why these two glaciers were chosen specifically and further elaborate on it in the section 2. They are for example not WGMS reference glaciers and it is not the first time that they are modelled, why are they specifically important and are they representative for the Alps in general?

L125: remove 'a'

L132: adjust 'regionla'

L170: The shortwave radiation is decreasing with elevation. Is this the incoming only or the total shortwave radiation (minus outgoing)? Usually, atmospheric transmissivity increases with elevation, enhancing the incoming shortwave radiation at higher elevations.

L220: can you show the equation used for the cosine interpolation? Or at least a graphical representation of it in the Appendix maybe.

L239: can you give an indication of the computational cost? How long does it take to run the model?

L234: I understand the albedo is used as a tuning mechanism, but do you have anything to compare their values to for checking its credibility? Data from an on-glacier AWS?

L280: do you have evidence that both glaciers are indeed isothermal such that a thermomechanical coupling is not necessary? Also adjust 'thermochemical'.

L306: ice velocities after initialization agree really well, nice! What about the ice thickness? Can you provide an RMSE for those as well?

L315: Add additional bracket )

Figure 6 caption: can you indicate here again where the SMB observations come from where you tune against? You show accumulation and melt separately, but do you have data to validate whether the distinction between these two is correct? Or do you only have observed data of the final SMB?

L350: missing point at end of sentence

L360: The term "disappearance" can be ambiguous (e.g. complete ice-free, negligible residual ice, disconnected patches, below a threshold volume or thickness?). You should clearly define here how you define glacier "disappearance".

L363: diverge -> diverges

L368: project -> projects

L371: Figure 10a and b displays -> Figures 10a and b display

Figure 10 caption: presents -> represent

Figure 9b and 10b: can you include the reference time period to which the volume loss (%) is compared in the y axis label? You may also want to include it in the captions of Fig. 10 and 12.

L388: leading -> lead

L389: project -> projects

L401: 2800 m, a.s.l. -> remove comma

L409: sime -> some

L412: the accuracy of ice velocity retrieval from satellites depends on the acquisition method. SAR interferomerty is usually very accurate for slower moving ice.

Section 7.3: I don't really like the title 'east-west comparison'. To me this section just reads like an attempt to generalize the behavior of the two glaciers over a certain region. To explain the different behavior of the two I think you can also elaborate more on (1) the climatic setting that may differ (can you give some quantitative climate data for both glacier environments to corroborate this?), but (2) also the climate sensitivity of the glaciers related to their geometry. Hans Oerlemans did a lot of research into this and you can maybe compare the glacier characteristics of both glaciers to explain their different future behavior/sensitivities (mass balance gradient, overall slope, glacier size, hypsometry (e.g. large accumulation area vs. narrow snout), etc.). This also adds to the difficulty of generalizing glacier behavior for a certain region.

**References:**

Dussaillant, I., Hugonnet, R., Huss, M., Berthier, E., Bannwart, J., Paul, F., and Zemp, M. (2025). Annual mass change of the world's glaciers from 1976 to 2024 by temporal downscaling of satellite data with in situ observations, Earth Syst. Sci. Data, 17, 1977–2006, https://doi.org/10.5194/essd-17-1977-2025.

Hausfather, Z., K. Marvel, G.A. Schmidt, J.W. Nielsen-Gammon, and M. Zelinka (2022). Climate simulations: Recognize the 'hot model' problem. Nature, 605, 26-29, https://doi.org/10.1038/d41586-022-01192-2.

The GlaMBIE Team (2025). Community estimate of global glacier mass changes from 2000 to 2023. Nature 639, 382–388 (2025). https://doi.org/10.1038/s41586-024-08545-z.

Jouvet, G., Huss, M., Funk, M., and Blatter, H. (2011). Modelling the retreat of grosser aletschgletscher, switzerland, in a changing climate. Journal of Glaciology, 57(206):1033–1045. https://doi.org/10.3189/002214311798843359

---

## Referee Comment (RC3)

**Summary**

The manuscript of Rückamp et al. (2025) uses a full-stokes ice dynamics model together with an energy-balance model to simulate the glacier evolution from 1997 to 2100 for two well-known and well-studied glaciers in Central Europe, the Greater Aletsch Glacier in the Western Alps and the Hintereisferner in the Eastern Alps. By using observed ice velocities which were, in case of HEF, manually extracted just for this study, the basal friction parameter and rheology parameter were constrained. Observed MB gradients were used to tune the SEB albedo parameters. In general, they find that HEF is disappearing no matter which scenario is chosen while Aletsch glacier may survive for lower emission scenarios. These outcomes were expected, as they were equally found in other studies. However, the methodological approaches of combining ISSM together with the EB model of Evatt et al. (2015) by including the observed ice velocity and MB profiles to tune some free parameters are novel.

In summary, this manuscript has methodological novelty and directly compares two well-observed glaciers by using observations, model and calibration approaches that haven't been used before. Overall, the topic is interesting, the methods are understandable and the results are described in-depth. It is also great that all data and code will be made available (if I understood it correctly). However, there are a few major aspects that need to be revised, such as some description within the manuscript, an uncertainty analysis of your projections, the comparison to projections from other glacier models, the generalisability of the outcomes to all glaciers in Central Europe and the greater implications of the study. Therefore I believe that the study needs some major revisions before it is ready for publication. I don't expect that all of my comments are directly addressed, but I hope that some kind of uncertainty analysis can still be done together with clarifying some of the analysis and interpretation.

Thanks a lot for this nice manuscript. Please don't feel overwhelmed by my many comments, just try to consider my suggestions when somehow possible.

**Major Comments**

**1. Manuscript in general:**

- Overall the manuscript is quite long. Some "figure descriptions" can be condensed as these are already visible in the figures. For example, by always separating HEF from GAG, there is quite a lot of repetition in the manuscript, and it makes it more difficult to actually compare the behaviour of the two glaciers which is one of the actual motivations of the paper, or not? Would it be possible to reduce the amount of text by directly comparing the two glaciers? In addition it seems like sometimes the same things are described in different ways (for HEF, the "mostly gone" definition was introduced, while for GAG, you write again "... ice volumes drops below 10%" ...
- I found a few statements that are without references or proofs. Please double-check these (see my line-by-line comments). Some passages within the discussion and conclusion could be a bit more concrete and related to your specific study.
- See more suggestions in the line-by-line comments.

**2. Added value of this study and comparability to other glacier model simulations**

- What was the goal of this study? Creating the "most robust" glacier projections of GAG and HEF by using a full-stokes model together with an EB model? Or was it to understand if it results in other estimates than existing glacier models? On line 75, you write that the aim is to "investigate the potential variability of glacier responses in the frame of physical process representation". I did not feel like this was the aim of your paper. You analysed two glaciers (see 'extrapolation comment' below) and did not look into the uncertainties within your approach (see 'uncertainty comment').
- In Sect. 7.2, you do a comparison to other glacier models which is great. Though, it
  doesn't seem like your study's goal was to directly compare results to other glacier
  models. You used other weather-station corrected ERA5 climate data, other climate
  models (with/without EURO-CORDEX), calibration data, initial volumes, RGI
  versions, (see line by line comments) which makes comparisons very difficult to
  interpret.
- Conclusion I. 528 onwards: I completely agree that it is difficult to compare your estimates to other model estimates. But this entire paragraph somehow also describes the limitation of your study, so I am wondering why you did not choose to do standardised tests in your study, e.g., by comparing directly SIA vs full-stokes. Of course, this is difficult to do, as SIA flowline-models are not really expected to match ice velocities. But, at least showing a range of different full-stokes modelled outcomes (e.g. in terms of equifinality and/or observation uncertainties) may help to understand how variable projections can be within the full-stokes approach.
- What you could for example do is to compare your glacier volume estimates at the RGI year to the ice thickness community estimate from Farinotti et al. (2019). As many large-scale glacier modelling studies use that estimate for their initialisation, this comparison would be nice to have. Another example is that you use RGI7, while most (all) existing modelling studies in Central Europe use RGI6. How different is RGI7 to RGI6 on these two glaciers? All these aspects, together with trying to compare projections from exactly the same climate models (maybe just use one climate model where data exists for all studies) and/or doing an internal uncertainty assessment of your model (see uncertainty comment below) would be necessary to understand a bit better from where the differences come from.
- If a comparison of the exact same climate models is not possible, then you may reduce the "interpretation" within the discussion Sect. 7.2, and just show the figure with the potential variability of outcomes. In the following are some important aspects for Sect. 7.2 to consider.
  - You write that under RCP 8.5, JH19 results in more mass loss, while under RCP 2.6 it results in less mass loss. Then, you later argue that this is because JH19 used a TI-model which overestimates melt. Isn't that contradictory? Why is JH19 not resulting in more melt as well for RCP 2.6? If you don't look into the full spectrum of possible MB parameters (equifinality), these statements are very difficult to validate. Specifically you mention in the next paragraph that "Under RCP 2.6, GloGEMow predicts a volume reduction of 57.6% for GAG, aligning somewhat more closely with our forecast than the JH19 study; likewise, for RCP 8.5, GloGEMow projects a 89.4% volume decline, which is close to our estimate than the JH19 study." Both JH19 and GloGEMflow (Z19) are temperature-index models, but are apparently more different compared to your study and Z19. For me, this shows that other

- aspects like equifinality or downscaling choices may explain these differences and not specifically the choice of TI-model vs EB model.
- What I find interesting is for example that OGGM projects regrowth for GAG under SSP1-2.6, but your study did not. However, as you are likely not comparing the exact same climate models, it is very difficult to interpret if this comes from the SIA vs full-stokes choice, the different mass-balance model, climate downscaling choice or actually from the different climate model ensemble where the one shows some regional cooling while the other does not. Is it possible to look into this further?
- From where do you have the glacier model evolution data of GAG for the RCP 8.5 / RCP 2.6 scenarios of OGGM. Are these really available from Schuster et al. (2023). I couldn't find them there. Or did you use the OGGM standard projections? I assume you show the ensemble median over the different five climate models and over the various calibration approaches and TI model choices? If you can compare the exact same climate models: Is the variability of ensemble members from Schuster et al. (2023) similarly large as the differences between your study and Schuster et al. (2023)?

**3. Missing uncertainty estimates:**

- In the abstract, you claim that you give a well-constrained estimate of HEF and GAG projections that complements large-scale modelling efforts. Although you do use more complex and physical models (full-stokes ice dynamics, EB SMB modelling), I am wondering about the uncertainties in your input data and the equifinality of the additionally introduced free parameters. In my opinion, analysing whether different parameter combinations within the "input uncertainty and equifinality" space affect the projections would be essential to understand how well-constrained your projections are. You mention some of these uncertainties qualitatively in the discussion, but you haven't done any quantitative uncertainty assessment yourself. Doing this is difficult, but at least a rough analysis with multiple model estimates is necessary in my opinion.
  - For example, aren't there any other parameter combinations that result in a similar performance as the chosen one for the albedo calibration? If yes, how does that equifinality influence the validation and projection results?
  - By using an EB instead of a temperature-index model, you also have an increased equifinality due to the large amount of introduced new free parameters that need to be downscaled (SW, LW radiation, temperature, surface wind speed, humidity, precipitation). You mention the downscaling uncertainties in the discussion (I.427 to I438) but do not really do anything about it. I am wondering if the quality of the data is sufficient to apply an EB model without accounting for its uncertainties. For example, on I.173, I was thinking: How certain are the radiation gradients? And, isn't it strange to use a precipitation gradient for HEF that was only used before for GAG?
  - The data assimilation does not at all account for the uncertainties in the ice velocity. How strongly can you trust your ice velocity observations? Can the velocity extraction approach similarly be applied to other glaciers? How different is this approach to e.g. Millan et al. 2022?

- Of course you can't check all of these aspects. I am just wondering if the glacier projections are sensitive to these assumptions? Are these choices less important than the choice of using SIA vs. full-Stokes or a TI-model instead of an EB model?

**East-West comparison (Sect. 7.3)**

- Isn't the longer lifetime of GAG also the higher elevation-area distribution (see Fig. 14)? Not just the larger elevation range? It seems like you partly mention that aspect in I. 509 onwards, but it would make sense to condense these things all together. Specifically, GAG has most of its area around 3500 meters, while HEF has it more around 3000 meters.
- These area-elevation distributions may also explain why the MB gradient is difficult to match at the upper part of HEF (Fig. 6): there is actually not so much area at the highest elevation-bands, which is quite different to GAG. Does this specific shape of the area-elevation distribution also explain parts of the response differences of the two glaciers.
- What is the influence of the glacier slope? Is the current HEF steeper than GAG, and how will that evolve in the future? The different glacier geometries could e.g. be mentioned in I. 504 508.

**4. Extrapolation of HEF and GAG results to all over Central Europe:**

- I find it difficult to extrapolate results from HEF and GAG to the entire Alps. You could check how representative HEF and GAG are by analysing their characteristics compared to all other glaciers in the Alps, and/or compare per-glacier model estimates from regional-scale glacier models of HEF/GAG to regional-scale estimates. Maybe GAG is "representative" in some sense for the entire glacier mass evolution as it represents a relatively large fraction of the entire glacier mass in Central Europe, but this is not mentioned/analysed here. In addition, the "representativeness" also changes over time.
- I. 502: "In the worst scenarios, most glaciers also disappear at the end of the 21st century". To my knowledge, almost all (or even all) published studies over Central Europe write about glacier mass evolution, not number of glaciers. From which source do you have this estimate?
- Conclusion on I. 524: "Our findings suggest that glaciers in the eastern European Alps are likely to diminish by the mid-21st century, and only larger glaciers with higher elevation ranges in the western European Alps will remain until the end of the century.": Can your study really prove that sufficiently? Please consider removing or rephrasing.
- I am missing a little bit the discussion that your approach is only possible for these two glaciers because of the available data (good-quality remotely-sensed ice velocities and MB profile). Specifically because you do not account for the uncertainties. Maybe add a paragraph on the following aspects: Could your specific model combination approach (using ISSM and a SEB model) also be used for follow-up studies on the two glaciers or on other glaciers? How can your study be used to improve/validate regional-scale glacier modelling/projections?

**5. Novelty of the data assimilation approach (l. 78):**

I am not an expert in data assimilation, but isn't e.g. IGM (Jouvet et al. (2023) or

Cook et al. (2023)), Jouvet et al. (2019), also using observed ice velocities for the inversion? Or is it the specific approach with ISSM that is novel? Please specify.

**6. Climate model choice and scenario descriptions:**

You compare RCP scenarios to SSPs. You show that RCP2.6 & RCP 8.5 result in less warming over Europe than SSP1-2.6 & SSP5-8.5, and with that in less glacier volume loss. Is the reason for these differences the EURO-CORDEX downscaling of the RCP scenarios, or do you see the same effect globally, i.e. does the chosen climate model ensemble result in less global warming for RCP2.6/RCP8.5 vs. the SSP scenarios? That means, please mention the median global warming from the different scenarios for your chosen climate model ensemble (with that you may also rephrase I.421 to I.426 and other related line-by-line comments).

**7. **EB model calibration approach with calibration vs validation data** (Table 1 & Fig. 7, Sect. 5.2 & 5.3):**

- Do I understand correctly that you use the average observed SMB gradient from each year (i.e., n=13, 8 gradient observations). That means you do not use the entire MB profiles, but the timeseries of MB gradients and not just the average MB gradient over the time period (i.e., one observation), correctly? I think it would be important to clarify a bit more this calibration procedure within Sect. 5.2.
- I am specifically asking because I was first astonished to see how well the cumulative MB is matched. If you used the entire SMB profile timeseries, the cumulative MB over that same time period would not be anymore a true "validation" as it is somehow an integrated value of the MB profile? But, as you wrote in I. 340 to I. 342 that you used independent model fields, I expect, you only used the gradients (such as e.g. 0.007 m w.e. m-1) without an information of e.g. the intercept. These MB gradients, are, I guess, sufficiently "independent" enough from the cumulative MB. But, are they also completely independent from the elevation change observations?
- Related to that (specifically on I. 322): How much does it make sense to match the gradient over the entire elevation-area distribution, if there is at the upper part only little glacier area (in case of HEF)?

**Minor Comments**

- maybe replace always ice volume/ice area to glacier volume/area (all over the text). At the moment it is a mix of both.
- In all maps (Fig. 5, 7, 8; always a,b), it would be quite interesting to see the differences between modelled estimates and observations. I guess this is not always possible, as the observations are partly only available on point scales, but at least where possible, this would be maybe great to at least have a look into instead of the modelled values. The scatterplots are relatively difficult to interpret due to the many overlapping dots.

**Abstract:**

I.2: maybe mention here also the used surface mass balance model complexity?

I.8: "comprehensive glaciological observations"  $\rightarrow$  maybe clarify that you use in-situ MB gradients for calibration

- e.g. I. 11-12: maybe mention the median global warming in 2100 from these RCP/SSPs, otherwise it is unclear to understand. The choice of climate models strongly influences the actual warming within one RCP/SSP scenario ...
- I. 15-16: "...; however, a rough model-intercomparison study reveals a large spread of volume projections with the different glacier models.": would be great to compare this to a model-internal projection spread if you can do this additional analysis (see major comments)

**Introduction:**

I.20-26: references missing, a bit "vague" ...

- I.28-32 vs I.32-35: in the first sentence you mention different applications (not the models "per se"). Therefore, I would recommend to write "regional-scale/global-scale projections". You also present examples, therefore, it would be good to add "e.g." ... maybe also mention e.g. an IGM study (e.g., Cook et al., 2023) as that one emulates full-stokes.
- I.32: Maybe remove "due to computational constraints", because there are other reasons such as the only poor velocity data available for the inversion on a regional to global scale compared to HEF and GAG which are one of the best monitored glaciers world-wide
- I.35: "In most of these models": well, actually only in two of the four mentioned models. Please adapt and differentiate between retreat parameterizations (GloGEM, PyGEM) versus SIA (GloGEMFlow, OGGM).
- I. 37: "Despite potential shortcomings ... ": when doing regional projections, there are always shortcomings, but these are often due to missing glacier observations, thus consider reformulating.
- I. 38-39: RCP8.5/SSP5-8.5 is by today's standards a **very** high-emission scenario resulting in more than 4.0°C global warming in 2100. It could even be 4.5°C depending on the choice of climate models. I would suggest adding "very" to high-emission scenario.
- I. 41: "or the low-emission scenarios RCP 2.6 and SSP1-2.6, which are basically in line with the global warming target of 1.5°C ... (UNFCCC, 2015)": Please double-check the resulting global warming of your ensemble of chosen climate models, I am pretty sure it is higher than 1.5°C.

**I.44-47: some references?**

- I.58-62: references are missing (daily vs. monthly available in Schuster et al., 2023, bias-correction in Weathers et al. (2025)). What about the influence of equifinality (Rounce et al., 2020; Schuster et al., 2023)? These may also be very important sources of uncertainties.
- I.75 & I.81: Related to one of the 'Major Comments': "To investigate the potential variability of glacier responses in the frame of physical process representation" --> you compare two different glaciers, how largely do they represent the variability of glacier responses within Central Europe? Maybe the "aim" should be reduced a little bit? Similarly you write later: "In

order to capture and analyse regional differences of ice volume loss, the model is applied to two valley glaciers in the Alps". By just choosing two glaciers, you can't capture the regional differences. Please adapt.

I.77: SMB is not yet defined, I think.

Fig. 1: for GAG another time period is chosen than for HEF, isn't there a common time period? Also mention in the caption that these are the outlines from 1850 until 2015 (I guess they are not all coming from RGI7?). From which study do these outlines come from? At the dz legend: --> is the unit "m w.e. yr-1"?

**Study sites**

- I. 90: "outlines from the last decades": maybe replace by "outlines from 1850 until 2015"
- I. 94: "15km³, 900m": at which year? Also in 1999, if yes, consider joining the sentence together with the last sentence to clarify this. Maybe also mention the percentage of volume relative to the entire Central Europe (I think around 10%)?
- I. 97: What means "relatively dry", "very large amounts of precipitation"? Any quantitative numbers?
- I. 107: HEF has less than 5% of the glacier volume of GAG. Maybe it is worth mentioning that?. Eventually consider comparing these estimates to Farinotti et al. (2019) as many glacier evolution models use this estimate. What is the relative amount of glacier volume loss in a relatively recent but common time period for the two different glaciers?
- I. 111: Can you be more concrete? How many years are available for each of the glaciers? This is hard to see from Fig. 2.
- Figure 2: What means "b"? Maybe rather use SMB as in other parts (similar issue in e.g. Fig.7c). Annual SMB should be in mm w.e. yr-1? ... "since 1961": from which year to which year do you show the MB estimates? Could it be of interest to estimate with a scatterplot and a spearman rank correlation coefficient how much the two correlate? Consider increasing the size of the figure and eventually even adding a scatterplot to the left of it showing on the x-axis GAG annual MB and on the y-axis HEF annual MB.
- I.125-127: How representative are these precipitation data? It is quite far away from the actual glacier, but I know that there is no better possibility.
- I. 131: How do you justify a precipitation gradient?
- I. 130: "almost minor mass balance changes" -->when you look at the cumulative MB, it is still relatively considerable for HEF at least.
- I.132: typo in "regionla"
- I.154: Maybe also mention over which period you compare ERA5 with the raw GCM/RCM outputs for the "bias adjustment"? Another comment: Could you call the temperature lapse

rate/precipitation gradient the "statistical downscaling"? And the correction of the GCMs/RCMs then the bias adjustment? Just to better coincide with how the different steps are called in other studies.

I.158: As you use ERA5 corrected by weather stations as a dataset for the calibration, isn't another reason why you do the additional bias-adjustment for the ISIMIP GCMs the usage of weather-station corrected ERA5 instead of W5E5? Or not?

- Figure 3: Over which area is this? Averaged over the two gridpoints of HEF and GAG? I guess it is over the Alps, as described in line 160. Please clarify the exact region over which you average. As precipitation is shown as a ratio, the unit should be removed. Consider writing "Precipitation ratio" or "Precipitation relative anomaly". Consider also adding the amount of ensemble members to the legend. I guess this is n=10, n=10, n=65, n=22.
- I.163: It is great that you mention the regional temperature increases, but I think comparing the SSP1-2.6 regional warming to the global warming target of 1.5°C is a bit strange. As mentioned in one of the 'Major comments', can you also compute the global warming of this ensemble of GCMs/RCMs and compare that to the regional one?
- I.167: I would suggest adding this information directly into the statistical downscaling paragraph of I. 130. When I first read I. 130, I thought that you did not correct the radiation terms, for example.
- I. 212: I am not sure what the consequences are of setting the minimum ice thickness to 5m. Does that mean that an elevation where the glacier has an ice thickness below 5 m is considered as ice-free?
- I.230 onwards: over which time step is the albedo updated? The SMB profiles are only available on an annual scale
- I. 257: What does 0.05 and 0.25a mean, does 0.25a mean a geometry update four times a year?
- Table 1: HEF SMB gradient: shouldn't the reference be WGMS (2024) as the elevation-band mass-balance estimates are published there, or not?
- Fig. 5: (a) and (b): For GAG, it seems like low observed velocities are overestimated in the model. It would be nice to see the velocity differences directly on the map. Make sure to use a consistent description of the units (double-check with "The Cryosphere" regulations). At the moment, it is mixed ("m a-1" and "m/a"). "c" and "d": What do you represent with "vmod -vobs"? Is this the bias, and with that the mean over the differences? If yes, add an "average bar" on top of the equation. RMS or RMSE? Aren't you showing the root mean square error (RMSE) or do you really just show the magnitude of the observed velocity (RMS)? How many data points are used, please add the number. Do you believe that using an average over eight years for the initialisation in the year 2011 is problematic for a quickly melting glacier? Why did you choose a different velocity dataset than for Hintereisferner?
- I. 322: "RMS" shouldn't that be RMSE (similar on other lines before, and in Fig. 7)

- Fig. 6: Consider using the same color/line style scheme as in e.g. Fig. 7 for modelled vs observed estimates. That means maybe use always dashed lines for observations and solid lines for modelled estimates. At the moment, the green line in Fig. 6 is very thin and one has to compare that to the black line? What does the black line mean? Is that the average over the entire period? This is not very clear from the legend (I think the two "observed" labels have to be switched). You could also just have one legend outside of the figures with those labels that are the same and make sure to write out the accronyms. In (c) and (d), the lines seem to be even thinner than in (a) and (b), consider making them thicker. Also, the y-labels are different between (a) and (c). Can you maybe clarify in the figure or caption which period is used for "calibration" and which one for "validation"? Interestingly, I guess by coincidence, the "validation" period is matched better than the calibration period for both glaciers (in terms of RMSE and MSD).
- I. 348: Could these localised modelled spots come from e.g. "shading" which is not accounted for in the gridded radiation data?
- I. 366: The "mostly gone" definition and likely range used here seem to be very similar to what is used in the <a href="https://goodbye-glaciers.info/">https://goodbye-glaciers.info/</a> project. Is that a coincidence? Although the "mostly gone" numbers for GAG and HEF are available from that project (based on three large-scale glacier models, <a href="https://goodbye-glaciers.info/glaciers/RGI60-11.01450.html">https://goodbye-glaciers.info/glaciers/RGI60-11.01450.html</a>, <a href="https://goodbye-glaciers.info/glaciers/RGI60-11.00897.html">https://goodbye-glaciers.info/glaciers/RGI60-11.00897.html</a>), it does probably not really make sense to compare the numbers directly as other climate models are used, SSP126 is between 1.5 and 2.7°C and you would need to recompute your values to be relative to 2020 instead of relative to 1997.

Table 2: please use consistent descriptions. In the text you always had median [q17 to q83], In the table it is a mix of round brackets with sometimes "-" and sometimes "->". Why do you mention here the initial volume of the glaciers for two different time steps? The "gone" and "mostly gone" definitions are in both cases relative to 1997, or not?

Fig. 9&10: Why does ERA5 result in slightly less glacier volume loss than SSP1-2.6/SSP5-8.5 for both HEF and GAG? Is that because the bias-correction period is not the same as the ERA5 period? It seems like the RCP (EURO-CORDEX) scenarios match better to the ERA5 simulations for both glaciers.

Fig. 10&11: How do you define that a certain part of the glacier is gone? Is this the 5m threshold you mention somewhere in the methods section? Here it would be again great to have the global warming estimates for these two chosen scenarios.

I.409 : change "sime" to "some

I. 412: typo -> "methodological"

Fig. 14: It would be extremely helpful to use the same y-scales for the two glaciers. If it is really not possible, than mention this in the caption. The grey barplots are missing in the legend. Consider removing double legends, legends in (a) and (b) show the same, just show it in (a).

- I. 511 to I. 515: You mention at the beginning and at the end the same statement, please reduce one of the two sentences.
- I. 535: What do you mean with "GlacierMIP (2025)"? Maybe mention that the existing GlacierMIP studies (Hock 2019, Marzeion et al. 2020, Zekollari, Schuster et al. 2025) did not focus on that, but maybe this will actually happen in the next GlacierMIP round (i.e. GlacierMIP4).
- I. 518: Did you check how representative HEF and GAG are in terms of valley glaciers in the European Alps? I would rather say, that these two glaciers are among the glaciers with a lot of observations which allows to apply the data assimilation approach using full-stokes and an EB model. But how similar are these glaciers to other glaciers in terms of their area-elevation distribution? Wouldn't you need to check the RGI inventory and some statistics there to evaluate whether they are actually well representative? See related 'Major Comment'.
- I. 520: Here it would be great again to know the actual global warming of these scenarios.

**References:**

Farinotti, D. et al. A consensus estimate for the ice thickness distribution of all glaciers on Earth. *Nat. Geosci.* **12**, 168–173 (2019).

Weathers, M., Rounce, D. R., Fasullo, J., & Maussion, F. (2025). Evaluating the role of internal climate variability and bias adjustment methods on decadal glacier projections. *Earth's Future*, 13, e2024EF005624. https://doi.org/10.1029/2024EF005624

Rounce DR, Khurana T, Short MB, Hock R, Shean DE, Brinkerhoff DJ. Quantifying parameter uncertainty in a large-scale glacier evolution model using Bayesian inference: application to High Mountain Asia. *Journal of Glaciology*. 2020;66(256):175-187. doi:10.1017/jog.2019.91

Schuster L, Rounce DR, Maussion F. Glacier projections sensitivity to temperature-index model choices and calibration strategies. *Annals of Glaciology*. 2023;64(92):293-308. doi:10.1017/aog.2023.57

Jouvet G, Cordonnier G. Ice-flow model emulator based on physics-informed deep learning. *Journal of Glaciology*. 2023;69(278):1941-1955. doi:10.1017/jog.2023.73

Cook, S. J., Jouvet, G., Millan, R., Rabatel, A., Zekollari, H., & Dussaillant, I. (2023). Committed ice loss in the European Alps until 2050 using a deep-learning-aided 3D ice-flow model with data assimilation. *Geophysical Research Letters*, 50, e2023GL105029. https://doi.org/10.1029/2023GL105029

HOCK R, BLISS A, MARZEION B, et al. GlacierMIP – A model intercomparison of global-scale glacier mass-balance models and projections. *Journal of Glaciology*. 2019;65(251):453-467. doi:10.1017/jog.2019.22

Marzeion, B., Hock, R., Anderson, B., Bliss, A., Champollion, N., Fujita, K., et al. (2020). Partitioning the uncertainty of ensemble projections of global glacier mass change. *Earth's Future*. 8, e2019EF001470. https://doi.org/10.1029/2019EF001470

Zekollari, H., Schuster, L., Maussion, F., Hock, R., Marzeion, B., Rounce, D. R., ... & Sakai, A. (2025). Glacier preservation doubled by limiting warming to 1.5° C versus 2.7° C. *Science*, *388*(6750), 979-983.

Millan, R., Mouginot, J., Rabatel, A. *et al.* Ice velocity and thickness of the world's glaciers. *Nat. Geosci.* **15**, 124–129 (2022). https://doi.org/10.1038/s41561-021-00885-z

---

## Author Comment (AC2)

**Review #1**

We would like to thank the reviewer for the constructive comments that helped to improve the manuscript. We followed most of the reviewers suggestions and if not, they might be addressed by comments of the other reviewers; we tried to mark that accordingly. Based on all reviewer comments, we made considerable changes to the manuscript by improving the description of the climate data and we add a chapter 'Parameter sensitivity' in the Discussion. We have revised the manuscript accordingly and will be happy to provide a new manuscript. Please find below the reviewer's comments in black and a point-by-point response in blue.

Review of Rückamp et al. (2025) 'Future Retreat of Great Aletsch Glacier and Hintereisferner – application of a full-Stokes model to two valley glaciers in the European Alps'

**Summary**

This paper presents a full-Stokes modelling study of Hintereisferner and Great Aletsch Glacier over the course of the 21st century. Hintereisferner is initialised in 1997, Aletsch in 2011. The authors calibrate their model against observations up to the point these are available and additionally perform inversions for unknown parameters (basal sliding coefficient, ice hardness). The paper finds that Hintereisferner, even under a low-warming scenario compatible with the Paris Agreement, will likely disappear by mid-century, while Aletsch will likely survive in a dramatically reduced state. Under a high warming scenario, however, both glaciers will be (almost) entirely gone by 2100.

I think this paper is well-executed: the method is solid and the results convincing and thoroughly discussed. I have a few minor comments detailed below, the most substantial of which bears on re-writing some of the discussion to make it more impactful, but this is purely a question of emphasis and presentation rather than any sort of fundamental flaw in the paper. Therefore, I recommend that the paper be returned to the authors for minor revision.

Page and line numbers refer to those in the clean version of the revised manuscript.

**Major Comments**

- Discussion: see the detail in the individual minor comments below, but I think the discussion could be reworked a bit to make it snappier and so that it focuses on the more interesting points the authors raise. At the moment, a lot of it feels either very obvious or doesn't really lead anywhere.
  Based on the Comments by Reviewer 2 and 3 the Discussion is substantially rewritten. In particular we add a chapter about parameter sensitivity, and the chapter 'east-west comparison' is more generalized.

**Minor Comments**

p.2, l. 28-43: To bring this completely up to date, might it be worth including IGM in the list here? It's not been applied globally yet, but there are some regional studies, and it seems to be aiming for the same sort of applications as this paragraph is concerned with

IGM is undoubtedly a big tool but so far not applied to alpine glaciers under climate scenarios like RCP or SSP, therefore we excluded it here. However, since Reviewer 3 also suggested including IGM in the citation list, we followed your suggestion to bring the citation list up to date. See also answer to major point 6 of Reviewer 3. We also added:

*"The instructured glacier model (IGM, Jouvet, 2022; Jouvet and Cordonnier, 2023; Cook et al., 2023) emulates Stokes ice flow and is therefore a promising alternative to traditional solvers even on regional scales thanks to its high computational efficiency. However, IGM has not yet been applied to investigate mountain glacier evolution under RCP or SSP climate scenarios. Cook et al. (2023) found that the resulting committed ice loss exceeds a third of the present-day ice volume by 2050."*

p.3, l.81: It may well be correct that this is the first time the specific form of data assimilation used here has been applied to mountain glaciers, but it's certainly not true in a more general sense, which is how this sentence reads. I might suggest toning this down slightly.

The sentence is rewritten to: *"The latter is a common initialization approach for large-scale ice sheet modelling (e.g., Goelzer et al. 2020), but rarely applied for mountain glaciers."*

p.6, l.125: Delete 'a'
Done

p.6, l.132: 'regional'
Done

p.6, l.139: The plural of 'RCM' is 'RCMs'. Same with 'GCMs' on the next line. No need to put an apostrophe in. Make sure there aren't other instances – I won't bother flagging them all up.
Done

p.7, l.164: 'As expected'
Done

Section 5.1: I think there are two points here. First, what happens if the inversions are done the other way round, i.e. an initial guess for beta is provided to do an inversion for B and then beta is inverted based on that B profile? Ideally, the model would end up in the same place, but it might not, and I think it would be worth checking this. Otherwise, it all feels a bit circular and arbitrary.

We do not agree with the reviewer. We think setting up the inversion for C and B the other way around  as suggested by the reviewer didn't make sense and not worth checking. To explain: We initialize the ice hardness B with an analytical 1D temperature profile, which is not perfect, but somehow resembles the overall temperature distribution in the ice (at least colder ice at higher altitude compared to lower altitude; warmer ice close to the ice base compared to the ice surface). In case of running the friction inversion first, the rheology is somehow well constrained. Setting it up the other way around requires a good estimate of the friction parameter, which is very difficult to constrain. Based on our inversion chain, we think the setting is more realistic: The friction inversion is inline with a rough estimate of the temperature, the subsequent rheology inversion captures missed ice dynamic processes e.q. ice weakening by crevasse zones which can not solely be represented by the initial temperature and the inferred friction coefficient.

Second, I'm not sure I agree about the L-curves. The one in panel d does have a clear corner and therefore optimal value, but the other three, especially a and c, really don't. Now, I don't think it would make much difference whether the next value up or down were picked on any of the curves, but I might walk back the statement about lambda being easily pickable at l.302-3.

Agreed, we rephrase ist accordingly.

p.12, l.279: Is 'thermochemical' a typo for 'thermomechanical'?

Done -> thermomechanical

p.14, l.315: There's a missing closing bracket for the one opened before 'corresponding' on the previous line

Done

Figure 9 caption: Here, an apostrophe is needed for 'scenarios''

Done

p.17, l.362: ...and here one isn't – it's just 'SSPs' for the plural

Done

p.17, l.363: 'diverges'

Done

p.17, l.368: 'projects'

Done

Figure 10 caption: 'present'

Done

Figure 11 caption: same as for Figure 9

Done

p.19, l.388: 'lead'
Done

p.19, l.389: 'projects'
Done

Figure 12 caption: same as for Figure 10
Done

p.20, l.409: 'some'
Done

p.21, l.416: There's a word missing – 'The basal friction parameter and thus the...what...associated...'
Done

p.21, l.417: OK, yes, fair enough, but are basal erosion rates really that high that one would expect them to lead to any sort of noticeable impact on the glacier's behaviour in the next 80 years? This feels a bit of a reach rather than a worthwhile point to include in the discussion. I'd replace it with changes in the glacier's stress regime, which would alter the distribution of sticky and slippery spots on the bed (and therefore beta) in a noticeable manner on the timescales relevant to this study.
Yes, you are right. Maybe basal erosion rates are not relevant on the investigated time scales. Just to clarify, the inferred 'optimal' friction parameter should be independent of the stress regime, as it is a material parameter describing the basal friction. So, we just drop basal erosion and rephrase it accordingly.

Section 7.2: I get that the authors are not attempting an exhaustive comparison with all previous studies, but this entire section pretty much just says 'different models with different set-ups give different results and we're not going to dig any further', which is maybe not the best use of a page- and-a-half of the discussion. That could be asserted in one line and no one would bat an eyelid. I feel it would be more valuable if the authors focused on the comparison with the Jouvet and Huss (2019) study, which should be most comparable, and tease out why they think the results are different, which then tells us something useful about the impact of different parts of the model setup. I would probably then just reduce the comparison with OGGM and GloGEM to a couple of lines, noting the difference, because there I agree with the authors that the models are so fundamentally different that it's very difficult to establish exactly where the differences in the results are coming from.
We agree with the reviewer that such a one-line conclusion can be drawn from the comparison and that the paragraph can be shortened. However, we wanted to demonstrate the large variability of the results in order to support our conclusion for the need of "standardized tests" (Linie 533). Since Reviewer 2 didn't complain about this comparison and Reviewer found the comparison "great" we decided to keep it as is (while making the changes suggested by Reviewer 3).

Your suggestion of directly comparing our GAG results with JH19 is indeed very interesting. However, the JH19 doesn't provide any model output publicly making a comparison of e.g. SMB changes, elevation changes etc. impossible. In addition, performing such a detailed comparison should then be performed together with the authors of JH19.

Section 7.3: Similarly, this feels like quite a lot of effort to go to to point out that Hintereisferner is going to do worse than Aletsch because it's smaller, at a lower elevation, and in a drier part of the Alps. It's again something that would be an uncontroversial one-line assertion, particularly if the authors reference it to Figure 14, which summarises all the relevant information. The other material in this section about what this tells us about Alpine glaciers more generally is, I think, the interesting bit, so I'd suggest cutting down the first few paragraphs at the bottom of p. 23 and the last paragraph on p. 24 and condensing it into one paragraph that points out the obvious reasons for the different behaviour of the two glaciers.
We think the paragraph at the bottom of page 23 cannot be deleted as it explain the Figure 14. However, the next paragraph on page 24 is reworked as we add some new material on this discussion as suggested by Reviewer 3.

**Review #2**

We would like to thank the reviewer for the constructive comments that helped to improve the manuscript. We followed most of the reviewers suggestions and if not, they might be addressed by comments of the other reviewers; we tried to mark that accordingly. We didn't follow one of his/her major comments (3) by including the debris effect for GAG as we think this requires a more sophisticated model for debris evolution compared to simple ad-hoc approach which is beyond the scope of this paper.
Based on all reviewer comments, we made considerable changes to the manuscript by improving the description of the climate data and we add a chapter 'Parameter sensitivity' in the Discussion. We have revised the manuscript accordingly and will be happy to provide a new manuscript. Please find below the reviewer's comments in black and a point-by-point response in blue.

**Summary**

This manuscript of Rückamp et al. (2025) presents projections of the future evolution of two Alpine glaciers (Great Aletsch Glacier and Hintereisferner) using a full-Stokes ice flow model coupled with a surface mass balance (SMB) scheme driven by the surface energy balance method. The model is initialized using observed velocity fields, and future projections are forced with bias-corrected climate data under different scenario's. Their projections suggest that Hintereisferner will vanish mid-century even under the low-emission scenario, while Aletsch will significantly shrink, possibly with a near-complete disappearance under high- emission scenarios by 2100 AD. The paper addresses an important scientific and societally relevant topic, as glacier-scale projections using full-Stokes models remain relatively rare, and the work provides valuable insight into the dynamic response of two iconic glaciers. The manuscript is generally well written (also some typo's remain and some things can be formulated better) and the results are of interest to both glaciological and the broader climate-impact communities.

The manuscript is scientifically valuable and should ultimately be suitable for publication in The Cryosphere. However, several scientific and some small linguistic issues here and there need to be addressed before publication. I recommend major revisions to (i) strengthen the treatment of the model forcing and bias-corrected climate data, and (ii) elaborate a bit more on model uncertainty related to the future projections. I suggest publication once the comments below have been addressed.

**Major comments**

1. Model forcing and bias-corrected climate data: The model forcing is described in Section 3, but I would like to have a more detailed explanation of some things that have not become very clear to me from the text.

1.1.    Applied bias correction: In L124, you mention a "simple correction" has been applied to ERA5-Land data using observational temperature and precipitation data, and in L132 you say that you "shifted and scaled" the temperature and precipitation for the 1961-2023 period. Can you be more specific about what you did? Did you just scale the data so the mean of the overlapping period matches? Did you also adjust the variability (standard deviation)? Did you scale temperature and precipitation differently (i.e. additive for temperature, multiplicative for precipitation)? Did you apply the bias correction with reference data using a daily temporal resolution? Moreover, you mention other data as well for the SEB model (wind speed, radiation, humidity) but some explanation for a bias correction procedure for those are not mentioned for the "observational" (1961-2023) period. Did you scale these too for 1961-2023? I think the whole procedure can be further elaborated step by step into more detail to make it clearer for the reader. Moreover, applying a bias correction to the future climate data is a necessary step but it may result also in a change in the long-term trend of the data contained in the original output. Have you checked the preservation of the trends after bias correction

We agree that this paragraph was not well presented and requires rewriting. We did a substantial editings by first introducing the following processing steps:

"*1. Identification of error between ERA5 data and meteorological recordings*
*2. Bias adjustment of the projection data to the ERA5 reference data.*
*3. Downscaling of climate data, either ERA5 or GCM/RCM, to the glacier area*"

All steps are then further elaborated and we use a clear wording for bias adjustment (step 2) and downscaling (step 3) as requested by reviewer 3 (see answer to Line 154 by reviewer 3). In step 1 we then provide some error estimates between meteorological recordings and ERA5 data that hopefully answers your comment to Sect 7.3.

1.2.    Compatibility of ERA5-Land for glacier-specific meteorology: You explained well that you use ERA5-Land for the bias correction of the meteorological forcing. Have you checked how well ERA5-Land and your observational data (from the meteo stations) agree during the overlapping periods, using some statistics? It is, for example, well-known that reanalysis data like ERA5-Land may not fully represent small-scale processes like glacier winds (which should be however the dominant wind regime over the glacier), but rather synoptic-scale wind patterns. Is this also applicable to your data? You briefly mention this in the text (in Section 7.1) in a qualitative way but I think that this warrants some further investigation or at least a statistical quantification of the level of agreement during the overlapping period.

We certainly did some statistical analysis, but did not include it in the manuscript. Apart from that, we see the point of the reviewer and added some more detail to the offset-correction section in order to show how this already improved data agreement. Moreover, we added additional information regarding a statistical comparison. The section receives considerable improvements inline with your comment 1.2.

1.3.  Model selection for future projections: It would be beneficial to have a list of used GCM/RCMs (for example as an Appendix) that are used for the future projections. I say this because a subset of the CMIP models are found to be 'too hot' and may lead to an overestimation of glacier mass loss. Hausfather et al. (2022) has suggested that models with a TCR that lies outside the 'likely' range of 1.4-2.2 °C should be left out to avoid overestimation. You may look into this or at least mention it in the text.

Thank you for this suggestion. However, there is also a correspondence to Hausfather et al. (2022) by Bloch-Johnson et. al. (2022), that the approach of excluding (too hot) models is particularly not encouraged (at least if the GCM/RCM is not compromised by a known physical error). Therefore we don't follow your suggestion, as the GCM/RCMs used are not known to be compromised by physical errors. We will include a GCM/RCM list for the CMIP5 and 6 models in the Supplement and reference them in the text. The tables will look as follows:

**Table S3.** Overview of projected year when HEF are gone (i.e. volume drops below 1% of the initial volume) or mostly gone (i.e. volume drops below 10% of the initial volume) of the 10 SSP 585 and SSP 126 ISIMIP3b GCM simulations utilized in this study. If a percentage value is given, the volume has not fallen below the corresponding threshold value in 2100 and the remaining glacier volume is given in comparison to 1997. The simulations marked in red are the ones with a future volume evolution that is closest to the multi-model median.

| GCM | SSP 585 | | SSP 126 | |
|---|---|---|---|---|
| | mostly gone | gone | mostly gone | gone |
| CANESM5 | 2040 | 2048 | 2039 | 2052 |
| CNRM-CM6 | 2045 | 2053 | 2048 | 2073 |
| CNRM-ESM2 | 2047 | 2064 | 2044 | 2086 |
| EC-EARTH3 | 2032 | 2040 | 2036 | 2046 |
| GFDL-ESM4 | 2051 | 2064 | 2047 | 1.85% |
| IPSL-CM6A | 2044 | 2052 | 2047 | 2075 |
| MIROC6 | 2043 | 2051 | 2037 | 2058 |
| MPI-ESM1-2 | 2051 | 2061 | 2051 | 3.24% |
| MRI-ESM2-0 | 2036 | 2043 | 2041 | 2059 |
| UKESM1 | 2033 | 2041 | 2035 | 2042 |

2.  Uncertainty of future model output: Some more incorporation and/or discussion of model output uncertainty by the variation of some key variables (for example related to the SEB/SMB) is warranted in my opinion. Also, the evolution of supraglacial debris is not included in the model, which I can agree

on given the current minor debris extent on both glaciers, but its potential future effects should be more clearly acknowledged in the text.

2.1. Model output uncertainty: In my opinion the discussion of the model uncertainty can be improved. I understand that most quantified model uncertainty in the Figs. 9 and 11 and in the uncertainty intervals comes from the different GCMs/RCMs and future climate scenario's, but what about the uncertainty of the SMB profile that results from these climate data? This is only very briefly mentioned in section 7.1 but still important because the future evolution of the glaciers is in the end SMB-driven. For example, from my understanding, the gradients used in this study (L169-172) are taken over from other studies and not tested for validity for this study. How do they affect the SMB profile? In other words, how robust is the SMB model to internal parameter choices? This can be briefly discussed. I understand that it is computationally expensive to also include the sensitivity of various parameters to the SMB model in all calculations, but I think the manuscript would benefit from at least a sensitivity analysis of the SMB model (for example, with a Monte Carlo approach or a figure or table summarizing the sensitivity to major parameter uncertainties).

Thanks for this suggestion. See our answer to major point 3.3.3 of Reviewer 3. In brief we run a few sensitivity experiments by changing key parameters and add a new chapter in the Discussion. A Monte Carlo analysis is way too expensive, we think, for the goal achieved here.

2.2. Mentioning of supraglacial debris cover effects: Currently, debris cover effects are minor but certainly already present. A clear inversion of the SMB gradient in the lower parts of Aletsch is seen in your figure 6b, and a clear dampening of the surface lowering on the southeastern part of the Hintereisferner is apparent from your figure 1b. This coincides with an area of debris cover on the snout, which is clearly seen on satellite imagery. The effects of debris are indeed highly glacier-specific (depending on debris thickness/area, debris properties, and climatic conditions) and future trends are difficult to establish. However, given that the effects of debris are already present to some degree and generally expected to increase in the future (e.g. due to enhanced melt-out, increased bedrock exposure and slope instability, decreased flow velocities/debris discharge off-glacier) and given that it can potentially have an impact in the future (which was already modelled on the specific Aletsch glacier by Jouvet et al. (2011)), I think it warrants at least some further explanation of why you didn't include it and/or discussion on its potential effects in the paper.

Your are right, the debris effect might be important (at least on longer time scales (consider lag time between deposition and entrainment)). GAG, particularly the snout retreat, is known to be influenced by a

debris cover. However, just introducing a debris cover is misleading in terms of glacier melt, as the enhancement/damping depends on the debris thickness (damping of melt rates for debris larger than approx 7cm, otherwise enhanced melt (Østrem curve, Østrem 1959)). A dynamical model for debris (redistribution of supraglacial debris) is needed to cover these effects (in development for ISSM and hopefully we can submit a paper draft soon). Although, GAGs snout SMB might be influenced by the debris effect, we refrain from setting up a dynamical debris model (which comes along with a lot of uncertain and tuning parameters with almost no observational basis). We will add a sentence to the discussion.

**Minor comments:**

L20: The new papers of Dussaillant et al. (2025) and GlaMBIE (2025) may be a good reference here.
Done

L56: you can elaborate maybe a bit more on the specific advantages of using full-Stokes over the SIA or higher-order approximation (e.g. what type of stresses are included). Does it really make that much of a difference and if yes, in which areas do you expect the most significant improvements?
We did some reviews about SIA vs. FS flow approximation and their relevance in L53. Maybe it was too short and we extended this section, with briefly mentioning the relevant stresses (but just as a comparison for FS vs. SIA):
*"Until now, there has been no clear understanding of whether FS simulations have the ability to narrow uncertainties in current sea-level predictions (IPCC, 2013; Meredith et al., 2019; Oppenheimer et al., 2019). This is a challenging task, since assessing whether FS is needed compared to simpler models is complicated because of many interacting processes (e.g., numerical model used, initialization procedure, design of forward experiments). However, FS models are the most accurate representation of viscous ice flow and, compared to SIA, FS resolves lateral shear stresses and captures the entire stress tensor; for instance lateral drag by valley glacier sidewalls might be better represented."*
Reviewing the several HO approximations and their relevant stresses is way too much for this paper. There is a zoo of HO approximations with different stresses considered implemented in ice flow models (see Hindmarsh 2004).
From my experience the SIA and HO models usually work really well for glaciers, so what are the main advantages of using the full Stokes when compared to other approximations? As you mention in the conclusion, a detailed quantitative comparison would be out of scope but I do think it can be briefly discussed.
Can you clarify your statement "working well"? What is your criterion for good performance? From theory, SIA is not applicable for glaciers; it is designed for long extended ice masses with almost no bedrock undulations (Grever & Blatter 2009). In

the context of data assimilation and friction inversion, most ice flow approximations may look good in terms of surface velocity. However, this does not necessarily mean that the stresses are correctly captured. For example, SIA neglects, among others, transverse stresses that could be relevant for valley glaciers with lateral resistance.

L61: remove second )
Done

L71: you can maybe mention here the dynamic calibration procedure that is usually performed (e.g. artificially adjusting the historic mass balance after a steady-state spin-up so that the observed and modelled lengths over the historical period agree until present-day). Was this dynamical calibration procedure not feasible for your model? Were you able to reconstruct and compare historic front variations?
The method you suggest here is fundamentally different to our approach. Your suggested approach is one of the counterparts discussed in Zekollari et al. (2022); what we intent to say in Line 72-74. Tuning such a model so it matches historic front positions - and additionally agrees with observed velocities and ice volume, is expensive and very cumbersome, especially for full-Stokes. Therefore, we rely on the data assimilation approach.

L82: I think you can already briefly mention here why these two glaciers were chosen specifically and further elaborate on it in the section 2. They are for example not WGMS reference glaciers
HEF is a WGMS reference glacier (https://wgms.ch/products_ref_glaciers/)
and it is not the first time that they are modelled,
HEF was not modelled before with a detailed 3D ice flow model.
why are they specifically important and are they representative for the Alps in general?
We refined the text here and in section 2 and explained better why we selected HEF and GAG. They represent the largest glaciers with a valley glacier extension for the eastern and western part of the Alps, respectively. In addition, sufficient data are available to drive the inversion model.To our knowledge, HEF was not modelled before with a detailed 3D ice flow model.
*"The model is applied to two iconic valley glaciers in the Alps with a sufficient data base for the employed ice flow model: (1) the Great Aletsch Glacier (Switzerland) is located in the western Alps and the largest glacier in the Alps with a valley glacier extension. (2) Hintereisferner (Austria) is the largest valley glacier in the eastern Alps and has been classified as one of the 'reference glaciers' by the World Glacier Monitoring Service (WGMS)."*

L125: remove 'a'
Done

L132: adjust 'regionla'
Done

L170: The shortwave radiation is decreasing with elevation. Is this the incoming only or the total shortwave radiation (minus outgoing)? Usually, atmospheric transmissivity increases with elevation, enhancing the incoming shortwave radiation at higher elevations.

It is the incoming shortwave radiation. Gradients are based on the measurements by Marty et al. (2002). We added "*incoming*" for clarity.

L220: can you show the equation used for the cosine interpolation? Or at least a graphical representation of it in the Appendix maybe.

We consider such a function to be very basic and therefore not necessary to demonstrate. The graphical representation is as follows, which we think is easy to recompute:

[Figure]

L239: can you give an indication of the computational cost? How long does it take to run the model?

We added a rough number of the computational cost at the end of chapter 4.4. It reads: *"... This results in a mesh with ~1.11 million elements for GAG; ~0.15 million elements for HEF. [… ] Due to the different mesh sizes, the computational demand of the two glaciers is vastly different. For HEF, the future climate runs over 103 simulation years requiring 1.5h each on 128 cores (2 MPI tasks with each 64 cores). By contrast, a GAG future climate run over 90 simulation years requires ~20h on 672 cores (7 MPI tasks with each 96 cores). One CPU consists of 2xAMD EPYC 7702 64-Core Processor with 2.0 GHz."*

L234: I understand the albedo is used as a tuning mechanism, but do you have anything to compare their values to for checking its credibility? Data from an on-glacier AWS?

Unfortunately, we didn't find much in the literature. Studies we found are restricted to summer snapshots, not covering the full glacier or not available for our tuning/validation periods (e.g. Dirmhirn & Trojer (1955)).

L280: do you have evidence that both glaciers are indeed isothermal such that a thermomechanical coupling is not necessary?

This must be a misunderstanding, we don't assume that both glaciers are isothermal. Actually, we just mention that we don't run a coupled model. We run an inversion for

rheology. The inferred rheology is somewhat difficult to interpret as it covers the effect of the ice temperature and e.g. crevasse weakening (see Borstad et al. 2013).

Also adjust 'thermochemical'.
Done

L306: ice velocities after initialization agree really well, nice! What about the ice thickness? Can you provide an RMSE for those as well?
The ice geometry is still fixed to the observed state during the inversion approach. So, there is a perfect match except the interpolation error from the datssdet to the computational mesh.

L315: Add additional bracket )
Done

Figure 6 caption: can you indicate here again where the SMB observations come from where you tune against?
We think it is not necessary to provide the references again. They are provided in Table 1 and in the text (Paragraph starting at Line 111). In addition, Reviewer 3 also requested to avoid repeating information to keep the manuscript concise.
You show accumulation and melt separately, but do you have data to validate whether the distinction between these two is correct? Or do you only have observed data of the final SMB?
Unfortunately, we only have SMB. We delete "accumulation" and "melt" from that figure and just focus on SMB. See answer to line-comment of Figure 6 by Reviewer 3 where we posted an updated Figure.

L350: missing point at end of sentence
Done

L360: The term "disappearance" can be ambiguous (e.g. complete ice-free, negligible residual ice, disconnected patches, below a threshold volume or thickness?). You should clearly define here how you define glacier "disappearance".
We had defined the term "disappearance" in the next paragraph. However, this sentence is clarified to *"...HEFs ice volume disappears completely …"*.

L363: diverge -> diverges
Done

L368: project -> projects
Done

L371: Figure 10a and b displays -> Figures 10a and b display
Done
Figure 10 caption: presents -> represent
Done

Figure 9b and 10b: can you include the reference time period to which the volume loss (%) is compared in the y axis label? You may also want to include it in the captions of Fig. 10 and 12.
Done

L388: leading -> lead
Done

L389: project -> projects
Done

L401: 2800 m, a.s.l. -> remove comma
Done

L409: sime -> some
Done

L412: the accuracy of ice velocity retrieval from satellites depends on the acquisition method. SAR interferometry is usually very accurate for slower moving ice.
Yes, (differential) SAR interferometry is usually better suited for slower moving ice, however, the correct separation of the topographic phase (depending on accuracy and topicality of the used DSM) and atmospheric effects from the interesting deformation phase will be more challenging for very slow moving glaciers.

Section 7.3: I don't really like the title 'east-west comparison'. To me this section just reads like an attempt to generalize the behavior of the two glaciers over a certain region. To explain the different behavior of the two I think you can also elaborate more on (1) the climatic setting that may differ (can you give some quantitative climate data for both glacier environments to corroborate this?),
We changed the title of this chapter to *Generalization.* In Lines 509-515, we already provided differences in the climate setting. However, the updated Climate data section provides now annual values of air temperature and precipitation of both locations in order to give the reader a rough insight into this topic.
but (2) also the climate sensitivity of the glaciers related to their geometry. Hans Oerlemans did a lot of research into this and you can maybe compare the glacier characteristics of both glaciers to explain their different future behavior/sensitivities (mass balance gradient, overall slope, glacier size, hypsometry (e.g. large accumulation area vs. narrow snout), etc.). This also adds to the difficulty of generalizing glacier behavior for a certain region.
We add this as a caveat into the discussion: "*In addition to the regional setting of each glacier that probably influences the glaciers' response to increasing temperatures, the glacier sensitivity depends on glacier slope and exposition, ice thickness and area-elevation distribution, mass balance gradient and hypsometry (e.g. Oerlemans, 1992; Jiskoot et al., 2009)".* See answer to major point 4 of Reviewer 3.

**References:**

Dussaillant, I., Hugonnet, R., Huss, M., Berthier, E., Bannwart, J., Paul, F., and Zemp, M. (2025). Annual mass change of the world's glaciers from 1976 to 2024 by temporal downscaling of satellite data with in situ observations, Earth Syst. Sci. Data, 17, 1977–2006, https://doi.org/10.5194/essd-17-1977-2025.

Hausfather, Z., K. Marvel, G.A. Schmidt, J.W. Nielsen-Gammon, and M. Zelinka (2022). Climate simulations: Recognize the 'hot model' problem. Nature, 605, 26-29, https://doi.org/10.1038/d41586-022-01192-2.

The GlaMBIE Team (2025). Community estimate of global glacier mass changes from 2000 to 2023. Nature 639, 382–388 (2025). https://doi.org/10.1038/s41586-024-08545-z.

Jouvet, G., Huss, M., Funk, M., and Blatter, H. (2011). Modelling the retreat of grosser aletschgletscher, switzerland, in a changing climate. Journal of Glaciology, 57(206):1033– 1045. https://doi.org/10.3189/002214311798843359

**Review #3**

The authors thank the reviewer for constructive comments that helped improve the manuscript. The reviewer raised several points to be addressed; at the same time, the reviewer mentioned that these points cannot be all addressed by our study. We tried to solve several of the comments, for example, adding the parameter sensitivity and improving the "Generalization" chapter. In cases where we do not follow the reviewers suggestion we try to sharpen our description or adding caveats in the discussion. However, we followed several of the major comments and almost all of the minor comments of the reviewers suggestions, and if not, they might be addressed by comments of the other reviewers; we tried to mark that accordingly. Based on all reviewer comments, we made considerable changes to the manuscript by improving the description of the climate data and we added a chapter 'Parameter sensitivity' in the Discussion. We have revised the manuscript accordingly and will be happy to provide a new manuscript. Please find in the following the reviewer's comments in black and a point-by-point response in blue.

**Summary**

The manuscript of Rückamp et al. (2025) uses a full-stokes ice dynamics model together with an energy-balance model to simulate the glacier evolution from 1997 to 2100 for two well-known and well-studied glaciers in Central Europe, the Greater Aletsch Glacier in the Western Alps and the Hintereisferner in the Eastern Alps. By using observed ice velocities which were, in case of HEF, manually extracted just for this study, the basal friction parameter and rheology parameter were constrained. Observed MB gradients were used to tune the SEB albedo parameters. In general, they find that HEF is disappearing no matter which scenario is chosen while Aletsch glacier may survive for lower emission scenarios. These outcomes were expected, as they were equally found in other studies. However, the methodological approaches of combining ISSM together with the EB model of Evatt et al. (2015) by including the observed ice velocity and MB profiles to tune some free parameters are novel.

In summary, this manuscript has methodological novelty and directly compares two well-observed glaciers by using observations, model and calibration approaches that haven't been used before. Overall, the topic is interesting, the methods are understandable and the results are described in-depth. It is also great that all data and code will be made available (if I understood it correctly). However, there are a few major aspects that need to be revised, such as some description within the manuscript, an uncertainty analysis of your projections, the comparison to projections from other glacier models, the generalisability of the outcomes to all glaciers in Central Europe and the greater implications of the study. Therefore I believe that the study needs some major revisions before it is ready for publication. I don't expect that all of my comments are directly addressed, but I hope that some kind of uncertainty analysis can still be done together with clarifying some of the

analysis and interpretation. Thanks a lot for this nice manuscript. Please don't feel overwhelmed by my many comments, just try to consider my suggestions when somehow possible.

**Major Comments**

1. Manuscript in general:

    1.1. Overall the manuscript is quite long. Some "figure descriptions" can be condensed as these are already visible in the figures. For example, by always separating HEF from GAG, there is quite a lot of repetition in the manuscript, and it makes it more difficult to actually compare the behaviour of the two glaciers which is one of the actual motivations of the paper, or not? Would it be possible to reduce the amount of text by directly comparing the two glaciers? In addition it seems like sometimes the same things are described in different ways (for HEF, the "mostly gone" definition was introduced, while for GAG, you write again "... ice volumes drops below 10%" …

    We agree, the paper is quite long, but we are a little bit in a conflict for shortening the manuscript: On the one hand you say the manuscript is quite long, on the other hand you (and also the other reviewers) suggest a lot of new analyses (e.g. parameter sensitivity) that comes along with new figures and text. Reviewr 1 suggest shortening of section 7.2 and 7.3. Based on your and the other reviewer comments, we tried to keep the text concise by deleting some material (e.g. moving of L-curve plots to Appendix).

    1.2. I found a few statements that are without references or proofs. Please double-check these (see my line-by-line comments). Some passages within the discussion and conclusion could be a bit more concrete and related to your specific study.

    Done, in most instances we followed your suggestions. See answers to your line comments.

    1.3. See more suggestions in the line-by-line comments.

    Done, in most instances we followed your suggestions. See answers to your line comments.

2. Added value of this study and comparability to other glacier model simulations

    2.1. What was the goal of this study? Creating the "most robust" glacier projections of GAG and HEF by using a full-stokes model together with an EB model? Or was it to understand if it results in other estimates than existing glacier models? On line 75, you write that the aim is to "investigate the potential variability of glacier responses in the frame of physical process representation". I did not feel like this was the aim of your paper. You analysed two glaciers (see 'extrapolation comment' below) and did not look into the uncertainties within your approach (see 'uncertainty comment').

Our aim was to set up a model that is based on the most accurate description of ice flow (full-Stokes) and surface SMB calculation (energy balance model). Although we use the most robust physical description, we don't expect our results to be the most reliable description since e.g. (1) the SMB model is still subject to climate data uncertainty and unconstrained parameters; (2) the full-Stokes model is influenced by the initialization approach. We tried to make the main goal clearer in the new version of the manuscript by writing:

*"In this paper, our main aim is to perform glacier projections that rely on the most robust physical description on ice flow and surface mass balance calculation. Such a computer-intensive work complement the current research of large-scale glacier volume projections of Central Europe with an individual glacier evolution model based on a higher complexity to investigate the potential variability of glacier responses in the frame of physical process representation."*

2.2.    In Sect. 7.2, you do a comparison to other glacier models which is great. Though, it doesn't seem like your study's goal was to directly compare results to other glacier models. You used other weather-station corrected ERA5 climate data, other climate models (with/without EURO-CORDEX), calibration data, initial volumes, RGI versions, (see line by line comments) which makes comparisons very difficult to interpret.

Indeed, comparison is very difficult due to the various approaches. However, comparing our results to other modelling efforts is a logical consequence of our study and is very useful in order to demonstrate the variability of model outcomes that share the common goal of glacier projections. Therefore we conclude that standardized tests are mandatory for a meaningful comparison (Line 533 of current manuscript version).

2.3.    Conclusion l. 528 onwards: I completely agree that it is difficult to compare your estimates to other model estimates. But this entire paragraph somehow also describes the limitation of your study, so I am wondering why you did not choose to do standardised tests in your study, e.g., by comparing directly SIA vs full-stokes. Of course, this is difficult to do, as SIA flowline-models are not really expected to match ice velocities. But, at least showing a range of different full-stokes modelled outcomes (e.g. in terms of equifinality and/or observation uncertainties) may help to understand how variable projections can be within the full-stokes approach.

I fully agree with your concern. But that's not easy to achieve, since ISSM doesn't provide SIA or HO-flowline models similar to what is used in the large-scale/reginal-scale models. However, we do some parameter sensitivity tests that demonstrate the uncertainty in model projections with a full-Stokes model (see answer to your point 3.3).

2.4. What you could for example do is to compare your glacier volume estimates at the RGI year to the ice thickness community estimate from Farinotti et al. (2019). As many large-scale glacier modelling studies use that estimate for their initialisation, this comparison would be nice to have. Another example is that you use RGI7, while most (all) existing modelling studies in Central Europe use RGI6. How different is RGI7 to RGI6 on these two glaciers? All these aspects, together with trying to compare projections from exactly the same climate models (maybe just use one climate model where data exists for all studies) and/or doing an internal uncertainty assessment of your model (see uncertainty comment below) would be necessary to understand a bit better from where the differences come from.

We think a comparison between RGI6 and RGI7 states is far beyond the scope of the paper; and comparison between the exact GCM/RCM is not possible among all studies included in our comparison since JH19 and GloGEMflow don't provide projections for the individual GCM/RCM: they just provide ensemble mean. Well, we could get in touch with the authors, but I think if such a detailed comparison is performed, the authors of the other models should be invited to the paper. However, as we the focus of the paper is not model comparison, we refrain from adding such an analysis.

2.5. If a comparison of the exact same climate models is not possible, then you may reduce the "interpretation" within the discussion Sect. 7.2, and just show the figure with the potential variability of outcomes. In the following are some important aspects for Sect. 7.2 to consider.

2.5.1. You write that under RCP 8.5, JH19 results in more mass loss, while under RCP 2.6 it results in less mass loss. Then, you later argue that this is because JH19 used a TI-model which overestimates melt. Isn't that contradictory? Why is JH19 not resulting in more melt as well for RCP 2.6?

Indeed, this is contradictory. We will clarify this the updated version of the manuscript

If you don't look into the full spectrum of possible MB parameters (equifinality), these statements are very difficult to validate. Specifically you mention in the next paragraph that "Under RCP 2.6, GloGEMow predicts a volume reduction of 57.6% for GAG, aligning somewhat more closely with our forecast than the JH19 study; likewise, for RCP 8.5, GloGEMow projects a 89.4% volume decline, which is close to our estimate than the JH19 study. " Both JH19 and GloGEMflow (Z19) are temperature-index models, but are apparently more different compared to your study and Z19. For me, this shows that other aspects like equifinality or downscaling choices may explain these differences and not specifically the choice of TI-model vs

EB model.

Thanks for your analysis. We will rewrite our interpretation accordingly.

2.5.2.  What I find interesting is for example that OGGM projects regrowth for GAG under SSP1-2.6, but your study did not. However, as you are likely not comparing the exact same climate models, it is very difficult to interpret if this comes from the SIA vs full-stokes choice, the different mass-balance model, climate downscaling choice or actually from the different climate model ensemble where the one shows some regional cooling while the other does not. Is it possible to look into this further?

Yes, we also identified this behaviour but without having detailed model outputs of OGGM it is difficult to asses the source of the differences or of the regrowth. I could also imagine that the regrowth in our simulations is somewhat delayed, i.e. extending the simulations beyond 2100 would be needed to test this. The temperature change in the SSP1.26 and RCP2.6 scenarios is highest around 2060 (Fig. 3), maybe the glacier mass balance recovers afterweards. However, we agree that this is an interesting questions that deserves further analysis, we only aim to demonstrate the variability of the models.

2.5.3.  From where do you have the glacier model evolution data of GAG for the RCP 8.5 / RCP 2.6 scenarios of OGGM. Are these really available from Schuster et al. (2023). I couldn't find them there. Or did you use the OGGM standard projections?

We have not presented any RCP projections of OGGM for GAG in Fig. 13. The line styles may be difficult to distinguish. We only show RCP from our work, JH19 (pers. comm with G. Jouvet) and GloGEMflow (available from their Supplement).

For the SSP scenarios I used the data of Schuster et al. (2023). They are available on zenodo: https://zenodo.org/records/7660887. The link is provided in their paper.

I assume you show the ensemble median over the different five climate models and over the various calibration approaches and TI model choices?

Yes.

If you can compare the exact same climate models: Is the variability of ensemble members from Schuster et al. (2023) similarly large as the differences between your study and Schuster et al. (2023)?

We updated Figure 13 by just showing the median of the same GCM models as in Schuster et al. (2023) to facilitate the comparison. Overall, it didn't change much.

We agree with the reviewer that there are several aspects in the model comparison that requires further attention. In our study we refrain from performing a detailed comparison for various reasons (see comments above), however, we clearly state in the new version of the manscuript that the comparison is intended to show the variability of model outcomes without further analysis.

3. Missing uncertainty estimates:
    3.1. In the abstract, you claim that you give a well-constrained estimate of HEF and GAG projections that complements large-scale modelling efforts. Although you do use more complex and physical models (full-stokes ice dynamics, EB SMB modelling), I am wondering about the uncertainties in your input data and the equifinality of the additionally introduced free parameters. In my opinion, analysing whether different parameter combinations within the "input uncertainty and equifinality" space affect the projections would be essential to understand how well-constrained your projections are. You mention some of these uncertainties qualitatively in the discussion, but you haven't done any quantitative uncertainty assessment yourself. Doing this is difficult, but at least a rough analysis with multiple model estimates is necessary in my opinion.

    Based on this comment, comment 3.2 and 3.3 we introduced a new chapter in the Discussion "Parameter Sensitivity". We simply varied a) the downscaling gradients for temperature, precipitation, LW and SW radiation by +/- 10%, b) the albedo +/-10%, and c) also the error estimate (inferred by comparing the time series of temperature and precipitation of the ERA5 and meteorological station datasets) by +/- 10%. This parameter-ensemble also shows reasonable SMB profiles when compared to observations (see Figure below), but with profiles well above/below the observed mean but within the max/min range of the observations.

    For the parameter-ensemble test, we re-run the RCP 2.6 and SSP585 scenarios (considered as the upper and lower bounds) with the GCM/RCM combination closest to the ensemble median. For each glacier we count 16 simulations. We will present the results as follows

and describe them accordingly in the new introduced section.

[Figure]

**Figure 12.** Computed SMB profiles of the parameter ensemble (red lines) compared to the best estimate (blue line, see Fig. 5) for HEF for the period 2001–2013 (a) and GAG for the period 2011–2019 (b). The grey shading shows the observed minimum and maximum and the black dashed line the mean over the respective period. Note the different scales of the x- and y-axis for HEF and GAG, respectively.

[Figure]

**Figure 13.** Glacier volume projections of the parameter ensemble for HEF (a) and GAG (b). The ensemble median (thick lines), the 17th-83rd percentiles (dark shaded areas) and total range (light shaded areas) is shown.

3.2.     For example, aren't there any other parameter combinations that result in a similar performance as the chosen one for the albedo calibration? If yes, how does that equifinality influence the validation and projection results?
see answer to 3.1

3.3.     By using an EB instead of a temperature-index model, you also have an increased equifinality due to the large amount of introduced new free parameters that need to be downscaled (SW, LW radiation, temperature, surface wind speed, humidity, precipitation). You mention the downscaling uncertainties in the discussion (l.427 to l438) but do not really do anything about it. I am wondering if the quality of the data is sufficient to apply an EB model without accounting for its uncertainties. For example, on l.173, I was thinking: How certain are the radiation gradients? And, isn't it strange to use a precipitation gradient for HEF that was only used before for GAG?
see answer to 3.1

3.4.  The data assimilation does not at all account for the uncertainties in the ice velocity. How strongly can you trust your ice velocity observations? Below we show the velocity map of Millan et al. (2022). Although it is not directly comparable to our remotely-sensed velocity map (different time stamp: 2021 (Millan) vs 1997/98 (our study)), the Millan product shows overall a somewhat noisy pattern without the valley-shaped velocity pattern which seems unreasonable.

[Figure]

Can the velocity extraction approach similarly be applied to other glaciers? How different is this approach to e.g. Millan et al. 2022? Yes, of course. The DInSAR method can be applied to every glacier within the restrictions of the acquisition geometry i.e. very low sensitivity of North-South displacements for (near-)polar orbiting SAR satellites. SAR specific effects of foreshortening, layover and shadow may mask out some interesting parts of glaciers. Depending on the spatial resolution of the used SAR system, very small glaciers might be difficult to monitor.

3.5.  Of course you can't check all of these aspects. I am just wondering if the glacier projections are sensitive to these assumptions? Are these choices less important than the choice of using SIA vs. full-Stokes or a TI-model instead of an EB
That's an important question, but I think we cannot answer this question with our study. Unfortunately, there is no TI model implemented in ISSM to perform a standardized test on surface mass balance modelling. Although ISSM provides a SIA solution, the inversion for the friction coefficient (and rheology coefficient) is not implemented for SIA.

4.  East-West comparison (Sect. 7.3)
     4.1.  Isn't the longer lifetime of GAG also the higher elevation-area distribution (see Fig. 14)? Not just the larger elevation range? It seems like you partly mention that aspect in l. 509 onwards, but it would make sense to condense these things all together. Specifically, GAG has

most of its area around 3500 meters, while HEF has it more around 3000 meters.

We add "higher elevation-area distribution" and slightly rewrote the section to condense the findings together.

4.2. These area-elevation distributions may also explain why the MB gradient is difficult to match at the upper part of HEF (Fig. 6): there is actually not so much area at the highest elevation-bands, which is quite different to GAG. Does this specific shape of the area-elevation distribution also explain parts of the response differences of the two glaciers.

Of course, the area-elevation distribution defines the lifetime of the glacier. We will rephrase our discussion to clarify this accordingly.

4.3. What is the influence of the glacier slope? Is the current HEF steeper than GAG, and how will that evolve in the future? The different glacier geometries could e.g. be mentioned in l. 504 - 508.

We add this as a caveat into the discussion: "*In addition to the regional setting of each glacier that probably influences the glaciers' response to increasing temperatures, the glacier sensitivity depends on glacier slope and exposition ice thickness and area-elevation distribution, mass balance gradient and hypsometry (e.g. Oerlemans, 1992; Jiskoot et al., 2009)."*

5. Extrapolation of HEF and GAG results to all over Central Europe:

5.1. I find it difficult to extrapolate results from HEF and GAG to the entire Alps. You could check how representative HEF and GAG are by analysing their characteristics compared to all other glaciers in the Alps, and/or compare per-glacier model estimates from regional-scale glacier models of HEF/GAG to regional-scale estimates. Maybe GAG is "representative" in some sense for the entire glacier mass evolution as it represents a relatively large fraction of the entire glacier mass in Central Europe, but this is not mentioned/analysed here. In addition, the "representativeness" also changes over time.

We agree, that the extrapolation of GAG and HEF to the entire alps is very difficult. To justify our rough extrapolation of general glacier shrinking we show a figure of the area-elevation distribution of the western and eastern Alps (based on RGI6.0) in Figure 15 and rephrase the discussion accordingly.

[Figure]

**Figure 15.** Area-elevation distribution of GAG (a), HEF (b) and all glaciers in the western and eastern Alps . Distribution of glacier area per 100 m and 50 m elevation bands for and GAG (a) and HEF (b), respectively, for the year 2023 (with respect to bottom x-axis). Coloured lines show the evolution of the ELA of the four scenarios SSP5-8.5, SSP1-2.6, RCP 8.5 and RCP 2.6. (with respect to top x-axis). Distribution of glacier area per 100 m and 50 m elevation bands for all glaciers in the western and eastern Alps based on RGI6.0 (Farinotti et al., 2009)

5.2.  l. 502: "In the worst scenarios, most glaciers also disappear at the end of the 21st century". To my knowledge, almost all (or even all) published studies over Central Europe write about glacier mass evolution, not number of glaciers. From which source do you have this estimate?

We apologize, the sentence was misleading. We changed to "*However, in the worst scenarios, we extrapolate that most glaciers in the western alps disappear at the end of the 21st century that corroborates with Van Tricht et al. (2025).*"

5.3.  Conclusion on l. 524: "Our findings suggest that glaciers in the eastern European Alps are likely to diminish by the mid-21st century, and only larger glaciers with higher elevation ranges in the western European Alps will remain until the end of the century.": Can your study really prove that sufficiently? Please consider removing or rephrasing.

We rephrase the sentence to "*Our findings indicate that glaciers in the eastern European Alps are likely to diminish by the mid-21st century, and only larger glaciers with higher area-elevation distribution will likely remain until the end of the century.*"

5.4.  I am missing a little bit the discussion that your approach is only possible for these two glaciers because of the available data (good-quality remotely-sensed ice velocities and MB profile). Specifically because you do not account for the uncertainties. Maybe add a paragraph on the following aspects: Could your specific model combination approach (using ISSM and a SEB model) also be used for follow-up studies on the two glaciers or on other glaciers? How can your study be used to improve/validate regional-scale glacier modelling/projections?

We added a new paragraph to the chapter Generalization (formerly East-West comparison): "*The generalization of our modelling results to*

> *a larger area is based on a specific model combination approach applied to two individual glaciers. Further modelling attempts based on standardized tests are necessary to infer how our modelling approach can be used to improve regional-scale glacier projections. In addition, follow-up studies focussing on glaciers with a sufficient data basis for our approach facilitate the generalization or extrapolation to a larger area.”*

6. Novelty of the data assimilation approach (l. 78): I am not an expert in data assimilation, but isn't e.g. IGM (Jouvet et al. (2023) or Cook et al. (2023)), Jouvet et al. (2019), also using observed ice velocities for the inversion? Or is it the specific approach with ISSM that is novel? Please specify.
   You are right, IGM also assimilate observed ice velocities. We rewrite the sentence to *“The latter is a common initialization approach for large-scale ice sheet modelling (e.g., Goelzer et al. 2020), but rarely applied for mountain glaciers.”*

7. Climate model choice and scenario descriptions:
   You compare RCP scenarios to SSPs. You show that RCP2.6 & RCP 8.5 result in less warming over Europe than SSP1-2.6 & SSP5-8.5, and with that in less glacier volume loss. Is the reason for these differences the EURO-CORDEX downscaling of the RCP scenarios, or do you see the same effect globally, i.e. does the chosen climate model ensemble result in less global warming for RCP2.6/RCP8.5 vs. the SSP scenarios? That means, please mention the median global warming from the different scenarios for your chosen climate model ensemble (with that you may also rephrase l.421 to l.426 and other related line-by-line comments).
   Unfortunately we cannot perform such a comparison as the EURO-CORDEX data are not available globally. By definition, they are downscaled by an RCM using a GCM output as boundary condition and therefore only available over Europe. Therefore, Lines 421 to 426 cannot be reformulated. See also our answer to your comment to Line 11-12 and 41.

8. EB model calibration approach with calibration vs validation data
   8.1. (Table 1 & Fig. 7, Sect. 5.2 & 5.3):
       8.1.1. Do I understand correctly that you use the average observed SMB gradient from each year (i.e., n=13, 8 gradient observations). That means you do not use the entire MB profiles, but the timeseries of MB gradients and not just the average MB gradient over the time period (i.e., one observation), correctly? I think it would be important to clarify a bit more this calibration procedure within Sect. 5.2.
          We use the mean over the corresponding period. We clarified by including the following sentence: *“Note, that we compare the mean over the period and not each individual year as we are interested in the long-term behaviour.”*

8.1.2. I am specifically asking because I was first astonished to see how well the cumulative MB is matched. If you used the entire SMB profile timeseries, the cumulative MB over that same time period would not be anymore a true "validation" as it is somehow an integrated value of the MB profile? But, as you wrote in l. 340 to l. 342 that you used independent model fields, I expect, you only used the gradients (such as e.g. 0.007 m w.e. m$^{-1}$) without an information of e.g. the intercept. These MB gradients, are, I guess, sufficiently "independent" enough from the cumulative MB. But, are they also completely independent from the elevation change observations?

We apologize for the misleading use of "gradient". This is common language when looking at an SMB profile. We corrected this by using SMB profile instead of SMB gradient. Indeed, as we use the SMB profiles for tuning our SMB model. our transient model is not fully independent to the observed MB dataset, as this is an integrated result of the observed SMB profiles. However, we want to stress, that we are able to match the MB time series of each glacier without tuning the modelled SMB profiles for each individual year. We clarified this in the updated version.

8.1.3. Related to that (specifically on l. 322): How much does it make sense to match the gradient over the entire elevation-area distribution, if there is at the upper part only little glacier area (in case of HEF)?

Of course, matching the SMB profiles that only cover little glacier areas, or becoming less important due to glacier retreat, isn't very important. However, matching the overall SMB behaviour ensures that the SMB profiles reveals a trend that is in-line with the elevation range.

**Minor Comments**

- maybe replace always ice volume/ice area to glacier volume/area (all over the text). At the moment it is a mix of both.

Done. We use "glacier".

- In all maps (Fig. 5, 7, 8; always a,b), it would be quite interesting to see the differences between modelled estimates and observations. I guess this is not always possible, as the observations are partly only available on point scales, but at least where possible, this would be maybe great to at least have a look into instead of the modelled values. The scatterplots are relatively difficult to interpret due to the many overlapping dots.

For Fig 5, we refrain from showing a difference map. The scatter plots basically demonstrate, there is almost no difference. But we agree, for 7 and 8 a map showing the difference might help, however, we put these Figures in the Supplement.

**Abstract:**
l.2: maybe mention here also the used surface mass balance model complexity?
Done

l.8: "comprehensive glaciological observations"

→ maybe clarify that you use in-situ MB gradients for calibration

Well, we do not use MB gradients only. With 'comprehensive' we include all data used in the initialization and tuning approach. This encompasses glacier geometry (surface and bed), surface elevation changes, MB changes, and SMB gradients.

e.g. l. 11-12: maybe mention the median global warming in 2100 from these RCP/SSPs, otherwise it is unclear to understand. The choice of climate models strongly influences the actual warming within one RCP/SSP scenario …
We see the reviewers point, and agree that providing the global warming values of the corresponding climate projection would help to put it in the right context. However, for the EURO-CORDEX simulations no global warming values are available as these are regional model simulations over Europe. We could provide values for the ISIMIP3b simulation, but maybe that will be a bit confusing if only values for the SSPs are provided.

l. 15-16: "...; however, a rough model-intercomparison study reveals a large spread of volume projections with the different glacier models.": would be great to compare this to a model-internal projection spread if you can do this additional analysis (see major comments)
This is impossible. The answer to the major comment.

**Introduction:**
l.20-26: references missing , a bit "vague" …
We don't think it is necessary to provide references on these topics as it is general knowledge; providing references has also no relevance for our paper.

l.28-32 vs l.32-35: in the first sentence you mention different applications (not the models "per se"). Therefore, I would recommend to write "regional-scale/global-scale projections".
Done.
You also present examples, therefore, it would be good to add "e.g. "... maybe also mention e.g. an IGM study (e.g., Cook et al., 2023) as that one emulates full-stokes.
Done. See answer to line comment "p.2, l. 28-43" of Reviewer 1.

l.32: Maybe remove "due to computational constraints", because there are other reasons such as the only poor velocity data available for the inversion on a regional to global scale compared to HEF and GAG which are one of the best monitored glaciers world-wide
Done

l.35: "In most of these models": well, actually only in two of the four mentioned models. Please adapt and differentiate between retreat parameterizations (GloGEM, PyGEM) versus SIA (GloGEMFlow, OGGM).
Done.

l. 37: "Despite potential shortcomings ...": when doing regional projections, there are always shortcomings, but these are often due to missing glacier observations, thus consider reformulating.
With shortcomings we referred to the ice dynamic representation, as we said "Despite potential shortcomings due to a reduced representation of ice dynamic …". The sentence don't say anything about missing data; it is just about the physics in the model. We don't reformulate.

l. 38-39: RCP8.5/SSP5-8.5 is by today's standards a very high-emission scenario resulting in more than 4.0°C global warming in 2100. It could even be 4.5°C depending on the choice of climate models. I would suggest adding "very" to high-emission scenario.
Done

l. 41: "or the low-emission scenarios RCP 2.6 and SSP1-2.6, which are basically in line with the global warming target of 1.5°C ... (UNFCCC, 2015)": Please double-check the resulting global warming of your ensemble of chosen climate models, I am pretty sure it is higher than 1.5°C.
We don't check global warming (by the way it is not available for the EURO CORDEX ensemble, as it is a regional setup). However, we just wanted to say that the SSP1.26 and rcp26 scenarios are the closest climate pathway to meet the Paris agreement. Therefore, we change "... which are basically in line with the global warming target of 1.5°C .." to "... assumed in line with the global warming target of 1.5°C …".

l.44-47: some references?
See our answer to your comment line 20-26.

l.58-62: references are missing (daily vs. monthly available in Schuster et al., 2023, bias-correction in Weathers et al. (2025)). What about the influence of equifinality (Rounce et al., 2020; Schuster et al., 2023)? These may also be very important sources of uncertainties.
Done.

l.75 & l.81: Related to one of the 'Major Comments': "To investigate the potential variability of glacier responses in the frame of physical process representation" --> you compare two different glaciers, how largely do they represent the variability of glacier responses within Central Europe? Maybe the "aim" should be reduced a little bit? Similarly you write later: "In order to capture and analyse regional differences of ice volume loss, the model is applied to two valley glaciers in the Alps". By just choosing two glaciers, you can't capture the regional differences. Please adapt.
Done. See answer to major comment.

l.77: SMB is not yet defined, I think.
Done

Fig. 1: for GAG another time period is chosen than for HEF, isn't there a common time period?
Are you referring to the outlines or to the DGMs? However, for both datasets we do not have a common period; we don't think it is relevant here, as both datasets just demonstrate the current glacier retreat without comparison to each other.
Also mention in the caption that these are the outlines from 1850 until 2015 (I guess they are not all coming from RGI7?). From which study do these outlines come from?
The outlines are all taken from RGI7. Why do you guess they are not from RGI7?
At the dz legend: --> is the unit "m w.e. yr-1"?
No, it is just m/a. It is the difference between the surface elevation of two DGM states. We don't know anything about the density.

**Study sites**
l. 90: "outlines from the last decades": maybe replace by "outlines from 1850 until 2015"
Done

l. 94: "15km³, 900m": at which year? Also in 1999, if yes, consider joining the sentence together with the last sentence to clarify this.
Done
Maybe also mention the percentage of volume relative to the entire Central Europe (I think around 10%)?
Done.

l. 97: What means "relatively dry", "very large amounts of precipitation"? Any quantitative numbers?
Values of precipitation are now added. See also answer to major comment 1.2 by Reviewer 2.

l. 107: HEF has less than 5% of the glacier volume of GAG. Maybe it is worth mentioning that?

Well, isn't that clear from the ice volumes provided in Lines 94 and 107?

Eventually consider comparing these estimates to Farinotti et al. (2019) as many glacier evolution models use this estimate. What is the relative amount of glacier volume loss in a relatively recent but common time period for the two different glaciers?

We dont think that tells us more that the comparison of the observed MBs (Fig. 7c).

l. 111: Can you be more concrete? How many years are available for each of the glaciers? This is hard to see from Fig. 2.

Done. We added "(covering the period from 1960 to present-day).

Figure 2: What means "b"? Maybe rather use SMB as in other parts (similar issue in e.g. Fig.7c).

Done. "b" is replaced with "SMB".

Annual SMB should be in mm w.e. yr-1 ?

No, it is m w.e. yr-1

… "since 1961": from which year to which year do you show the MB estimates?

It is from one year to the next, i.e. 1961-1962, 1962-1963, …., 2022-2023. We added *"since 1961 to 2023"*.

Could it be of interest to estimate with a scatterplot and a spearman rank correlation coefficient how much the two correlate? Consider increasing the size of the figure and eventually even adding a scatterplot to the left of it showing on the x-axis GAG annual MB and on the y-axis HEF annual MB.

We think investigating the observed MB correlation is far beyond our study. We just show these graphs, to demonstrate in the Intro that the glacier underwent a dramatic change; and the cumulative MB of each glacier is later used for validation.

l.125-127: How representative are these precipitation data? It is quite far away from the actual glacier, but I know that there is no better possibility.

As you said, there is no better possibility for this kind of bias-adjustment. Indeed, choosing the climate data of the nearest meteorological stations is similar to Jouvet and Huss (2019). However, in order to highlight this shortcoming, we add *"Both locations, particularly Lauterbrunnen, are situated at the edge or even at the north of the glacier and might not be fully representative for precipitation sums at the glacier, but we rely on these data because of the absence of a better recording."*

l. 130: "almost minor mass balance changes" -->when you look at the cumulative MB, it is still relatively considerable for HEF at least.

We clarified: *"... and almost minor cumulative mass balance changes when compared to the period beyond 1990 …"*

l. 131: How do you justify a precipitation gradient?

It is a well known fact that precipitation changes with elevation in mountainous regions.

l.132: typo in "regionla"
Done

l.154: Maybe also mention over which period you compare ERA5 with the raw GCM/RCM outputs for the "bias adjustment"?
Done, this part is substantially rewritten. See Answer to comment 1.1 of Reviewer 2
Another comment: Could you call the temperature lapse rate/precipitation gradient the "statistical downscaling"? And the correction of the GCMs/RCMs then the bias adjustment? Just to better coincide with how the different steps are called in other studies.
Good point, we will follow the reviewers suggestion when rewriting this part of the manuscript. See Answer to comment 1.1 of Reviewer 2

l.158: As you use ERA5 corrected by weather stations as a dataset for the calibration, isn't another reason why you do the additional bias-adjustment for the ISIMIP GCMs the usage of weather-station corrected ERA5 instead of W5E5? Or not?
We dont understand this question.

Figure 3: Over which area is this? Averaged over the two gridpoints of HEF and GAG?  I guess it is over the Alps, as described in line 160. Please clarify the exact region over which you average.
Indeed, this information is missing. It is over the Alps as mentioned in the text; we add this information to the figure caption
As precipitation is shown as a ratio, the unit should be removed. Consider writing "Precipitation ratio" or "Precipitation relative anomaly". Consider also adding the amount of ensemble members to the legend. I guess this is n=10, n=10, n=65, n=22.
Done. We followed the reviewers suggestion.
l.163: It is great that you mention the regional temperature increases, but I think comparing the SSP1-2.6 regional warming to the global warming target of 1.5°C is a bit strange. As mentioned in one of the 'Major comments', can you also compute the global warming of this ensemble of GCMs/RCMs and compare that to the regional one?
You are right, we dropped the comparison to the global 1.5° target. As already answered above, the global warming for the EURO-CORDEX simulations are not possible to compute.

l.167: I would suggest adding this information directly into the statistical downscaling paragraph of l. 130. When I first read l. 130, I thought that you did not correct the radiation terms, for example.

Good point, thanks. However, we rephrased the description of the climate data (see Answer to major point 2.2 of reviewer 2), but take your suggestion into account.

l. 212: I am not sure what the consequences are of setting the minimum ice thickness to 5m. Does that mean that an elevation where the glacier has an ice thickness below 5 m is considered as ice-free?

Yes, ice thicknesses reaching this threshold value are considered as ice-free. This is explained with the (de-)activation method in line 215. Setting a minimum ice thickness is standard when working with vertical extruded meshes. If we allow a zero ice thickness, the vertical layers would collapse and cause numerical instability. We don't think that needs extra explanation.

l.230 onwards: over which time step is the albedo updated? The SMB profiles are only available on an annual scale

I agree, this is not well explained in the text. We aim to improve the description of the SMB calculation in the updated manuscript. The albedo (i.e. SMB) is updated every year.

l. 257: What does 0.05 and 0.25a mean, does 0.25a mean a geometry update four times a year?

Yes, the timestep defines the update of the masstransport model. We slightly reformulate the sentence.

Table 1: HEF SMB gradient: shouldn't the reference be WGMS (2024) as the elevation-band mass-balance estimates are published there, or not?

Well, maybe the SMB gradients of HEF are also published in the WGMS database (I didn't checked that). However, we use the dataset published on PANGAEA (that is the reference Fischer et al. (2013) in Tab. 1).

Fig. 5: (a) and (b): For GAG, it seems like low observed velocities are overestimated in the model. It would be nice to see the velocity differences directly on the map.
We put a difference plot in the Supplement.
Make sure to use a consistent description of the units (double-check with "The Cryosphere" regulations). At the moment, it is mixed ("m a-1" and "m/a"). "c" and "d":
Done.
What do you represent with "vmod-vobs"? Is this the bias, and with that the mean over the differences? If yes, add an "average bar" on top of the equation.
It is the mean signed difference (MSD). We updated the figure accordingly.
RMS or RMSE? Aren't you showing the root mean square error (RMSE) or do you really just show the magnitude of the observed velocity (RMS)? How many data points are used, please add the number.
It is RMSE. We updated the figure accordingly.
Do you believe that using an average over eight years for the initialisation in the year 2011 is problematic for a quickly melting glacier?

I don't understand this point with respect to Figure 5. Average of what? Velocity? For GAG, we have an average of 8 years, but for HEF only two years average … Are you referring to that?

I guess you are referring to Figure 8 with this question where we perform the validation? If so, my answer: I agree, the time period is short for validation. It would be better to cover a larger period (starting before 2011, preferably before the 90s) to validate the model response for different climate states or MB trends. But in the absence of data for the initialization (surface velocity, ice geometry), we are restricted to this method.

Why did you choose a different velocity dataset than for Hintereisferner?

Well, there is nothing other available. The e.g. McMillan dataset is not reliable for HEF (see above)

l. 322: "RMS" shouldn't that be RMSE (similar on other lines before, and in Fig. 7)

Now, we consistently use RMSE. See next comment and Figure legend.

Fig. 6: Consider using the same color/line style scheme as in e.g. Fig. 7 for modelled vs observed estimates. That means maybe use always dashed lines for observations and solid lines for modelled estimates. At the moment, the green line in Fig. 6 is very thin and one has to compare that to the black line? What does the black line mean? Is that the average over the entire period? This is not very clear from the legend (I think the two "observed" labels have to be switched). You could also just have one legend outside of the figures with those labels that are the same and make sure to write out the accronyms. In (c) and (d), the lines seem to be even thinner than in (a) and (b), consider making them thicker. Also, the y-labels are different between (a) and (c). Can you maybe clarify in the figure or caption which period is used for "calibration" and which one for "validation"? Interestingly, I guess by coincidence, the "validation" period is matched better than the calibration period for both glaciers (in terms of RMSE and MSD).

Thanks for the detailed comments to this figure. The Figure indeed contains some minor errors, I apologize for that. The Figure is reworked and now looks as follows. We actually don't use similar x- and y-axis for HEF and GAG but we noted that in the figure caption.

[Figure]

**Figure 6.** Yearly averaged SMB gradients (blue line) for different periods as computed by the EBM compared to observed SMB gradients (black dashed line). The grey shading shows the observed minimum and maximum of the respective period. (a, b) Computed SMB gradients for GAG for the period 1961–1990 (a) and 2011–2019 (b). (c, d) Computed SMB gradients for HEF for the period 1961–1990 (c) and 2001–2013 (d). Note the different scales of the x- and y-axis for HEF and GAG, respectively.

l. 348: Could these localised modelled spots come from e.g. "shading" which is not accounted for in the gridded radiation data?
Well, it could be shading - or more general speaking - a geometry that is still not in balance with the smb and ice flux divergence. Although we did some relaxation (Line >309), the length of the simulation time was chosen to balance the "initial shock" and ensure that the glacier's ice volume did not deviate too much from the initial volume.

l. 366: The "mostly gone" definition and likely range used here seem to be very similar to what is used in the https://goodbye-glaciers.info/ project. Is that a coincidence? Although the "mostly gone" numbers for GAG and HEF are available from that project (based on three large-scale glacier models, https://goodbye-glaciers.info/glaciers/RGI60-11.01450.html, https://goodbye-glaciers.info/glaciers/RGI60-11.00897.html), it does probably not really make sense to compare the numbers directly as other climate models are used, SSP126 is between 1.5 and 2.7°C and you would need to recompute your values to be relative to 2020 instead of relative to 1997.
Indeed, the definition is similar to the "Goodbye glacier" project. However, they also define 'mostly gone' if the area falls below a threshold which we don't. They don't have a definition of 'gone', which we have.

Table 2: please use consistent descriptions. In the text you always had median [q17 to q83], In the table it is a mix of round brackets with sometimes "-" and sometimes "->".
Done
Why do you mention here the initial volume of the glaciers for two different time steps? The "gone" and "mostly gone" definitions are in both cases relative to 1997, or not?
No, the definition of 'gone' and 'mostly-gone' is relative to the projection start date.

As provided in the table caption, HEFs start date is 1997 while GAGs date is 2011. Providing the initial volume is necessary - I guess - to judge the 10% and 1% thresholds, which are different for HEF and GAG, respectively. Of course, we could have a common 1997 start date, but then we have to extrapolate GAGs initial volume from 2011 to 1997.

Fig. 9&10: Why does ERA5 result in slightly less glacier volume loss than SSP1-2.6/SSP5-8.5 for both HEF and GAG? Is that because the bias-correction period is not the same as the ERA5 period? It seems like the RCP (EURO-CORDEX) scenarios match better to the ERA5 simulations for both glaciers. We have also observed this behavior, but cannot explain it. It may be related to the time period selected for bias correction, but we have not tested other periods.

Fig. 10&11: How do you define that a certain part of the glacier is gone? Is this the 5m threshold you mention somewhere in the methods section? Here it would be again great to have the global warming estimates for these two chosen scenarios. Exactly, if the ice thickness falls below 5m, the mesh element is masked to be ice-free. The 5m threshold is a value only needed for stability of the numerical model (see Line 212). The 5m threshold prevents the collapse of vertical layers if the ice thickness exceeds 0m which will cause the model to crash. It's a common problem in most of the ice flow models like ISSM or Elmer/Ice.

l.409 : change "sime" to "some
Done
l. 412: typo -> "methodological"
Done

Fig. 14: It would be extremely helpful to use the same y-scales for the two glaciers. If it is really not possible, than mention this in the caption. The grey barplots are missing in the legend. Consider removing double legends, legends in (a) and (b) show the same, just show it in (a).
Done. We adjusted Figure 14b and updated the legend.

l. 511 to l. 515: You mention at the beginning and at the end the same statement, please reduce one of the two sentences.
Done. We deleted the latter sentence.

l. 518: Did you check how representative HEF and GAG are in terms of valley glaciers in the European Alps? I would rather say, that these two glaciers are among the glaciers with a lot of observations which allows to apply the data assimilation approach using full-stokes and an EB model. But how similar are these glaciers to other glaciers in terms of their area-elevation distribution? Wouldn't you need to check the RGI inventory and some statistics there to evaluate whether they are actually well representative? See related 'Major Comment'.

l. 520: Here it would be great again to know the actual global warming of these scenarios.
It isn't possible to give global warming values for the EURO-CORDEX simulations.

l. 535: What do you mean with "GlacierMIP (2025)"? Maybe mention that the existing GlacierMIP studies (Hock 2019, Marzeion et al. 2020, Zekollari, Schuster et al. 2025) did not focus on that,
Done
but maybe this will actually happen in the next GlacierMIP round (i.e. GlacierMIP4).
Since the protocol of GlacierMIP4 is not yet released, we refrain from speculating whether they will focus on such a comparison or not.

**References:**
Farinotti, D. et al. A consensus estimate for the ice thickness distribution of all glaciers on Earth. Nat. Geosci. 12, 168–173 (2019).

Weathers, M., Rounce, D. R., Fasullo, J., & Maussion, F. (2025). Evaluating the role of internal climate variability and bias adjustment methods on decadal glacier projections. Earth's Future, 13, e2024EF005624. https://doi.org/10.1029/2024EF005624

Rounce DR, Khurana T, Short MB, Hock R, Shean DE, Brinkerhoff DJ. Quantifying parameter uncertainty in a large-scale glacier evolution model using Bayesian inference: application to High Mountain Asia. Journal of Glaciology. 2020;66(256):175-187. doi:10.1017/jog.2019.91

Schuster L, Rounce DR, Maussion F. Glacier projections sensitivity to temperature-index model choices and calibration strategies. Annals of Glaciology. 2023;64(92):293-308. doi:10.1017/aog.2023.57

Jouvet G, Cordonnier G. Ice-flow model emulator based on physics-informed deep learning. Journal of Glaciology. 2023;69(278):1941-1955. doi:10.1017/jog.2023.73

Cook, S. J., Jouvet, G., Millan, R., Rabatel, A., Zekollari, H., & Dussaillant, I. (2023). Committed ice loss in the European Alps until 2050 using a deep-learning-aided 3D ice-flow model with data assimilation. Geophysical Research Letters, 50, e2023GL105029. https://doi.org/10.1029/2023GL105029

HOCK R, BLISS A, MARZEION B, et al. GlacierMIP – A model intercomparison of global-scale glacier mass-balance models and projections. Journal of Glaciology. 2019;65(251):453-467. doi:10.1017/jog.2019.22

Marzeion, B., Hock, R., Anderson, B., Bliss, A., Champollion, N., Fujita, K., et al. (2020). Partitioning the uncertainty of ensemble projections of global glacier mass change. Earth's Future. 8, e2019EF001470. https://doi.org/10.1029/2019EF001470

Zekollari, H., Schuster, L., Maussion, F., Hock, R., Marzeion, B., Rounce, D. R., ... & Sakai, A. (2025). Glacier preservation doubled by limiting warming to 1.5° C versus 2.7° C. Science, 388(6750), 979-983.

Millan, R., Mouginot, J., Rabatel, A. et al. Ice velocity and thickness of the world's glaciers. Nat. Geosci. 15, 124–129 (2022). https://doi.org/10.1038/s41561-021-00885-z

**References**

J. Bloch-Johnson, M. Rugenstein, J. Gregory, B. B. Cael, and T. Andrews, "Climate impact assessments should not discount 'hot' models," *Nature*, vol. 608, no. 7924, p. 667, Aug. 2022, doi: 10.1038/d41586-022-02241-6.

Hindmarsh, R. C. A.: A numerical comparison of approximations to the Stokes equations used in ice sheet and glacier modeling, J. Geophys. Res.-Earth, 109, F01012, https://doi.org/10.1029/2003JF000065, 2004.

Østrem, G.: Ice Melting under a Thin Layer of Moraine, and the Existence of Ice Cores in Moraine Ridges, Geografiska Annaler, 41,495 228–230, https://doi.org/10.1080/20014422.1959.11907953, 1959.

Dirmhirn, I. and Trojer, E., Albedountersuchungen auf dem Hintereisferner 1955, *Archiv für Meteorologie, Geophysik und Bioklimatologie Serie B* , Vol. 6, No. 4 Springer Science and Business Media LLC, p. 400-416, doi: 10.1007/bf02242745

Jiskoot, H., Curran, C. J., Tessler, D. L., and Shenton, L. R.: Changes in Clemenceau Icefield and Chaba Group glaciers, Canada, related to hypsometry, tributary detachment, length–slope and area–aspect relations, Annals of Glaciology, 50, 133–143, https://doi.org/10.3189/172756410790595796, 2009.

Oerlemans, J.: Climate sensitivity of glaciers in southern Norway: application of an energy-balance model to Nigardsbreen, Hellstugubreen and Alfotbreen, Journal of Glaciology, 38, 223–232, https://doi.org/10.3189/S0022143000003634, 1992.

IPCC: Climate Change 2023: Synthesis Report. Contribution of Working Groups I, II and III to the Sixth Assessment Report of the Intergovernmental Panel on Climate

Change [Core Writing Team, H. Lee and J. Romero (eds.)]. IPCC, Geneva, Switzerland., https://doi.org/10.59327/ipcc/ar6-9789291691647, 2023.

*Meredith, M., Sommerkorn, M., Cassotta, S., Derksen, C., Ekaykin, A., Hollowed, A., Kofinas, G., Mackintosh, A., Melbourne-Thomas, J., Muelbert, M., Ottersen, G., Pritchard, H., and Schuur, E.: Polar Regions, in: IPCC Special Report on the Ocean and Cryosphere in a Changing Climate, edited by Pörtner, H.-O., Roberts, D., Masson-Delmotte, V., Zhai, P., Tignor, M., Poloczanska, E., Mintenbeck, K.,845 Alegría, A., Nicolai, M., Okem, A., Petzold, J., Rama, B., and Weyer, N., chap. 3, pp. 203–320, Cambridge University Press, Cambridge, United Kingdom and New York, NY, USA, https://doi.org/10.1017/9781009157964.005, 2019.*

*Oppenheimer, M., Glavovic, B., Hinkel, J., van de Wal, R., Magnan, A., Abd-Elgawad, A., Cai, R., Cifuentes-Jara, M., DeConto, R.,870 Ghosh, T., Hay, J., Isla, F., Marzeion, B., Meyssignac, B., and Sebesvari, T.: Sea level rise and implications for low-lying islands, coasts and communities, in: IPCC Special Report on the Ocean and Cryosphere in a Changing Climate, edited by Pörtner, H.-O., Roberts, D., Masson-Delmotte, V., Zhai, P., Tignor, M., Poloczanska, E., Mintenbeck, K., Alegría, A., Nicolai, M., Okem, A., Petzold, J., Rama, B., and Weyer, N., chap. 4, pp. 321–445, Cambridge University Press, Cambridge, United Kingdom and New York, NY, USA, https://doi.org/10.1017/9781009157964.006, 2019.*